# QUOTIENT-SPACE DIFFUSION MODELS

**Yixian Xu**[1][*]    **Yusong Wang**[2,5][*],    **Shengjie Luo**[1],    **Kaiyuan Gao**[3],    **Tianyu He**[4],
**Di He**[1][†]    **Chang Liu**[5][†]

[1] State Key Laboratory of General Artificial Intelligence, Peking University, Beijing, China
[2] State Key Laboratory of Human-Machine Hybrid Augmented Intelligence, Institute of Artificial Intelligence and Robotics, Xi'an Jiaotong University
[3] Huazhong University of Science and Technology, Wuhan, China
[4] Microsoft Research Asia, Beijing, China
[5] Zhongguancun Academy, Beijing, China

## ABSTRACT

Diffusion-based generative models have reformed generative AI, and also enabled new capabilities in the science domain, *e.g.*, fast generation of 3D structures of molecules. In such tasks, there is often a *symmetry* in the system, identifying elements that can be converted by certain transformations as equivalent. Equivariant diffusion models guarantee a symmetric distribution, but miss the opportunity to make learning easier, while alignment-based simplification attempts fail to preserve the target distribution. In this work, we develop *quotient-space diffusion models*, a principled generative framework to fully handle and leverage symmetry. By viewing the intrinsic generation process on the quotient space, the exact construction that removes symmetry redundancy, the framework simplifies learning by allowing model output to have an arbitrary intra-equivalence-class movement, while generating the correct symmetric target distribution with guarantee. We instantiate the framework for molecular structure generation which follows $\mathrm{SE}(3)$ (rigid-body movement) symmetry. It improves the performance over equivariant diffusion models and outperforms alignment-based methods universally for small molecules and proteins, representing a new framework that surpasses previous symmetry treatments in generative models.

## 1    INTRODUCTION

Diffusion models have emerged as the dominant approach for modeling distributions in high-dimensional spaces. Building on their success in real-world domains such as images (Ho et al., 2020; Song et al., 2021), audios (Kong et al., 2021; Evans et al., 2024), and videos (Ho et al., 2022; Li et al., 2023), diffusion models are now increasingly adopted in scientific applications, ranging from fluid field solving (Bastek et al., 2025), electronic structure prediction (Kim et al., 2025), molecular structure generation (Xu et al., 2022; Abramson et al., 2024; Hassan et al., 2024; Geffner et al., 2025), and thermodynamic ensemble modeling (Zheng et al., 2024; Lewis et al., 2025).

Compared to general applications, scientific tasks often exhibit inherent *symmetry* structures, wherein elements that can be converted by specific transformations are regarded as equivalent. As a representative example, a molecular structure can be represented as a $\mathbb{R}^{3N}$ vector by concatenating the 3D coordinates of its $N$ atoms. Due to spatial homogeneity, molecular structures that differ only by a global 3D translation and rotation (*i.e.*, a rigid-body movement) hold exactly the same physical properties, hence should be regarded as *symmetric/equivalent* in representing the same state, *i.e.*, the shape of the molecule. Mathematically, such transformations typically form a Lie group, *e.g.*, the 3D special Euclidean group $\mathrm{SE}(3)$ in the molecular case, which formally characterizes the symmetry.

The common treatment is assigning the same probability to equivalent elements in the original space, resulting in a group-invariant target distribution. This can be implemented by simply augmenting training data by applying random group actions (Abramson et al., 2024), or using a group-equivariant generation process starting from a group-invariant prior distribution (Köhler et al., 2020; Xu et al., 2022; Hoogeboom et al., 2022b). Nevertheless, these approaches have not leveraged the

---

[*]Equal contribution.
[†]Correspondence to: Di He <dihe@pku.edu.cn>, Chang Liu <liuchang@bza.edu.cn>.

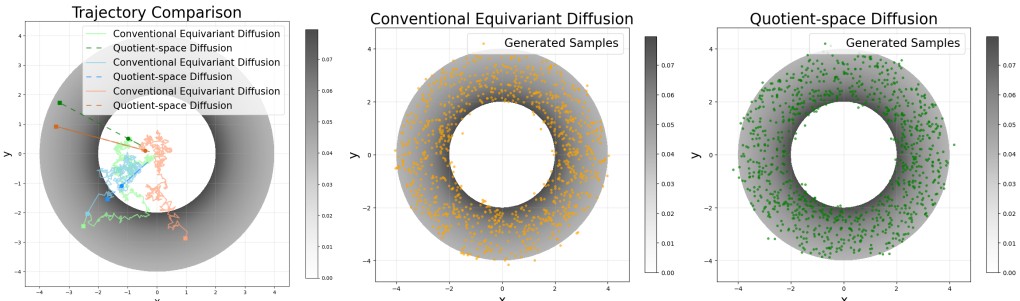

Figure 1: A conceptual illustration highlighting characteristics of the quotient-space diffusion model for a distribution (shown in gray scale) on $\mathbb{R}^2$ with $\mathrm{SO}(2)$ (*i.e.*, rotational) symmetry. **(Left)** SDE sampling trajectories. Same color indicates the same starting point (round dot). The quotient-space diffusion moves only radially along rays emanating from the origin, representing the quotient space $\mathbb{R}^2/\mathrm{SO}(2)$ which traverses across equivalence classes (origin-centered concentric circles) without moving within an equivalence class (angular movement). The conventional diffusion model moves over the whole $\mathbb{R}^2$ space, leading to jagged curves requiring subtler simulation. **(Middle)** Samples generated by the conventional equivariant diffusion model and **(Right)** by the quotient-space diffusion model. Both recover the same target distribution. The quotient-space diffusion also reduces learning difficulty (Eq. (11)) by allowing arbitrary model output in intra-equivalence-class movement (angular movement).

symmetry to reduce learning difficulty, as the neural network for modeling the generation process still needs to learn a specific equivalent movement (*e.g.*, translating and rotating a molecule as a rigid body ($\mathrm{SE}(3)$ action)), which is unnecessary as *any* such a movement does not update the essential system state (*e.g.*, the shape of a molecule). In hope to remove this redundancy for an easier learning task, a few heuristic attempts are proposed by landmark prior works, *e.g.*, GeoDiff (Xu et al., 2022) and AlphaFold 3 (Abramson et al., 2024). They align the target samples with respect to model input or output to reduce their degrees of freedom (DoFs) under equivalence. Nevertheless, we find (Sec. 3.4) these treatments alter the learning target from what is required in the sampling process, hence distort the generated distribution, even with heuristic fix attempts (Wohlwend et al., 2025).

In this work, we develop quotient-space diffusion models, a principled framework for the desire to *reduce learning difficulty* by leveraging symmetry, while *guaranteeing the correct symmetric target distribution* by its sampling process. *Quotient space* treats a set of equivalent elements, *i.e.*, an equivalence class, as one element. It is hence the exact mathematical construction that reflects the intrinsic variability without redundancy (*e.g.*, the shape space of a molecule). We first transform the conventional equivariant diffusion process onto the quotient space, describing the generation process in essential move. Considering that the quotient space is quite abstract and inconvenient to carry out simulation directly, we further leverage the associated construction of *horizontal lift* to pull the quotient-space process back to the original space, making simulation as easy as for the original process. The resulting process effectively projects each original generation update onto the subspace that moves only *across* equivalence classes (*e.g.*, deformation), removing the component that moves *within* an equivalence class (*e.g.*, rigid-body translation and rotation). An additional update term arises from deduction which *guarantees the correct distribution*. Fig. 1(Left) shows a conceptual illustration. In contrast to the conventional equivariant diffusion, which leads to highly jagged paths, the quotient-space diffusion process moves straight only in the radial direction in this rotationally symmetric system, while producing the same target distribution (Middle vs. Right).

Importantly, with the help of the projection, the model output is allowed to be arbitrary in the subspace of intra-equivalence-class movement (*e.g.*, rigid-body translation and rotation), since any output in this subspace would be projected out in each generation step. This *reduces learning difficulty* as the model does not need to learn anything in this subspace. A mechanistic comparison with prior treatments is shown in Table 1. The quotient-space diffusion admits either an equivariant model or a general model with data augmentation in training. Appx. A discusses more related works.

For the representative case of the $\mathbb{R}^{3N}/\mathrm{SE}(3)$ quotient space (shape space), we derive explicit expressions for the concepts, leading to concrete training and sampling algorithms. Particularly, the projection removes the *total linear and angular momentum* of the $N$ points, leaving only deformation updates without rigid-body movement, and relaxes the model from learning to predict the total

Table 1: Mechanistic comparison among diffusion models for handling symmetry. Regarding leveraging the symmetry for easier training, we consider whether a method removes the need to learn to predict a target in the equivalent degrees of freedom (DoFs), and (if not) whether it removes the variance in the equivalent DoFs of target samples. Regarding generation validity, we consider whether the modified learning target has a compatible sampler that produces the correct target distribution. The denoising form $\mathbf{D}_\theta$ of diffusion model is adopted. $\mathcal{A}_\mathbf{y}(\mathbf{x})$ (Eq. (13)) aligns $\mathbf{x}$ towards $\mathbf{y}$, and $\bar{\theta}$ treats $\theta$ as constant (*i.e.*, stop-gradient). See Sec. 3.4 for details.

| Training strategies for $\mathbf{D}_\theta$ | Learning targets of $\mathbf{D}_\theta$ | Reduction of learning difficulty | | Sampling compatibility |
|---|---|---|---|---|
| | | Removal of equivalent DoFs | Removal of variance on equivalent DoFs | |
| Conventional diffusion loss $\mathbb{E}\|\mathbf{D}_\theta(\mathbf{x}_t, t) - \mathbf{x}_1\|^2$ | $\mathbb{E}[\mathbf{x}_1\|\mathbf{x}_t]$ | ✗ | ✗ | ✓ |
| GeoDiff alignment loss $\mathbb{E}\|\mathbf{D}_\theta(\mathbf{x}_t, t) - \mathcal{A}_{\mathbf{x}_t}(\mathbf{x}_1)\|^2$ | $\mathbb{E}[\mathcal{A}_{\mathbf{x}_t}(\mathbf{x}_1)\|\mathbf{x}_t]$ | ✗ | ✓ | ✗ |
| AF3 alignment loss $\mathbb{E}\|\mathbf{D}_\theta(\mathbf{x}_t, t) - \mathcal{A}_{\mathbf{D}_{\bar{\theta}}(\mathbf{x}_t, t)}(\mathbf{x}_1)\|^2$ | $g \cdot \mathbb{E}[\mathcal{A}_{\mathbf{x}_t}(\mathbf{x}_1)\|\mathbf{x}_t]$ for arbitrary $g \in \mathcal{G}$ | ✓ | ✓ | ✗ |
| Quotient-space diffusion loss $\mathbb{E}\|P_{\mathbf{x}_t}(\mathbf{D}_\theta(\mathbf{x}_t, t) - \mathbf{x}_1)\|^2$ | $\mathbb{E}[P_{\mathbf{x}_t}(\mathbf{x}_1)\|\mathbf{x}_t] + \mathbf{v}^\mathcal{V}$ for arbitrary $\mathbf{v}^\mathcal{V} \in \mathrm{Ker}(P_{\mathbf{x}_t})$ | ✓ | ✓ | ✓ |

linear and angular momentum. In the application of molecular structure generation, our quotient-space diffusion models universally outperform conventional diffusion models as well as alignment-based treatments on a diverse range of generation quality metrics for small molecules and for proteins. Notably, with quotient-space diffusion, the 60M-parameter Proteína model (Geffner et al., 2025) even outperforms the much larger 200M-parameter model for protein structure generation.

## 2 BACKGROUND

### 2.1 DIFFUSION-BASED GENERATIVE MODELS ON EUCLIDEAN SPACE

The main idea of diffusion models is to construct a step-by-step transformation from a simple prior distribution $p_{\mathrm{prior}}$ at $t = 0$ to a complicated target distribution $p_{\mathrm{target}}$ at $t = 1$. We follow the Stochastic Interpolant framework (Albergo et al., 2023) unifying diffusion models (Ho et al., 2020; Song et al., 2021) and flow matching models (Lipman et al., 2023; Liu et al., 2023). It defines the intermediate distribution for $t \in [0, 1]$ by:

$$\mathbf{x}_t = \alpha_t \mathbf{x}_0 + \beta_t \mathbf{x}_1 + \gamma_t \boldsymbol{\epsilon}, \quad \text{where } (\mathbf{x}_0, \mathbf{x}_1) \sim p_{\mathrm{joint}}, \quad \boldsymbol{\epsilon} \sim \mathcal{N}(0, \mathbf{I}), \tag{1}$$

where $p_{\mathrm{joint}}(\mathbf{x}_0, \mathbf{x}_1)$ is a pre-defined joint distribution with marginals $p_{\mathrm{prior}}(\mathbf{x}_0)$ and $p_{\mathrm{target}}(\mathbf{x}_1)$, and $\alpha_t, \beta_t, \gamma_t$ satisfy the boundary conditions $\alpha_0 = 1$, $\beta_0 = \gamma_0 = 0$, and $\alpha_1 = \gamma_1 = 0$, $\beta_1 = 1$ so that $p_0 = p_{\mathrm{prior}}$ and $p_1 = p_{\mathrm{target}}$. This distribution transformation can be achieved by the following ordinary differential equation (ODE) (Albergo et al., 2023, Cor. 2.18):

$$d\mathbf{x}_t = \mathbf{v}(\mathbf{x}_t, t)\, dt, \quad \text{where } \mathbf{v}(\mathbf{x}_t, t) := \mathbb{E}[\dot{\alpha}_t \mathbf{x}_0 + \dot{\beta}_t \mathbf{x}_1 + \dot{\gamma}_t \boldsymbol{\epsilon} \mid \mathbf{x}_t], \tag{2}$$

and the dot decoration denotes time derivative. The velocity vector field $\mathbf{v}(\mathbf{x}_t, t)$ is typically trained with the objective: $\mathcal{L}(\theta) := \mathbb{E}_{p(t)} w(t) \mathbb{E}_{p_{\mathrm{joint}}(\mathbf{x}_0, \mathbf{x}_1) p(\boldsymbol{\epsilon})} \|\mathbf{v}_\theta(\mathbf{x}_t, t) - (\dot{\alpha}_t \mathbf{x}_0 + \dot{\beta}_t \mathbf{x}_1 + \dot{\gamma}_t \boldsymbol{\epsilon})\|^2$, where $p(t)$ and $w(t)$ control the sampling frequency and weight over time. The distribution transformation can also be achieved by a stochastic process given by the stochastic differential equation (SDE):

$$d\mathbf{x}_t = (\mathbf{v}(\mathbf{x}_t, t) + \eta_t \mathbf{s}(\mathbf{x}_t, t))\, dt + \sqrt{2\eta_t}\, d\mathbf{w}_t, \quad \text{where } \mathbf{s}(\mathbf{x}_t, t) := \nabla_{\mathbf{x}_t} \log p_t(\mathbf{x}_t) \tag{3}$$

is called the score function, and $\eta_t \geqslant 0$ controls stochasticity, and $\mathbf{w}_t$ is the Wiener process (Albergo et al., 2023, Cor. 2.10). When $p_{\mathrm{prior}} = \mathcal{N}(\mathbf{0}, \mathbf{I})$ (Albergo et al., 2023, Def. 3.4), $\mathbf{x}_0$ and $\boldsymbol{\epsilon}$ can be combined and $\mathbf{x}_t = \hat{\alpha}_t \boldsymbol{\epsilon} + \beta_t \mathbf{x}_1$, where $\hat{\alpha}_t = \sqrt{\alpha_t^2 + \gamma_t^2}$, and the score function can be expressed by the velocity field: $\mathbf{s}(\mathbf{x}_t, t) = \frac{\dot{\beta}_t \mathbf{x}_t - \beta_t \mathbf{v}(\mathbf{x}_t, t)}{\hat{\alpha}_t (\dot{\hat{\alpha}}_t \beta_t - \hat{\alpha}_t \dot{\beta}_t)}$. A common convenient parameterization for a $\mathbf{v}_\theta(\mathbf{x}_t, t)$ model is to use a neural network $\mathbf{D}_\theta(\mathbf{x}_t, t)$ in the following way, which turns the objective into:

$$\mathbf{v}_\theta(\mathbf{x}_t, t) := (\dot{\hat{\alpha}}_t / \hat{\alpha}_t) \mathbf{x}_t - (\dot{\hat{\alpha}}_t \beta_t / \hat{\alpha}_t - \dot{\beta}_t) \mathbf{D}_\theta(\mathbf{x}_t, t), \tag{4}$$

$$\mathcal{L}(\theta) = \mathbb{E}_{p(t)} w(t) (\dot{\hat{\alpha}}_t \beta_t - \hat{\alpha}_t \dot{\beta}_t)^2 / \hat{\alpha}_t^2\, \mathbb{E}_{p(\mathbf{x}_1, \mathbf{x}_t)} \|\mathbf{D}_\theta(\mathbf{x}_t, t) - \mathbf{x}_1\|^2, \tag{5}$$

where $p(\mathbf{x}_1, \mathbf{x}_t)$ is derived from Eq. (1) by integrating out $\mathbf{x}_0$ and $\boldsymbol{\epsilon}$. This objective conveys the intuition of predicting the clean-data sample $\mathbf{x}_1$ from a noisy sample $\mathbf{x}_t$, hence $\mathbf{D}_\theta(\mathbf{x}_t, t)$ is called a denoising model and suits prevalent architectures.

## 2.2 FROM MANIFOLD TO QUOTIENT SPACE

Symmetry can be formalized as identifying elements that can be converted to each other by a certain set of transformations. Such transformations typically form a Lie group, which is also a manifold. We consider the general case where the state space is a manifold. Fig. 2 illustrates the concepts.

**Manifold and tangent vectors** (Appx. B.1). A (smooth) manifold $\mathcal{M}$ is a space is locally isomorphic to a Euclidean space. It generalizes the Euclidean space to allow spatial heterogeneity and a different global topology. The velocity of movement at a point $\mathbf{x}$ on $\mathcal{M}$ is represented as a *tangent vector* at $\mathbf{x}$. All tangent vectors at $\mathbf{x}$ constitute a linear space $T_\mathbf{x}\mathcal{M}$ with the same dimension called the tangent space at $\mathbf{x}$. In contrast to the Euclidean space, tangent spaces at different points are different linear spaces, but a smooth transformation $F$ between the points (even on different manifolds) can induce a *push-forward* map $F_{*\mathbf{x}} : T_\mathbf{x}\mathcal{M} \to T_{F(\mathbf{x})}\mathcal{M}$ between the tangent spaces by linking infinitesimal movements around $\mathbf{x}$ and around $F(\mathbf{x})$. It can be seen as a generalization of the Jacobian matrix. A manifold is typically endowed with a *Riemannian metric*, *i.e.*, an inner product in each tangent space. It further induces common constructions like curve length, distance, measure, gradient, Laplacian, and the Wiener process (Appx. B.2) on the manifold.

**Lie group** (Appx. C.1). A Lie group is a group that is also a manifold, also regarded as a continuous group. Its role to characterize the symmetry of a state space $\mathcal{M}$ is by its (left) *group action* $L_g : g \mapsto g \cdot \mathbf{x}$ on $\mathcal{M}$ (*e.g.*, translating and rotating a molecular structure when $\mathcal{G} = \mathrm{SE}(3)$). To comply with the symmetry, we ask the Riemannian metric to be invariant under group action (or, the group action is *isometric*): $\langle \mathbf{v}, \mathbf{v}' \rangle_\mathbf{x} = \langle (L_g)_{*\mathbf{x}}(\mathbf{v}), (L_g)_{*\mathbf{x}}(\mathbf{v}') \rangle_{g\cdot\mathbf{x}}$, for all $\mathbf{v}, \mathbf{v}' \in T_\mathbf{x}\mathcal{M}$. Symmetry of con-

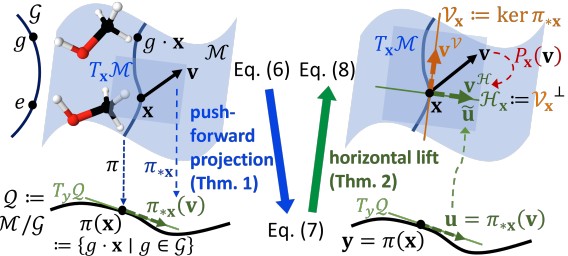

Figure 2: Illustration of mathematical concepts, their relations, and the scheme of technical development.

structions on $\mathcal{M}$ can then be characterized by $\mathcal{G}$: a distribution $p$ is said $\mathcal{G}$-*invariant* if $p(g\cdot\mathbf{x}) = p(\mathbf{x})$, $\forall g \in \mathcal{G}, \mathbf{x} \in \mathcal{M}$, and a vector field $\mathbf{v}$ is $\mathcal{G}$-*equivariant* if $(L_g)_{*\mathbf{x}}(\mathbf{v}_\mathbf{x}) = \mathbf{v}_{g\cdot\mathbf{x}}$.

**Quotient space** (Appx. C.2). The group action defines a symmetry/equivalence relation on $\mathcal{M}$: $\mathbf{x} \sim \mathbf{x}'$ if there exists $g \in \mathcal{G}$ such that $g \cdot \mathbf{x} = \mathbf{x}'$. The quotient space $\mathcal{Q} := \mathcal{M}/\mathcal{G}$ is defined under this equivalence, treating each equivalence class as one element. It hence reflects the intrinsic variability of the system. There is a natural *projection* mapping connecting the two spaces: $\pi : \mathbf{x} \in \mathcal{M} \mapsto \{g \cdot \mathbf{x} \mid g \in \mathcal{G}\} \in \mathcal{Q}$. Under certain conditions, the quotient space is also a smooth manifold.

## 3 METHODS

As the quotient space represents the "essential states" of a system with symmetry, a principled diffusion model for the system is expected to be built on it. We unroll the development by first deriving the projected diffusion process onto the quotient space, then lifting it back into the total space (*i.e.*, the original space $\mathcal{M}$) for convenient implementation. See Fig. 2 for illustration of involved concepts and the scheme of the development. We then derive the specification in the $\mathbb{R}^{3N}/\mathrm{SE}(3)$ case for molecular structure generation, followed by training and sampling algorithms. Finally, we highlight the merit of the quotient-space diffusion in reducing training difficulty and sampler validity with a comparative analysis with existing symmetry treatments.

### 3.1 DIFFUSION PROCESS ON A GENERAL QUOTIENT SPACE

When viewed in the total space $\mathcal{M}$, due to the intrinsic symmetry, the distribution should assign the same probability to elements in each equivalence class, formally regarded as a $\mathcal{G}$-invariant distribution. To guarantee this, a common approach is to start from a simple $\mathcal{G}$-invariant distribution $p_{\mathrm{prior}}$ and let it undergo a $\mathcal{G}$-equivariant generation process, which translates to the $\mathcal{G}$-equivariance of the vector field $\mathbf{f}_t$ (the drift term) for a diffusion process (the Wiener process $\mathbf{w}_t$ defined under the Riemannian metric induces a $\mathcal{G}$-invariant transition distribution when $\mathcal{G}$ acts on the Riemannian manifold $\mathcal{M}$ isometrically). This guarantees that each intermediate distribution along the generation process is also $\mathcal{G}$-invariant (Köhler et al., 2020; Hoogeboom et al., 2022a). As $\mathcal{G}$-invariant distributions vary only *across* equivalence classes, they correspond to distributions on the quotient space

$\mathcal{Q} := \mathcal{M}/\mathcal{G}$, and the generation process can also be exactly represented on the quotient space, as explicitly given by the following theorem.

**Theorem 1.** *Assume $\{\mathbf{x}_t\}_{t \in [0,T]}$ is a diffusion process on $\mathcal{M}$, specified by the following SDE:*

$$\mathrm{d}\mathbf{x}_t = \mathbf{f}_t(\mathbf{x}_t)\,\mathrm{d}t + \sigma_t\,\mathrm{d}\mathbf{w}_t, \quad \mathbf{x}_0 \sim p_{\mathrm{prior}}, \tag{6}$$

*where $\mathbf{f}_t$ is a $\mathcal{G}$-equivariant vector field on $\mathcal{M}$, and $p_{\mathrm{prior}}$ is $\mathcal{G}$-invariant. Then the projected process $\{\mathbf{y}_t := \pi(\mathbf{x}_t)\}_{t \in [0,T]}$ onto the quotient space $\mathcal{Q} := \mathcal{M}/\mathcal{G}$ is specified by the following SDE:*

$$\mathrm{d}\mathbf{y}_t = \left( (\pi_* \mathbf{f}_t)(\mathbf{y}_t) - \frac{\sigma_t^2}{2}\mathbf{h}(\mathbf{y}_t) \right)\mathrm{d}t + \sigma_t\,\mathrm{d}\boldsymbol{\omega}_t, \quad \mathbf{y}_0 \sim \pi_{\#}p_{\mathrm{prior}}, \tag{7}$$

*where: **(1)** $\pi_* \mathbf{f}_t$ is the pushed-forward vector field onto $\mathcal{Q}$ induced by $\pi$, which is well-defined due to the $\mathcal{G}$-equivariance of $\mathbf{f}_t$; **(2)** $\mathbf{h}(\mathbf{y}_t)$ is the mean curvature vector field induced on $\mathcal{Q}$; **(3)** $\boldsymbol{\omega}_t$ is the Wiener process on $\mathcal{Q}$; and **(4)** $\pi_{\#}p_{\mathrm{prior}}$ is the pushed-forward distribution of $p_{\mathrm{prior}}$; i.e., its samples follow $\mathbf{y}_0 = \pi(\mathbf{x}_0)$ where $\mathbf{x}_0 \sim p_{\mathrm{prior}}$.*

See Appx. D.1 for formal definitions and proof. Thm. 1 shows that the projected process on the quotient space is indeed a diffusion process, and running the original process Eq. (6) followed by projection $\pi$ gives the same process by running Eq. (7) on the quotient space. The quotient-space process Eq. (7) is expressed using the projected vector field $\pi_* \mathbf{f}_t$, the Wiener process $\boldsymbol{\omega}_t$ on $\mathcal{Q}$, and perhaps unexpectedly, an additional vector field $\mathbf{h}$ reflecting the curvature of $\mathcal{Q}$. As the quotient space squeezes different equivalence classes all into points, a process viewed on the quotient space should accommodate for the change of volume of the equivalence class along the movement. This additional vector reflects the change rate of the volume of the equivalence class (see also Appx. D.3).

Although the diffusion process on the quotient space is well defined, it is not convenient to simulate it in the quotient space directly, due to the difficulty in representing this quite abstract space, which usually cannot be represented by Euclidean vectors. We hence work towards deriving a diffusion process in the total space $\mathcal{M}$, which can be simulated in the same way as the original diffusion process, while inheriting key features of a quotient-space diffusion process.

A key feature from Thm. 1 is that if using $\mathbf{f}' = \mathbf{v} + \mathbf{f}$ where $\mathbf{v} \in \mathrm{Ker}\,\pi_{*\mathbf{x}} := \{\mathbf{v} \in T_{\mathbf{x}}\mathcal{M} \mid \pi_{*\mathbf{x}}(\mathbf{v}) = \mathbf{0}\}$ at each $\mathbf{x} \in \mathcal{M}$, then the corresponding SDE in Eq. (6) induces the same quotient-space diffusion Eq. (7). Indeed, vector components in this subspace are not really necessary, as it only induces a movement *within* the equivalence class $\pi(\mathbf{x})$. Under the view of the illustration in Fig. 2(Right), this subspace is called the *vertical space* $\mathcal{V}_{\mathbf{x}} := \mathrm{Ker}\,\pi_{*\mathbf{x}}$. Correspondingly, its orthogonal completion in $T_{\mathbf{x}}\mathcal{M}$ under the Riemannian metric, referred to as the *horizontal space* $\mathcal{H}_{\mathbf{x}} := \mathcal{V}_{\mathbf{x}}^{\perp}$, identifies vectors that leads to essential movements. Since they form $T_{\mathbf{x}}\mathcal{M}$ by direct sum $T_{\mathbf{x}}\mathcal{M} = \mathcal{V}_{\mathbf{x}} \oplus \mathcal{H}_{\mathbf{x}}$, any tangent vector $\mathbf{v} \in T_{\mathbf{x}}\mathcal{M}$ admits a unique decomposition: $\mathbf{v} = \mathbf{v}^{\mathcal{V}} + \mathbf{v}^{\mathcal{H}}$, where the horizontal component $\mathbf{v}^{\mathcal{H}}$ identifies the essential component. We denote this correspondence as the horizontal projection $P_{\mathbf{x}}(\mathbf{v}) := \mathbf{v}^{\mathcal{H}}$ in $T_{\mathbf{x}}\mathcal{M}$.

With this construction, there is a unique correspondence from a quotient-space tangent vector $\mathbf{u} \in T_{\mathbf{y}}\mathcal{Q}$ to a total-space horizontal tangent vector $\tilde{\mathbf{u}} \in \mathcal{H}_{\mathbf{x}}$ at any $\mathbf{x} \in \pi^{-1}(\mathbf{y})$ in the equivalence class, called the *horizontal lift* of $\mathbf{u}$ (Def. C.2.2). It forms a $\mathcal{G}$-equivariant vector field in $\mathcal{M}$, and if $\mathbf{u} = \pi_{*\mathbf{x}}(\mathbf{v})$, its lift $\widetilde{\pi_{*\mathbf{x}}(\mathbf{v})}$ is given by $P_{\mathbf{x}}(\mathbf{v})$. Since it is always horizontal, it preserves the gist of the quotient-space vector $\mathbf{u}$ of an essential update direction, we can leverage this notion to pull the desired quotient-space diffusion Eq. (7) back to the total space $\mathcal{M}$, which admits a straightforward representation and simulation.

**Theorem 2.** *The horizontal lift of Eq. (7) has the following explicit expression:*

$$\mathrm{d}\tilde{\mathbf{x}}_t = \left( P_{\tilde{\mathbf{x}}_t}(\mathbf{f}_t(\tilde{\mathbf{x}}_t)) - \frac{\sigma_t^2}{2}\tilde{\mathbf{h}}(\tilde{\mathbf{x}}_t) \right)\mathrm{d}t + \sigma_t\,\mathrm{d}\tilde{\mathbf{w}}_t, \quad \tilde{\mathbf{x}}_0 \sim p_{\mathrm{prior}}, \tag{8}$$

*where $P_{\mathbf{x}}(\mathbf{v}) := \mathbf{v}^{\mathcal{H}}$ is the horizontal projection in the tangent space of $\mathcal{M}$, and $\tilde{\mathbf{h}}$ is the horizontal lift of the mean curvature vector $\mathbf{h}$, and $\tilde{\mathbf{w}}_t$ is the horizontal lift of the Wiener process of $\mathcal{Q}$.*

See Appx. D.2 for proof. Note that this construction does not require $\mathcal{Q}$ to be embedded in $\mathcal{M}$, which is generally not possible. Comparing Eq. (6) and Eq. (8), the lifted process is not simply given by projecting the vector field and the Wiener process in Eq. (6), but the additional vector field $\tilde{\mathbf{h}}$ persists as the reflection of the spatial change rate of equivalence-class volume to compensate the disability to change the spread of mass within an equivalence class. Importantly, Eq. (8) only has horizontal movements, which focuses on movement across equivalence classes only. As a result, the length

of sampling paths is reduced. This is one of the key advantages over existing equivariant diffusion models, which still traverse within each equivalence class following a certain rule. Moreover, Thm. 2 indicates that Eq. (8) produces the same target distribution, guaranteeing the correctness of sampling. A representative illustration is given in Fig. 1. These merits are formally stated as follows.

**Corollary 3** (Proof in Appx. D.2). *(1) The resulting random variable $\tilde{\mathbf{x}}_1$ by the lifted diffusion process Eq. (8) and the resulting $\mathbf{x}_1$ by the original diffusion process Eq. (6) follow the same distribution in the total space: $p_{\tilde{\mathbf{x}}_1} = p_{\mathbf{x}_1} = p_{\text{target}}$. (2) When $\sigma_t \equiv 0$ and starting from the same $\mathbf{x}_0 \in \mathcal{M}$, Eq. (8) leads to a shorter trajectory than Eq. (6) does.*

## 3.2 Special Case: the Shape Space

The general abstract constructions can be specified and give instructions to the motivating and representative case of molecular structure generation, where $\mathcal{M}$ is the space of concatenated vectors of 3-dimensional coordinates of $N$ points, and $\mathcal{G}$ is the special Euclidean group SE(3) composed of the 3-dimensional translation group $\mathbb{T}(3)$ and rotation group SO(3), representing *rigid-body movements*. An element of $\mathcal{M}$ is structured as $\mathbf{x} := [\vec{x}^{(1)}, \cdots, \vec{x}^{(N)}]$ (compactly, $[\vec{x}^{(n)}]_n$), where each $\vec{x}^{(n)} \in \mathbb{R}^3$, and the SE(3) group acts on $\mathbf{x}$ by translating and rotating each $\vec{x}^{(n)}$.

Since $\mathbb{T}(3)$ is not compact, there does not exist a translational invariant distribution. We (as well as many others (Yim et al., 2023b; Lin et al., 2024a)) hence represent the quotient space with respect to $\mathbb{T}(3)$ by the center-of-mass(CoM)-free subspace $\mathbb{R}^{3N}_{\text{CoM}} := \{\mathbf{x} \in \mathbb{R}^{3N} \mid \frac{1}{N} \sum_{n=1}^{N} \vec{x}^{(n)} = \vec{0}\}$, and consider the SO(3) action on its "purified" version $\mathbb{R}^{3N}_{\text{CoM}\circ}$ (removing negligible degenerate cases; see Appx. C.3). The resulting quotient space $\mathcal{Q} := \mathbb{R}^{3N}_{\text{CoM}\circ}/\text{SO}(3)$, as the concrete construction for $\mathbb{R}^{3N}/\text{SE}(3)$, is a smooth manifold. Each element in $\mathcal{Q}$ represents $N$-point configurations that are equivalent under rigid-body movements; therefore, $\mathcal{Q}$ is regarded as the "*shape space*" reflecting essentially different $N$-point configurations that differ by deformations. As the manifold $\mathbb{R}^{3N}_{\text{CoM}\circ}$ can be embedded in a Euclidean space, and structures of SO(3) are well-established (Appx. C.3), we can derive the explicit expressions for the lifted quotient-space diffusion process (Eq. (8)):

**Theorem 4.** *Assume $\mathbf{x}_t$ is a diffusion process in a manifold $\mathcal{M}$ that can be embedded into a Euclidean space, specified by the SDE: $\mathrm{d}\mathbf{x}_t = \mathbf{f}_t(\mathbf{x}_t) \, \mathrm{d}t + \sigma_t \, \mathrm{d}\mathbf{w}_t$, where $\mathbf{f}_t(\mathbf{x}_t)$ is $\mathcal{G}$-equivariant and $\mathbf{x}_0 \sim p_{\text{prior}}$ is $\mathcal{G}$-invariant. Then the lifted quotient-space diffusion process on $\mathcal{Q} := \mathcal{M}/\mathcal{G}$ onto the total space $\mathcal{M}$ is given by:*

$$\mathrm{d}\tilde{\mathbf{x}}_t = \left( P_{\tilde{\mathbf{x}}_t}(\mathbf{f}_t(\tilde{\mathbf{x}}_t)) - \frac{\sigma_t^2}{2}\tilde{\mathbf{h}}(\tilde{\mathbf{x}}_t) \right)\mathrm{d}t + \sigma_t P_{\tilde{\mathbf{x}}_t} \, \mathrm{d}\mathbf{w}_t, \quad \tilde{\mathbf{x}}_0 \sim p_{\text{prior}}, \tag{9}$$

*where $P_{\tilde{\mathbf{x}}_t} \, \mathrm{d}\mathbf{w}_t$ can be simulated by projecting the infinitesimal Gaussian sample using $P_{\tilde{\mathbf{x}}_t}$ in each step. Particularly, for the shape space $\mathcal{Q} := \mathbb{R}^{3N}/\text{SE}(3) = \mathbb{R}^{3N}_{\text{CoM}\circ}/\text{SO}(3)$, the horizontal projection $P$ and the $\tilde{\mathbf{h}}$ vector field have the following explicit expressions: for any $\mathbf{x} = [\vec{x}^{(n)}]_n \in \mathbb{R}^{3N}_{\text{CoM}\circ}$, which is CoM-free, and any $\mathbf{v} = [\vec{v}^{(n)}]_n \in T_{\mathbf{x}}\mathbb{R}^{3N}_{\text{CoM}\circ}$, which is momentum-free,*

$$P_{\mathbf{x}}(\mathbf{v}) = \left[ \vec{v}^{(n)} - \mathbf{K}(\mathbf{x})^{-1}\left( \sum\nolimits_{n'=1}^{N} \vec{x}^{(n')} \times \vec{v}^{(n')} \right) \times \vec{x}^{(n)} \right]_n, \text{ and} \tag{10}$$

$$\tilde{\mathbf{h}}(\mathbf{x}) = \left[ \mathbf{K}(\mathbf{x})^{-1}\vec{x}^{(n)} - \mathrm{tr}(\mathbf{K}(\mathbf{x})^{-1})\vec{x}^{(n)} \right]_n, \text{ where } \mathbf{K}(\mathbf{x}) := \sum_{n=1}^{N} \|\vec{x}^{(n)}\|^2 \mathbf{I} - \sum_{n=1}^{N} \vec{x}^{(n)}\vec{x}^{(n)\top} \in \mathbb{R}^{3\times 3},$$

*and "$\times$" denotes the cross product in $\mathbb{R}^3$.*

See Appx. D.3 for proof. On $\mathbb{R}^{3N}_{\text{CoM}\circ}$ with SO(3) symmetry, a vertical vector is induced from an infinitesimal SO(3) action, leading to a total angular momentum describing a rigid-body rotation without deformation. Correspondingly, a horizontal vector leads to a movement with zero total angular momentum. As a projection onto the horizontal space, $P_{\mathbf{x}}(\mathbf{v})$ in Eq. (10) essentially removes the total angular momentum of $\mathbf{v}$, leaving only a deformation without rigid-body rotation; see the end of Appx. D.3 for more physical interpretations. This is analogous to the conventional treatment for $\mathbb{T}(3)$ symmetry by removing the total (linear) momentum of $\mathbf{v}$. Together, the lifted process only deforms the point cloud (molecule) without any rigid-body movement, exactly corresponding to a movement on the shape space $\mathbf{Q}$. With the correction of the additional $\tilde{\mathbf{h}}$ term (which is horizontal by definition), the process in Eq. (9) produces the same target distribution $p_{\text{target}}$ (by Cor. 3).

### 3.3 Training and Sampling

With the general recipe for constructing a quotient-space diffusion process from a conventional diffusion process in the same space, we can now develop training and sampling methods for a quotient-space diffusion model. We consider the Euclidean total space and a Gaussian prior $p_{\text{prior}} = \mathcal{N}(\mathbf{0}, \mathbf{I})$. A more general case is shown in Appx. E.

We start from the general diffusion process Eq. (3) of conventional diffusion model to generate $p_{\text{target}} = p_1$ from $p_{\text{prior}} = p_0$. With a Gaussian prior and noting Eq. (4), the effective drift term $\mathbf{f}_t(\mathbf{x}_t) = \mathbf{v}(\mathbf{x}_t, t) + \eta_t \mathbf{s}(\mathbf{x}_t, t)$ can be expressed using the denoising model $\mathbf{D}_\theta(\mathbf{x}_t, t)$ under an affine transformation. The quotient-space diffusion is then given by Eq. (9).

**Training objective.** With the help of the horizontal projection $P_{\mathbf{x}}(\mathbf{v})$, Eq. (9) indicates that only the projected component of $\mathbf{f}_t$ matters, and $\mathbf{f}_t$ is allowed to produce an arbitrary output on the vertical space. Since $\mathbf{f}_t$ and the $\mathbf{D}_\theta(\cdot, t)$ model differ only in an affine transformation and $P_{\mathbf{x}}$ is linear, the same conclusions apply to $\mathbf{D}_\theta(\cdot, t)$. Therefore, we can modify the original training objective in Eq. (5) to only optimize the projected component:

$$\mathcal{L}(\theta) := \mathbb{E}_{p(t)} w(t) \mathbb{E}_{p(\mathbf{x}_1, \mathbf{x}_t)} \| P_{\mathbf{x}_t} (\mathbf{D}_\theta(\mathbf{x}_t, t) - \mathbf{x}_1) \|^2, \tag{11}$$

where all the time-step weights are subsumed into $w(t)$. We can see that $\mathbf{D}_\theta + \mathbf{v}^{\mathcal{V}}$ has the same loss value with $\mathbf{D}_\theta$, where $\mathbf{v}^{\mathcal{V}}$ is an arbitrary vertical vector. So the model does not need to learn *anything* on the vertical space, corresponding to movement within each equivalence class. This effect fully leverages the symmetry in *reducing learning difficulty*, as does AF3 alignment, hence could converge faster than using GeoDiff alignment in training, meanwhile AF3 alignment does not have a compatible sampler with the modified learning target (Table 1). This is empirically observed as shown in Fig. G.3.1(Left) in Appx. G.3.

**ODE sampler.** The original ODE sampler in Eq. (2) corresponds to the case with $\sigma_t \equiv 0$. By Eq. (9), the corresponding quotient-space diffusion process in the same total space is given by: $\frac{d\mathbf{x}_t}{dt} = P_{\mathbf{x}_t}(\mathbf{v}_\theta(\mathbf{x}_t, t))$, where $\mathbf{v}_\theta(\mathbf{x}_t, t)$ is given by Eq. (4).

**SDE sampler.** Comparing the original SDE in Eq. (3) and Eq. (9), the lifted quotient-space SDE is:

$$d\mathbf{x}_t = P_{\mathbf{x}_t}\big(\mathbf{v}_\theta(\mathbf{x}_t, t) + \eta_t \mathbf{s}_\theta(\mathbf{x}_t, t)\big) dt + \eta_t \tilde{\mathbf{h}}(\mathbf{x}_t) dt + \sqrt{2\eta_t} P_{\mathbf{x}_t} d\mathbf{w}_t, \tag{12}$$

where $\mathbf{s}_\theta(\mathbf{x}_t, t) = -\frac{\mathbf{x}_t - \beta_t \mathbf{D}_\theta(\mathbf{x}_t, t)}{\hat{\alpha}_t^2}$ from Sec. 2.1. Details are summarized in Alg. 1 and 3 in Appx. E.

### 3.4 Comparative Analysis on Existing Treatments for Symmetry

We now make a detailed analysis on existing methods that handle symmetry, and verify the conclusions in Table 1. In contrast to our quotient-space diffusion, we find that they either have not fully leveraged the symmetry to reduce learning difficulty, or do not have a proper sampler.

**Conventional equivariant diffusion models and data augmentation.** The common treatment of equivariant diffusion model guarantees the invariance of $p(\mathbf{x}_1)$ (Sec. 3.1, Para. 1). This can be implemented by either augmenting data samples by applying randomly chosen group actions, mimicking sampling from the invariant distribution, or using an invariant prior distribution and an equivariant architecture securing the equivariance of the model. Training remains the same as Eq. (5), and the standard samplers by Eqs. (2, 3) remain valid. For each value of $\mathbf{x}_t$, this objective amounts to let the model minimize $\|\mathbf{D}_\theta(\mathbf{x}_t, t) - \mathbf{x}_1\|^2$ averaged over $\mathbf{x}_1 \sim p(\mathbf{x}_1|\mathbf{x}_t)$, so the optimal solution is the conditional expectation $\mathbb{E}[\mathbf{x}_1|\mathbf{x}_t]$.

Fig. 3 shows an example for generating the structure of a diatomic molecule, where the target distribution $p(\mathbf{x}_1)$ concentrates on a single structure $\mathbf{x}^\star$ up to a uniform random orientation (a). For a given $\mathbf{x}_t$, samples of $p(\mathbf{x}_1|\mathbf{x}_t)$ are $\mathbf{x}^\star$ structures with orientations distributed around the orientation of $\mathbf{x}_t$ (b). Indeed, an $\mathbf{x}_1$ sample more closely oriented with $\mathbf{x}_t$ would have a higher probability to produce the given $\mathbf{x}_t$, so there is a specific orientation correspondence between the learning target $\mathbb{E}[\mathbf{x}_1|\mathbf{x}_t]$ and $\mathbf{x}_t$. So the model is still asked to learn a correspondence in the equivalent degrees of freedom (DoFs) (*i.e.*, orientations), in contrast to the quotient-space case in Eq. (11) where the model output is unconstrained in the vertical space (*i.e.*, total angular momentum). Moreover, the $\mathbf{x}_1$ samples are not all in the orientation of $\mathbf{x}_t$ because $\mathbf{x}^\star$ in other orientations can also generate this $\mathbf{x}_t$, so the model learns the orientation correspondence from samples with a variance, leading to another aspect of learning difficulty.

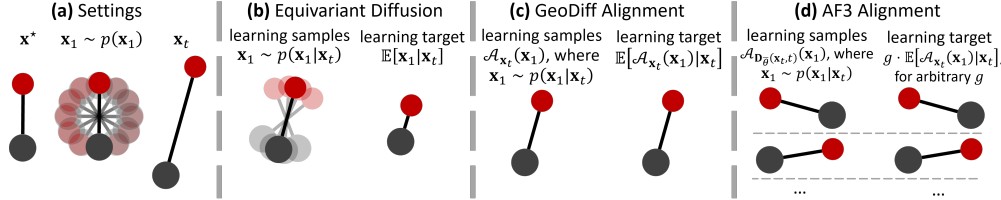

Figure 3: Illustration of denoising-model learning target using conventional equivariant diffusion, GeoDiff alignment, and AlphaFold 3 (AF3) alignment. **(a)** The example considers the structure distribution $p(\mathbf{x}_1)$ of a diatomic molecule, which concentrates on a single structure $\mathbf{x}^\star$ up to a uniform random orientation. A $\mathbf{x}_t$ sample is produced by scaling and adding noise to a $\mathbf{x}_1$ sample. **(b)** For the given $\mathbf{x}_t$, equivariant diffusion training asks the model to match $\mathbf{x}_1$ samples that distribute with a variance in orientation (the equivalent DoFs). The learning target $\mathbb{E}[\mathbf{x}_1|\mathbf{x}_t]$ has an orientation correspondence to $\mathbf{x}_t$, and is not in the same shape as $\mathbf{x}^\star$ (the bond is shorter). **(c)** Using GeoDiff alignment, all learning samples coincide with $\mathbf{x}^\star$ in the orientation of $\mathbf{x}_t$ (no variance in equivalent DoFs), so is the learning target $\mathbb{E}[\mathcal{A}_{\mathbf{x}_t}(\mathbf{x}_1)|\mathbf{x}_t]$ (has orientation correspondence), which is different from $\mathbb{E}[\mathbf{x}_1|\mathbf{x}_t]$ hence mismatches conventional diffusion samplers. **(d)** Using AlphaFold 3 (AF3) alignment, all learning samples coincide with $\mathbf{x}^\star$ in the arbitrary orientation of model output (no variance in equivalent DoFs), so is the learning target, which hence does not ask the model to learn an orientation correspondence to $\mathbf{x}_t$, but also mismatches conventional diffusion samplers.

**GeoDiff alignment.** To reduce the learning difficulty, some heuristic treatments are proposed based on alignment. A representative treatment in GeoDiff (Xu et al., 2022) modifies the training loss into: $\mathbb{E}_{p(\mathbf{x}_1,\mathbf{x}_t)}\|\mathbf{D}_\theta(\mathbf{x}_t,t) - \mathcal{A}_{\mathbf{x}_t}(\mathbf{x}_1)\|^2$, where the *alignment operation* is defined as:

$$\mathcal{A}_\mathbf{y}(\mathbf{x}) := \operatorname{argmin}_{\mathbf{x}'\in\{g\cdot\mathbf{x}|g\in\mathcal{G}\}}\|\mathbf{x}' - \mathbf{y}\|^2. \tag{13}$$

The learning task asks $\mathbf{D}_\theta(\mathbf{x}_t,t)$ to fit $\mathcal{A}_{\mathbf{x}_t}(\mathbf{x}_1)$ samples where $\mathbf{x}_1 \sim p(\mathbf{x}_1|\mathbf{x}_t)$, so the learning target becomes $\mathbb{E}[\mathcal{A}_{\mathbf{x}_t}(\mathbf{x}_1)|\mathbf{x}_t]$. As illustrated by Fig. 3(c), all the $\mathcal{A}_{\mathbf{x}_t}(\mathbf{x}_1)$ samples coincide with the $\mathbf{x}^\star$ structure in the orientation of $\mathbf{x}_t$. So the model learns $\mathbb{E}[\mathcal{A}_{\mathbf{x}_t}(\mathbf{x}_1)|\mathbf{x}_t]$ from samples with no variance in the equivalent DoFs (*i.e.*, orientation), reducing certain learning difficulty. Nevertheless, this target still requires the model to learn a specific orientation correspondence.

A caveat of this treatment is that using the conventional diffusion samplers is no longer valid, as they require a $\mathbb{E}[\mathbf{x}_1|\mathbf{x}_t]$ model, which is different (not just in orientation) from $\mathbb{E}[\mathcal{A}_{\mathbf{x}_t}(\mathbf{x}_1)|\mathbf{x}_t]$ since $SO(3)$ is not a linear space. Fig. 3(b,c) illustrates this difference: $\mathbb{E}[\mathbf{x}_1|\mathbf{x}_t]$ averages diversely oriented $\mathbf{x}^\star$ structures, resulting in a different shape than $\mathbf{x}^\star$ (the bond is shorter), while $\mathbb{E}[\mathcal{A}_{\mathbf{x}_t}(\mathbf{x}_1)|\mathbf{x}_t]$ has the same shape as $\mathbf{x}^\star$ (and is oriented following $\mathbf{x}_t$).

**AF3 alignment.** Alphafold 3 (AF3) (Abramson et al., 2024) introduces another alignment treatment by aligning the $\mathbf{x}_1$ samples towards the model output: $\mathbb{E}\|\mathbf{D}_\theta(\mathbf{x}_t,t) - \mathcal{A}_{\mathbf{D}_{\bar\theta}(\mathbf{x}_t,t)}(\mathbf{x}_1)\|^2$, where $\bar\theta$ is treated constant in optimization. This loss function allows the model output to vary by an arbitrary group action (*e.g.*, rotation) (Appx. A), hence removes the need to learn a specific target in the equivalent DoFs. Up to this DoF, the learning target is the same as that of GeoDiff $\mathbb{E}[\mathcal{A}_{\mathbf{x}_t}(\mathbf{x}_1)|\mathbf{x}_t]$, since all the $\mathbf{x}_1$ samples are averaged after aligned with the same reference.

Nevertheless, in the sampling process, the arbitrariness in the equivalent DoFs (*e.g.*, orientation) of the model output relative to $\mathbf{x}_t$ leads to an essential arbitrariness in the vector field $\mathbf{v}_\theta(\mathbf{x}_t,t)$ through Eq. (4); therefore, there is no guarantee of recovering the target distribution using conventional diffusion samplers. This problem is also noted by Boltz-1 (Wohlwend et al., 2025), which proposes to align the prediction $\mathbf{D}_\theta(\mathbf{x}_t,t)$ towards $\mathbf{x}_t$ in the sampling process. As the AF3 target is the same as that of GeoDiff up to an arbitrary rotation, this amounts to using the GeoDiff model for sampling, which still cannot guarantee producing the target distribution as discussed above.

In contrast, our *quotient-space diffusion* in Eq. (9) guarantees a valid sampling process through a solid deduction (Cor. 3), while the horizontal projection $P_\mathbf{x}$ allows the model to have an arbitrary total angular momentum hence reduces learning difficulty. Table 1 summarizes the conclusions.

## 4 EXPERIMENTS

To show the empirical advantages of our quotient-space diffusion model in real-world applications, we consider the molecular structure generation and protein structure generation tasks, both of which models $\mathbb{R}^{3N}$ distributions with $SE(3)$ symmetry. See Appx. F for experimental details.

Table 2: The effect of the quotient-space diffusion framework for molecular structure generation on the GEOM-QM9 and the GEOM-DRUGS datasets using the ET-Flow(SO(3)) and ET-Flow(O(3)) architectures. Best results are in **bold**. Best results using the same architecture are underlined.

| Datasets | Methods | Recall | | | | Precision | | | |
|---|---|---|---|---|---|---|---|---|---|
| | | Coverage (%) ↑ | | AMR (Å) ↓ | | Coverage (%) ↑ | | AMR (Å) ↓ | |
| | | mean | median | mean | median | mean | median | mean | median |
| GEOM-QM9 (0.5 Å RMSD for coverage) | CGCF | 69.47 | 96.15 | 0.425 | 0.374 | 38.20 | 33.33 | 0.711 | 0.695 |
| | GeoDiff | 76.50 | **100.00** | 0.297 | 0.229 | 50.00 | 33.50 | 1.524 | 0.510 |
| | GeoMol | 91.50 | **100.00** | 0.225 | 0.193 | 87.60 | **100.00** | 0.270 | 0.241 |
| | Torsional Diff. | 92.80 | **100.00** | 0.178 | 0.147 | 92.70 | **100.00** | 0.221 | 0.195 |
| | MCF | 95.0 | **100.00** | 0.103 | 0.044 | 93.7 | **100.00** | 0.119 | 0.055 |
| | ET-Flow(SO(3)) | 95.98 | **100.00** | 0.076 | 0.030 | 92.10 | **100.00** | 0.110 | 0.047 |
| | + GeoDiff alignment | 95.71 | **100.00** | 0.085 | 0.040 | **95.20** | **100.00** | 0.098 | 0.050 |
| | + AF3 alignment | 92.67 | **100.00** | 0.131 | 0.070 | 84.38 | **100.00** | 0.205 | 0.146 |
| | **+ Quotient-space diffusion** | **96.40** | **100.00** | **0.069** | **0.024** | 93.30 | **100.00** | **0.096** | **0.036** |
| GEOM-DRUGS (0.75 Å RMSD for coverage) | GeoDiff | 42.10 | 37.80 | 0.835 | 0.809 | 24.90 | 14.50 | 1.136 | 1.090 |
| | GeoMol | 44.60 | 41.40 | 0.875 | 0.834 | 43.00 | 36.40 | 0.928 | 0.841 |
| | Torsional Diff. | 72.70 | 80.00 | 0.582 | 0.565 | 55.20 | 56.90 | 0.778 | 0.729 |
| | MCF - S (13M) | 79.4 | 87.5 | 0.512 | 0.492 | 57.4 | 57.6 | 0.761 | 0.715 |
| | MCF - B (62M) | 84.0 | 91.5 | 0.427 | 0.402 | 64.0 | 66.2 | 0.667 | 0.605 |
| | MCF - L (242M) | **84.7** | **92.2** | **0.390** | **0.247** | 66.8 | 71.3 | 0.618 | 0.530 |
| | ET-Flow(O(3)) (8.3M) | 79.53 | 84.57 | 0.452 | 0.419 | **74.38** | **81.04** | **0.541** | **0.470** |
| | + reproduction | 78.94 | 84.24 | 0.489 | 0.472 | 66.24 | 70.42 | 0.651 | 0.595 |
| | **+ Quotient-space diffusion** | 79.86 | 85.71 | 0.459 | 0.433 | 72.70 | 79.63 | 0.565 | 0.501 |
| | ET-Flow(SO(3)) (9.1M) | 78.18 | 83.33 | 0.480 | 0.459 | 67.27 | 71.15 | 0.637 | 0.567 |
| | + reproduction | 74.91 | 80.90 | 0.541 | 0.515 | 60.33 | 62.71 | 0.724 | 0.665 |
| | + GeoDiff alignment | 75.11 | 80.74 | 0.545 | 0.526 | 59.58 | 60.48 | 0.734 | 0.678 |
| | + AF3 alignment | 71.66 | 76.09 | 0.572 | 0.570 | 52.21 | 50.00 | 0.828 | 0.793 |
| | **+ Quotient-space diffusion** | 78.50 | 84.20 | 0.477 | 0.455 | 67.35 | 71.42 | 0.635 | 0.563 |

## 4.1 MOLECULAR STRUCTURE GENERATION

**Datasets.** Molecular structure generation requires generating the 3D structure of a molecule given its molecular graph. We consider the GEOM-QM9 and GEOM-DRUGS datasets (Axelrod & Gomez-Bombarelli, 2022), which provide a structure ensemble by metadynamics in CREST (Pracht et al., 2024) for each molecule. We follow the same dataset treatments as Hassan et al. (2024).

**Settings.** We follow the settings of Hassan et al. (2024), adopting the equivariant graph Transformer architecture ET-Flow(SO(3)) with Gaussian prior distribution on GEOM-QM9, and the ET-Flow(O(3)) and ET-Flow(SO(3)) architectures with the harmonic prior on GEOM-DRUGS (Volk et al., 2023). Following existing convention, we report results in metrics based on RMSD, including Coverage (the ratio of test samples lying within a certain RMSD from a reference sample) and Average (over test samples) Minimum (over reference samples) RMSD (AMR), where "recall/precision" takes test samples from dataset/generated and reference samples from generated/dataset.

**Results.** As shown in Table 2, our quotient-space diffusion framework consistently outperforms prior methods and alignment-based treatments in terms of generation quality,[1] due to its merits to reduce learning difficulty by leveraging the symmetry while guaranteeing a correct sampler. On both datasets, our framework significantly improves over the vanilla ET-Flow and outperforms alignment-based methods, and even surpasses strong baselines such as MCF (Wang et al., 2023) on GEOM-QM9 and achieves competitive precision results with the much larger MCF-L (242M) model (Wang et al., 2023) on GEOM-DRUGS. The alignment-based methods often even degrade the performance, due to the incompatibility between their new learning targets and the samplers. Appx. G.3 verifies the faster training by quotient-space diffusion, and the advantages under other sampling settings.

## 4.2 PROTEIN STRUCTURE GENERATION

**Settings.** To demonstrate the advantage of our quotient-space diffusion framework for larger and more relevant molecules, we evaluate it on the protein structure generation task and compare it with the state-of-the-art Proteína model (Geffner et al., 2025). As a basis for protein design, the task is to learn the aggregated, unconditional (not even on an amino acid sequence) distribution of backbone structures (*i.e.*, coordinates of amino-acid residues) of all valid proteins. For the quotient-space and

---

[1]We reproduce the results using the released configurations: https://github.com/shenoynikhil/ETFlow. Due to changes in the data processing pipeline, our reproduced results do not exactly match those reported in the original paper.

Table 3: The effect of the quotient-space diffusion framework for protein structure generation using the Proteína model. Best results are in **bold**. Best results under the same settings are underlined.

| Settings | Methods | Designability (%) ↑ | FPSD ↓ vs. | | fS ↑ in | fJSD ↓ vs. | |
|---|---|---|---|---|---|---|---|
| | | | PDB | AFDB | C/A/T | PDB | AFDB |
| Representative References | FrameDiff | 65.4 | 194.2 | 258.1 | 2.46/5.78/23.35 | 1.04 | 1.42 |
| | FoldFlow (base) | 96.6 | 601.5 | 566.2 | 1.06/1.79/9.72 | 3.18 | 3.10 |
| | FoldFlow (stoc.) | 97.0 | 543.6 | 520.4 | 1.21/2.09/11.59 | 3.69 | 2.71 |
| | FoldFlow (OT) | 97.2 | 431.4 | 414.1 | 1.35/3.10/13.62 | 2.90 | 2.32 |
| | FrameFlow | 88.6 | 129.9 | 159.9 | 2.52/5.88/27.00 | 0.68 | 0.91 |
| | ESM3 | 22.0 | 933.9 | 855.4 | 3.19/6.71/17.73 | 1.53 | 0.98 |
| | Chroma | 74.8 | 189.0 | 184.1 | 2.34/4.95/18.15 | 1.00 | 1.08 |
| | RFDiffusion | 94.4 | 253.7 | 252.4 | 2.25/5.06/19.83 | 1.21 | 1.13 |
| | Proteus | 94.2 | 225.7 | 226.2 | 2.26/5.46/16.22 | 1.41 | 1.37 |
| | Genie2 | 95.2 | 350.0 | 313.8 | 1.55/3.66/11.65 | 2.21 | 1.70 |
| SDE Sampling | Proteína $\mathcal{M}_{\mathrm{FS}}^{\mathrm{small}}$, $\gamma = 0.35$ | 96.0 | 386.5 | 378.2 | 1.77/4.97/17.78 | 2.17 | 1.73 |
| | **+ Quotient-space diffusion** | **97.6** | 274.7 | 277.1 | 2.24/6.69/20.99 | 1.68 | 1.55 |
| | Proteína $\mathcal{M}_{\mathrm{FS}}^{\mathrm{small}}$, $\gamma = 0.45$ | 92.2 | 332.9 | 320.4 | 1.83/5.01/20.22 | 1.93 | 1.49 |
| | **+ Quotient-space diffusion** | 92.6 | 244.5 | 246.3 | 2.24/6.68/23.47 | 1.43 | 1.28 |
| | Proteína $\mathcal{M}_{\mathrm{FS}}^{\mathrm{small}}$, $\gamma = 0.50$ | 89.2 | 306.2 | 290.8 | 1.86/4.92/21.15 | 1.81 | 1.36 |
| | **+ Quotient-space diffusion** | 90.2 | 228.0 | 228.7 | 2.25/6.59/25.24 | 1.32 | 1.17 |
| ODE Sampling | Proteína $\mathcal{M}_{\mathrm{FS}}$ | 19.6 | 85.4 | 21.4 | 2.51/5.65/27.35 | 0.59 | **0.09** |
| | Proteína $\mathcal{M}_{\mathrm{FS}}^{\mathrm{small}}$ | 13.8 | 83.2 | 21.9 | 2.45/5.63/31.76 | 0.58 | 0.12 |
| | + AF3 alignment | 3.8 | 229.0 | 82.4 | 2.18/4.30/14.28 | 1.35 | 0.36 |
| | **+ Quotient-space diffusion** | 15.6 | **69.9** | **17.6** | **2.57/6.40/32.14** | **0.41** | 0.11 |

alignment-based models, we select the most efficient Proteína architecture version $\mathcal{M}_{\mathrm{FS}}^{\mathrm{small}}$, a 60M-parameter Transformer, and train it from scratch on the FoldSeek AFDB clusters dataset $\mathcal{D}_{\mathrm{FS}}$. For evaluation, all of our trained models and the officially released Proteína checkpoints generate samples using 400 steps with self-conditioning. We adopt both the structure-level metric of designability, and distributional metrics of FPSD, fS and fJSD measuring the mismatch between generated and reference structure distributions (Geffner et al. (2025); Appx. F.2.5).[2] Following the original work, we evaluate the models under a range of designability-diversity trade-offs controlled by the noise scale $\gamma$ (global scale of $\eta_t$ in Eq. (12)) in SDE sampling, as well as using ODE sampling.

**Results.** As shown in Table 3, our quotient-space diffusion models outperform the vanilla Proteína model under *all* the settings and metrics, highlighting the practical advantages in handling the more complicated, higher-dimensional distribution of biomolecule structures. In contrast, the AF3 alignment treatment even degrades the performance significantly in this distributional setting, due to the fundamental incompatibility between its learning target and the sampler. As a practical benefit of reducing learning difficulty, the quotient-space diffusion framework even allows the 60M-parameter model $\mathcal{M}_{\mathrm{FS}}^{\mathrm{small}}$ to surpass the performance of the much larger 200M-parameter model $\mathcal{M}_{\mathrm{FS}}$ on most metrics. This provides a compelling evidence that the quotient-space diffusion model, ensuring both sampling fidelity and learning efficiency, is the key to advancing generative protein models.

## 5 CONCLUSION

In this work, we formally construct a framework for building diffusion models on the quotient space over a group, as a principled approach to handling symmetry for generative tasks. This is done by first revealing the essential generation process by projecting the conventional equivariant diffusion process onto the quotient space, then pulling it back to the original space so that practical implementation is as easy as the original process. The resulting process updates samples without any intra-equivalence-class movement by an explicit projection operator, making simulation easier, and with the derived additional vector field, the process guarantees producing the correct target distribution. Moreover, the projection allows the model to be arbitrary in the subspace of intra-equivalence-class movement, hence reduces learning difficulty by leveraging the symmetry. On this point, we formalize the desire that has been pursued by remarkable prior works including GeoDiff and AlphaFold, whose heuristic alignment-based treatments would distort the generated distribution. On the representative molecular structure generation tasks on $\mathbb{R}^{3N}$ with $\mathrm{SE}(3)$ symmetry, our quotient-space diffusion framework improves performance universally over conventional diffusion models as well as the alignment-based treatments, demonstrating the practical value of this principled approach for handling and leveraging symmetry.

---

[2]Due to a known bug in a previous version of FoldSeek (Daras et al., 2025, Appx. B), we only focus on these metrics in the main text. See Table F.2.2 for results in more comprehensive metrics.

ACKNOWLEDGMENTS

This work is supported by Zhongguancun Academy (Grant No. C20250506). DH is supported by National Science Foundation of China (NSFC62376007), National Science Foundation of China (under Key Project No. 92570203), Beijing Natural Science Foundation (Z250001) and Beijing Major Science and Technology Project under Contract no. Z251100008425004.

## 6 ETHICS STATEMENT

This work adheres to the ICLR Code of Ethics. Our study does not involve human subjects, personal data, or sensitive demographic information. All experiments are conducted on publicly available benchmark datasets, which are widely used in the machine learning community. No new data collection or human/animal experimentation was performed.

## 7 REPRODUCIBILITY STATEMENT

To facilitate the reproducibility of our research, we provide comprehensive details throughout the paper and its supplementary materials. We begin by establishing the necessary foundational knowledge in Sec. 2.1 and Appx. B. For all theoretical claims and proofs presented in the main text, we offer detailed step-by-step derivations in Appx. D. Our experiments are thoroughly documented; the datasets, training procedures, and evaluation protocols are carefully described in Sec. 4 and Appx. F. Upon acceptance of this paper, we commit to making our full codebase and all model checkpoints publicly available to ensure that the community can fully reproduce our results.

## 8 THE USE OF LARGE LANGUAGE MODELS (LLMS)

In the preparation of this manuscript, LLMs were employed as a writing assistant to refine the language and improve the grammar. Furthermore, we utilized LLMs to assist in verifying our mathematical formulas for notational consistency. Following this process, all textual and mathematical content was meticulously reviewed, revised, and validated by the authors, who assume full responsibility for the final work presented.

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

## APPENDIX

The organization of the appendix are as follows. In Appx. A, we briefly discuss the related work relevant to our research. In Appx. B, we review some background knowledge of Riemannian geometry and stochastic calculus on the manifold. In Appx. C, we give the details of the Riemannian structures of the quotient space. In Appx. D, we give all the proofs of the theorems in the main text. In Appx. E, we extend our methods to more general cases. Appx. F details experimental settings, and finally Appx. G provides additional experimental results and methodological discussions.

## A    RELATED WORK

**Diffusion models on Riemannian manifolds.**    As the quotient has the Riemannian manifold structure, several previous works construct the diffusion model on the Riemannian manifolds. De Bortoli et al. (2022) constructs diffusion models using different overlapping local coordinate systems of the manifold and requires geodesic random walk to simulate the forward process. Huang et al. (2022); Chen & Lipman (2023) construct diffusion models in an embedding space which allows a global representation but requires explicit geodesic formula of the manifold. Zhu et al. (2024) constructs the reverse of kinetic Langevin dynamics on a Lie group to perform generative modeling. Such an approach is not designed for and not readily applicable to the quotient space, which has a different geometric structure from the Lie group. In our quotient space case, the specialty with a quotient structure enables us to construct diffusion models using the coordinate systems of the total space without relying on an embedding of the quotient in the total space (unnecessarily an embedding space), which is more practical to implement yet still general.

**Geometric diffusion models.**    To ensure physical symmetry in the generation process, a mainstream strategy integrates fundamental physical constraints, such as SE(3) equivariance, directly into the diffusion-model architecture. This approach, pioneered by models like EDM (Hoogeboom et al., 2022a), typically employs an EGNN to operate directly on atomic coordinates, using techniques like zero center of mass adjustments to guarantee translational invariance. This foundational concept was subsequently extended in several directions. For instance, the approach was adapted for Diffusion Bridges in models like EDM-Bridge (Wu et al., 2022) and for diffusion in a latent space in models like GeoLDM (Xu et al., 2023). These equivariant diffusion techniques have been successfully applied across a range of molecular tasks. For structure generation, models like GeoDiff (Xu et al., 2022) predict 3D structures from molecular graphs. In molecular optimization, methods such as DiffHopp (Torge et al., 2023) refine existing molecules to enhance desired properties. For de novo design, a key advancement has been to combine discrete diffusion models (D3PM) (Austin et al., 2021) for 2D topology with continuous equivariant diffusion for 3D geometry, enabling joint generation as seen in models like DiffSBDD (Schneuing et al., 2024) and MUDiff (Hua et al., 2024). A similar problem has also been considered in crystalline structure generation, where the intrinsic periodic translation invariance is an intrinsic symmetry. Lin et al. (2024a) highlighted the intrinsic periodic translation symmetry that has been omitted for a long time in the field of periodic crystalline structure generation. The work designed a modified diffusion process that induces a transition kernel that is invariant under periodic translation, leading to a learning target for the score model that is invariant under periodic translation. Cornet et al. (2025) proposes a novel method that generalizes the Trivialized Diffusion Model framework for fractional coordinates to model the intrinsic periodic translation symmetry using flat coordinates. The proposed method considers the process with the velocity restricted to the CoM-free linear subspace. They have achieved the removal of variance on equivalent DoFs, but still asks the neural network model to learn to predict a specific target in the equivalent DoFs.

**Learning with alignment**    To reduce learning difficulty, some heuristic treatments (learning with alignment) have been proposed in hope to reduce the DoFs corresponding to the symmetry group action. The alignment strategy used in GeoDiff (Xu et al., 2022) aligns the target structure to the noisy input by finding an optimal rigid transformation that minimizes the distance between them.

Another approach, proposed in AlphaFold 3 (AF3) (Abramson et al., 2024), aligns the target samples to the model output structure: $\mathbb{E}\left\|\mathbf{D}_\theta(\mathbf{x}_t, t) - \mathcal{A}_{\mathbf{D}_{\bar\theta}(\mathbf{x}_t,t)}(\mathbf{x}_1)\right\|^2$, where $\bar\theta$ is treated constant in optimization. This loss function allows the model output to differ by an arbitrary group action (*e.g.*, rotation). Indeed, for an arbitrary group action $g_{\mathbf{x}_t,t}$, a new denoising model $g_{\mathbf{x}_t,t} \cdot \mathbf{D}_\theta(\mathbf{x}_t, t)$

achieves the same loss since $\|g_{\mathbf{x}_t,t} \cdot \mathbf{D}_\theta(\mathbf{x}_t, t) - \mathcal{A}_{g_{\mathbf{x}_t,t} \cdot \mathbf{D}_{\bar{\theta}}(\mathbf{x}_t,t)}(\mathbf{x}_1)\|^2 = \|g_{\mathbf{x}_t,t} \cdot \mathbf{D}_\theta(\mathbf{x}_t, t) - g_{\mathbf{x}_t,t} \cdot \mathcal{A}_{\mathbf{D}_{\bar{\theta}}(\mathbf{x}_t,t)}(\mathbf{x}_1)\|^2 = \|\mathbf{D}_\theta(\mathbf{x}_t, t) - \mathcal{A}_{\mathbf{D}_{\bar{\theta}}(\mathbf{x}_t,t)}(\mathbf{x}_1)\|^2$, where the last equality holds since the group preserves metric. As discussed in the main text, these two alignment-based training frameworks lack a definite guarantee for recovering the correct target distribution, and is incompatible with the sampling process.

Boltz-1 (Wohlwend et al., 2025), an open-source replication of AF3, noticed this issue and proposed a modification in sampling to align the denoised structure to the structure in the current generation step before updating. Nevertheless, as discussed in Sec. 3.4, this, together with the training protocol of AF3, amounts to the operation of GeoDiff, still questioning the sampling process.

# B  BACKGROUND IN RIEMANNIAN GEOMETRY AND STOCHASTIC CALCULUS

## B.1  RIEMANNIAN GEOMETRY

In this section, we review some background on differential geometry and Riemannian geometry. For a systematic treatment of the subject, please refer to standard textbooks Lee (2003; 2018).

First, we give the formal definition of the smooth manifold. A manifold is a general topological space that locally has a Euclidean structure.

**Definition B.1.1.** An $M$-**dimensional topological manifold** is a topological space $\mathcal{M}$ such that:

- $\mathcal{M}$ is locally Euclidean, *i.e.* locally homeomorphic to $\mathbb{R}^M$. Formally, $\forall x \in \mathcal{M}$,[3] there exists an open neighborhood $x \in \mathcal{U} \subset \mathcal{M}$ that is homeomorphic to some open set $\mathcal{V} \subset \mathcal{M}$. We call the homeomorphism $\phi : \mathcal{U} \to \mathcal{V} \subset \mathbb{R}^M$ a **coordinate system** or a chart.

- $\mathcal{M}$ is a Hausdorff topological space.

- $\mathcal{M}$ has a countable basis for its topology.

A smooth manifold is a topological manifold with an additional smooth structure, which is defined as follows.

**Definition B.1.2.** A smooth structure on a $M$-dimensional topological space $\mathcal{M}$ is a collection of coordinate systems $\mathscr{C} = \{(\mathcal{U}^{(\alpha)}, \phi^{(\alpha)}) : \alpha \in \mathcal{A}\}$ which satisfies the following properties:

- The collection $\mathscr{C}$ covers $\mathcal{M}$: $\bigcup_{\alpha \in \mathcal{A}} \mathcal{U}^{(\alpha)} = \mathcal{M}$;

- For any $\alpha, \beta \in \mathcal{A}$, the transition function $\phi^{(\alpha)} \circ \phi^{(\beta)^{-1}}$ is a smooth map;

- $\mathscr{C}$ is a maximal collection, *i.e.* if $(\mathcal{U}, \phi)$ is a coordinate system such that for all $\alpha \in \mathcal{A}$ that the maps $\phi \circ \phi^{(\alpha)^{-1}}$ and $\phi^{(\alpha)} \circ \phi^{-1}$ are smooth, then $(\mathcal{U}, \phi) \in \mathcal{C}$.

The pair $(\mathcal{M}, \mathscr{C})$ is called a **smooth manifold** of dimension $M$.

If there is a coordinate system $(\mathcal{U}, \phi)$ around a point $x \in \mathcal{M}$, then in this neighborhood of $x$, the manifold admits a coordinate chart $x^i(x) := \phi^i(x)$ and a manifold point in the neighborhood can be expressed as a vector $\mathbf{x}(x) = (x^1(x), \cdots, x^M(x))^\top$.

With the smooth structure, we can define a smooth function on the manifold and a smooth mapping between smooth manifolds.

**Definition B.1.3.** Let $\mathcal{M}, \mathcal{N}$ be smooth manifolds with dimensions $M, N$ respectively.

- A function $f : \mathcal{M} \to \mathbb{R}$ is called a **smooth function** if its vectorized form $f \circ \phi^{-1} : \phi^{-1}(\mathcal{U}) \to \mathbb{R}$ is smooth on $\phi^{-1}(\mathcal{U}) \subset \mathbb{R}^M$ for all smooth coordinate systems $(\mathcal{U}, \phi)$ of $\mathcal{M}$. Denote all the smooth functions on $\mathcal{M}$ as $C^\infty(\mathcal{M})$.

- A map $F : \mathcal{M} \to \mathcal{N}$ is called a **smooth map** if its vectorized form $\psi \circ F \circ \phi^{-1} : \phi^{-1}(\mathcal{U}) \to \psi(\mathcal{V})$ is smooth for all smooth coordinate systems $(\mathcal{U}, \phi)$ of $\mathcal{M}$ and $(\mathcal{V}, \psi)$.

---

[3]On an abstract manifold, a point is an abstract object and may not be a vector by itself, so we do not use a boldface notation. A vector representation as the coordinates is available after choosing a (local) coordinate system.

A smooth map $F : \mathcal{M} \to \mathcal{N}$ which is invertible and whose inverse is smooth is called a diffeomorphism. In this case we say that $\mathcal{M}$ and $\mathcal{N}$ are diffeomorphic manifolds.

To define movement on a smooth manifold $\mathcal{M}$, we need to define tangent vectors on the manifold.

**Definition B.1.4.** Let $\mathcal{M}$ be a smooth manifold, and $x \in \mathcal{M}$ is a point, and $\mathcal{U}$ is a neighborhood of it. A linear map $\mathbf{v} : C^\infty(\mathcal{U}) \to \mathbb{R}$ is called a derivative at $x$ if it satisfies

$$\mathbf{v}(fg) = f(x)\mathbf{v}(g) + g(x)\mathbf{v}(f), \quad \forall f, g \in C^\infty(\mathcal{U}).$$

The set of all the derivatives of $C^\infty(\mathcal{U})$ in $x$, denoted by $T_x\mathcal{M}$, is a vector space called the **tangent space** to $\mathcal{M}$ at $x$. An element of $T_x\mathcal{M}$ is called a **tangent vector** at $x$.

**Definition B.1.5.** Let $\mathcal{M}, \mathcal{N}$ be smooth manifolds and $F : \mathcal{M} \to \mathcal{N}$ be a smooth map. Let $x \in \mathcal{M}$ and $\mathcal{V} \subseteq \mathcal{N}$ be a neighborhood of $F(x)$. Then $F$ induces a **push-forward map** over the tangent spaces, $F_{*x} : T_x\mathcal{M} \to T_{F(x)}\mathcal{N}$, is defined as:

$$F_{*x}(\mathbf{v})(f) := \mathbf{v}(f \circ F), \quad \forall f \in C^\infty(\mathcal{V}), \mathbf{v} \in T_x\mathcal{M}.$$

When a coordinate system $(\mathcal{U}, \phi)$ around $x$ and $(\mathcal{V}, \psi)$ around $F(x)$ are chosen, the coordinate expression for $F_{*x}$ is just the Jacobian matrix of its vectorized form $\psi \circ F \circ \phi^{-1}$, *i.e.*, $\nabla(\psi \circ F \circ \phi^{-1})(x)$. So $F_*$ is also called the differential of $F$ and also admits the notation $\mathrm{d}F$.

The **tangent bundle** $T\mathcal{M}$ of a smooth manifold $\mathcal{M}$ is the union of the tangent spaces of each points, *i.e.* $T\mathcal{M} := \bigsqcup_{x \in \mathcal{M}} T_x\mathcal{M}$. Similar to the total derivative of the smooth map in Euclidean space, the differential of a smooth map between smooth manifolds is a linear map between tangent spaces.

A **vector field** $\mathbf{v}$ on a smooth manifold $\mathcal{M}$ is a correspondence that associates to each point $x \in \mathcal{M}$ a vector $\mathbf{v}_x \in T_x\mathcal{M}$. The vector field is smooth if the mapping $\mathbf{v} : \mathcal{M} \to T\mathcal{M}$ is smooth. Denote all the smooth vector fields on $\mathcal{M}$ by $\mathscr{X}(\mathcal{M})$. With the definition of a vector field, we can define the solution of ordinary differential equation (ODE) on the manifold. The idea is similar to the definition in Euclidean space, the solution of the ODE is a curve whose velocity at each point is the same as the vector field.

**Definition B.1.6.** Let $\mathbf{v}$ be a smooth vector field on the smooth manifold $\mathcal{M}$. An **integral curve of** $\mathbf{v}$ is a differentiable curve $\gamma : [0, T] \to \mathcal{M}$ whose velocity at each point is equal to the value of $\mathbf{v}$ at that point:

$$\dot{\gamma}_t = \mathbf{v}_{\gamma_t}, \quad \forall t \in [0, T].$$

**Definition B.1.7.** A **1-form** $\Theta$ on smooth manifold $\mathcal{M}$ is a correspondence that associates to each point $x \in \mathcal{M}$ a covector $\Theta_x \in T_x^*\mathcal{M}$. The 1-form is smooth if the mapping $\Theta : \mathcal{M} \to T^*\mathcal{M}$ is smooth. The **cotangent space** $T_x^*\mathcal{M}$ of $\mathcal{M}$ at $x$ is defined as the dual space of $T_x\mathcal{M}$. The **cotangent bundle** $T^*\mathcal{M}$ is the union of the cotangent space of each points, *i.e.*, $T^*\mathcal{M} := \bigsqcup_{p \in \mathcal{M}} T_x^*\mathcal{M}$.

A smooth manifold only has a topological structure. If we want to define concepts like the "length of the velocity" and distance between two points on the manifold, a metric on the tangent space is required. Such a metric endows the metric with an additional geometry structure. The formal definitions are as follows.

**Definition B.1.8.** A **Riemannian metric** on a smooth manifold is a correspondence which associates to each point $p$ of $\mathcal{M}$ an inner product $\langle \cdot, \cdot \rangle_x$ that varies smoothly on $\mathcal{M}$. In other words, for any two smooth vector fields $\mathbf{u}, \mathbf{v}$, $\langle \mathbf{u}, \mathbf{v} \rangle$ is a smooth function on $\mathcal{M}$. A smooth manifold with a given Riemannian metric is called a **Riemannian manifold**.

To define the "difference" between tangent space at different points, we need to introduce a concept called affine connection.

**Definition B.1.9.** An **affine connection** $\nabla$ on a Riemannian manifold is a mapping

$$\nabla : \mathscr{X}(\mathcal{M}) \times \mathscr{X}(\mathcal{M}) \to \mathscr{X}(\mathcal{M})$$

which is denoted by $(\mathbf{u}, \mathbf{v}) \to \nabla_{\mathbf{u}}\mathbf{v}$ which satisfies the following properties:

- $\nabla_{\mathbf{u}}\mathbf{v}$ is linear over $C^\infty(\mathcal{M})$ in $\mathbf{u}$: $\forall f^{(1)}, f^{(2)} \in C^\infty(\mathcal{M})$ and $\mathbf{u}^{(1)}, \mathbf{u}^{(2)} \in \mathscr{X}(\mathcal{M})$,

$$\nabla_{f^{(1)}\mathbf{u}^{(1)}+f^{(2)}\mathbf{u}^{(2)}}\mathbf{v} = f^{(1)}\nabla_{\mathbf{u}^{(1)}}\mathbf{v} + f^{(2)}\nabla_{\mathbf{u}^{(2)}}\mathbf{v};$$

- $\nabla_{\mathbf{u}}\mathbf{v}$ is linear over $\mathbb{R}$ in $\mathbf{v}$: $\forall a^{(1)}, a^{(2)} \in \mathbb{R}$ and $\mathbf{v}^{(1)}, \mathbf{v}^{(2)} \in \mathscr{X}(\mathcal{M})$,

$$\nabla_{\mathbf{u}^{(1)}}(a^{(1)}\mathbf{v}^{(1)} + a^{(2)}\mathbf{v}^{(2)}) = a^{(1)}\nabla_{\mathbf{u}}\mathbf{v}^{(1)} + a^{(2)}\nabla_{\mathbf{u}}\mathbf{v}^{(2)};$$

- $\nabla$ satisfies the following product rule: $\forall f \in C^{\infty}(\mathcal{M})$,

$$\nabla_{\mathbf{u}}(f\mathbf{v}) = f\nabla_{\mathbf{u}}\mathbf{v} + (\mathbf{u}f)\mathbf{v}.$$

A connection is called the **Levi-Civita connection** if satisfies the following additional properties:

- $\nabla$ is compatible with metric: $\nabla_{\mathbf{u}}\langle\mathbf{v}^{(1)}, \mathbf{v}^{(2)}\rangle = \langle\nabla_{\mathbf{u}}\mathbf{v}^{(1)}, \mathbf{v}^{(2)}\rangle + \langle\mathbf{v}^{(1)}, \nabla_{\mathbf{u}}\mathbf{v}^{(2)}\rangle;$

- $\nabla$ is torsion-free: $\nabla_{\mathbf{u}}\mathbf{v} - \nabla_{\mathbf{v}}\mathbf{u} = \mathbf{u}(\mathbf{v}(\cdot)) - \mathbf{v}(\mathbf{u}(\cdot)).$

The Levi-Civita connection is the connection with nice properties. Its existence and uniqueness is a fundamental result of Riemannian geometry.

**Theorem B.1.10.** *(Fundamental Theorem of Riemannian Geometry (Lee, 2018, Thm. 5.10)) Assume* $(\mathcal{M}, \langle\cdot, \cdot\rangle)$ *is a Riemannian manifold. Then there exists a unique Levi-Civita connection.*

As the end of this subsection, we introduce the Laplace-Beltrami operator on the manifold, which is used to define the Wiener process on the manifold.

**Definition B.1.11.** Let $\nabla$ be the Levi-Civita connection on the Riemannian manifold $(\mathcal{M}, \langle\cdot, \cdot\rangle)$. The **Hessian** of $f \in C^{\infty}(\mathcal{M})$ is then defined by:

$$\mathrm{Hess}(f)(\mathbf{u}, \mathbf{v}) := \mathbf{v}(\mathbf{u}(f)) - (\nabla_{\mathbf{v}}\mathbf{u})(f), \quad \forall \mathbf{u}, \mathbf{v} \in \mathscr{X}(\mathcal{M}).$$

The **Laplace-Beltrami operator** $\Delta$ is defined as the trace of Hessian. In other words, $\Delta f := \sum_{i=1}^{M} \mathrm{Hess}(\mathbf{e}_i, \mathbf{e}_i)$ where $\{\mathbf{e}_1, ..., \mathbf{e}_M\}$ is an orthonormal basis for $T_x\mathcal{M}$.

## B.2 STOCHASTIC CALCULUS ON A MANIFOLD

With the Riemannian structure defined in the previous section, we can consider the definition of stochastic differential equations (SDE) and diffusion processes on the manifold. For a systematic treatment of the subject, please refer to standard textbooks Hsu (2002); Thalmaier (2023). First, we recall the definition of SDE and diffusion process in Euclidean space.

**Definition B.2.1.** The **infinitesimal generator** of a stochastic process $(\mathbf{x}_t)_t$ for a function $\phi(\mathbf{x})$ is

$$\mathcal{L}_t\phi(\mathbf{x}) = \lim_{s\to 0^+} \frac{\mathbb{E}[\phi(\mathbf{x}_{t+s})|\mathbf{x}_t = \mathbf{x}] - \phi(\mathbf{x})}{s},$$

where $\phi$ is a suitably regular function. For an Itô process defined as the solution to the SDE $\mathrm{d}\mathbf{x}_t = \mathbf{f}(\mathbf{x}_t, t)\,\mathrm{d}t + \mathbf{\Sigma}(\mathbf{x}_t, t)\,\mathrm{d}\mathbf{w}_t$, the generator is

$$\mathcal{L}_t = \sum_{i=1}^{D} \mathbf{f}^i(\mathbf{x}, t)\partial_i + \frac{1}{2}\sum_{i,j=1}^{D} \left(\mathbf{\Sigma}(\mathbf{x}, t)\mathbf{\Sigma}(\mathbf{x}, t)^{\top}\right)^{ij}\partial_i\partial_j.$$

On the other hand, the diffusion process can also be defined by its generator.

**Definition B.2.2.** A $D$-dimensional stochastic process $\mathbf{x}_t$ with continuous sample path defined on a probability space $(\Omega, \mathscr{F}, \mathbb{P})$ is called a diffusion process generated by a smooth second-order elliptic operator $\mathcal{L}_t$ if the following hold: $\forall f \in C^{\infty}(\mathbb{R}^D)$, the process

$$M_t^f = f(\mathbf{x}_t) - f(\mathbf{x}_0) - \int_0^t \mathcal{L}_s f(\mathbf{x}_s)\,\mathrm{d}s$$

is a $\mathscr{F}_t$-martingale.

To generalize the definition of SDE to a Riemannian manifold $\mathcal{M}$, we need to define the second-order differential operator on the manifold. Let $\mathcal{M}$ be an $M$-dimensional Riemannian manifold. A second-order partial differential operator on $\mathcal{M}$ is of the form

$$\mathcal{L} = \mathbf{v}^{(0)} + \sum_{k=1}^{R} \mathbf{v}^{(k)^2}, \quad \text{where } \mathbf{v}^{(k)} \in \mathscr{X}(\mathcal{M}),$$

for some $R \in \mathbb{N}^+$. The square of a vector field is understood as the decomposition of derivatives:

$$\mathbf{v}^{(k)^2}(f) := \mathbf{v}^{(k)}(\mathbf{v}^{(k)}(f)), \quad \forall f \in C^\infty(\mathcal{M}).$$

The vector fields can be generalized to the time-dependent case. Now we can extend the definition of a diffusion process on a Riemannian manifold.

**Definition B.2.3.** (Thalmaier, 2023, Def. 1.1.3) Let $(\Omega, \mathscr{F}, \mathbb{P}; (\mathscr{F})_{t\geqslant 0})$ be a probability space equipped with increasing sequence of sub-$\sigma$-algebra $\mathscr{F}_t \subseteq \mathscr{F}$. An adapted continuous process $\mathbf{x}_t$ taking values in $\mathcal{M}$, is called $\mathcal{L}_t$-diffusion if for all test functions $f \in C_c^\infty(\mathcal{M})$, the process

$$N_t^f := f(\mathbf{x}_t) - f(\mathbf{x}_0) - \int_0^t (\mathcal{L}_s f)(\mathbf{x}_s)\, \mathrm{d}s, \quad t \geqslant 0,$$

is a martingale, *i.e.* $\mathbb{E}[N_t^f - N_s^f \mid \mathscr{F}_s] = 0, \quad \forall s \leqslant t.$

For a special case, we can define the Wiener process on the Riemannian manifold $\mathcal{M}$.

**Definition B.2.4.** The Wiener process $\mathbf{w}_t$ on $\mathcal{M}$ is a diffusion process with generator $\frac{1}{2}\Delta$, where $\Delta$ is the Laplace-Beltrami operator on the Riemannian manifold $(\mathcal{M}, \langle \cdot, \cdot \rangle)$ (Def. B.1.11); *i.e.* $\mathbf{w}_t$ is a continuous stochastic process on $\mathcal{M}$ such that for any $f \in C^\infty(\mathcal{M})$,

$$f(\mathbf{x}_t) - \frac{1}{2}\int_0^t \Delta f(\mathbf{w}_s)\, \mathrm{d}s,$$

is a local martingale up to a valid time period.

For stochastic differential geometry, the Stratonovitch integral is more convenient than the Itô Integral. The Stratonovitch differential effectively subsumes the deterministic second-order effect of the Wiener process from the quadratic variation into the drift term, so that it satisfies the ordinary chain rule of calculus. This property enables a clear correspondence between the diffusion process under a diffeomorphism between two Riemannian manifolds. Next, we give the definition of the Stratonovitch integral on the Euclidean space and its generalization to Riemannian manifolds.

**Definition B.2.5.** For continuous real-valued semimartingales $\mathbf{x}$ and $\mathbf{y}$, let $\mathbf{x} \circ \mathrm{d}\mathbf{y} := \mathbf{x}\, \mathrm{d}\mathbf{y} + \frac{1}{2}\mathrm{d}[\mathbf{x}, \mathbf{y}]$ be the Stratonovitch differential. Here $\mathbf{x}\, \mathrm{d}\mathbf{y}$ is the usual Itô differential, and $\mathrm{d}[\mathbf{x}, \mathbf{y}] := \mathrm{d}\mathbf{x}\mathrm{d}\mathbf{y}$ is the quadratic co-variation of $\mathbf{x}$ and $\mathbf{y}$. The integral

$$\int_0^t \mathbf{x} \circ \mathrm{d}\mathbf{y} = \int_0^t \left( \mathbf{x}\mathrm{d}\mathbf{y} + \frac{1}{2}\mathrm{d}[\mathbf{x}, \mathbf{y}]_t \right)$$

is called the Stratonovitch integral of $\mathbf{x}$ with respect to $\mathbf{y}$.

**Proposition B.2.6.** *(Itô-Stratonovitch formula (Thalmaier, 2023, Prop. 1.2.10)). Let $\mathbf{x}$ be a continuous $\mathbb{R}^D$-valued semimartingale and $f \in C^\infty(\mathbb{R}^D)$. Then $\mathrm{d}f(\mathbf{x}) = \langle \nabla f(\mathbf{x}), \circ\mathrm{d}\mathbf{x} \rangle$.*

The Itô-Stratonovitch formula shows the advantage of the Stratonovich differential: it satisfies the usual chain rule of classical calculus. So at least formally, classical differential calculus can be applied in calculations involving Stratonovich differentials.

**Proposition B.2.7.** *(Thalmaier, 2023, Prop. 1.2.11) Solutions to the Stratonovitch SDE*

$$\mathrm{d}\mathbf{x}_t = \mathbf{f}(\mathbf{x}_t, t)\, \mathrm{d}t + \boldsymbol{\Sigma}(\mathbf{x}_t, t) \circ \mathrm{d}\mathbf{w}_t \tag{B.2.1}$$

*define $\mathcal{L}$-diffusions for the operator*

$$\mathcal{L} = \mathbf{v}^{(0)} + \frac{1}{2}\sum_{k=1}^D \mathbf{v}^{(k)^2}, \quad \text{where } \mathbf{v}^{(0)} = \mathbf{f},\ \mathbf{v}^{(k)} = \sum_{i=1}^D \boldsymbol{\Sigma}_k^i \partial_i.$$

From this result, we can see that Eq. (B.2.1) describes the same diffusion process as the following Itô SDE:

$$\mathrm{d}\mathbf{x}_t = \left( \mathbf{f}(\mathbf{x}_t, t) + \frac{1}{2}\sum_{k=1}^D \mathbf{v}_*^{(k)}(\mathbf{v}^{(k)}) \right) \mathrm{d}t + \boldsymbol{\Sigma}(\mathbf{x}_t, t)\, \mathrm{d}\mathbf{w}_t,$$

where $\mathbf{v}_*^{(k)}(\mathbf{v}^{(k)}) := \sum_{i,j=1}^D \mathbf{v}^{(k)^j}(\partial_j \mathbf{v}^{(k)^i})\partial_i$.

Now we can generalize the definition of SDE to the Riemannian manifold case. An SDE on manifold $\mathcal{M}$ can be defined by vector fields $\mathbf{v}^{(0)}, \mathbf{v}^{(1)}, ..., \mathbf{v}^{(M)}$ on $\mathcal{M}$. Let $\mathbf{w}$ be the $\mathbb{R}^M$-valued Wiener

process and $\mathbf{x}_0$ be an $\mathcal{M}$-valued random variable serving as the initial value of the solution. The equation is symbolically written as

$$d\mathbf{x}_t = \mathbf{v}^{(0)}(\mathbf{x}_t, t)\, dt + \sum_{k=1}^{D} \mathbf{v}^{(k)}(\mathbf{x}_t, t) \circ d\mathbf{w}_t^k. \tag{B.2.2}$$

**Definition B.2.8.** An $\mathcal{M}$-valued semimartingale $\mathbf{x}_t$ defined up to a proper stopping time $\tau$ is a solution to the SDE Eq. (B.2.2) up to $\tau$ if for all $f \in C^\infty(\mathcal{M})$,

$$f(\mathbf{x}_t) = f(\mathbf{x}_0) + \int_0^t \left( \mathbf{v}^{(0)}(f)(\mathbf{x}_s, s)\, ds + \sum_{k=1}^{D} \mathbf{v}^{(k)}(f)(\mathbf{x}_s, s) \circ d\mathbf{w}_t^k \right), \quad 0 \leqslant t < \tau.$$

**Proposition B.2.9.** *(Thalmaier, 2023, Cor. 1.2.19) Let $\mathcal{L} = \mathbf{v}^{(0)} + \frac{1}{2} \sum_{k=1}^{D} \mathbf{v}^{(k)^2}$ and $\mathbf{x}_t$ be the solution to the SDE Eq. (B.2.2). Then for all $f \in C^\infty(\mathcal{M})$,*

$$N_t^f := f(\mathbf{x}_t) - f(\mathbf{x}_0) - \int_0^t (\mathcal{L}_s f)(\mathbf{x}_s) ds, \quad t \geqslant 0,$$

*is a martingale. In other words, the solution of SDE Eq. (B.2.2) is a $\mathcal{L}$ diffusion to the operator $\mathcal{L} = \mathbf{v}^{(0)} + \frac{1}{2} \sum_{k=1}^{D} \mathbf{v}^{(k)^2}$.*

## C   LIE GROUP AND QUOTIENT SPACE

In this section, we formally describe the Lie group and rigorously construct the quotient space as a manifold. Please refer to the standard textbooks Lee (2018) for details in the systematic treatment.

### C.1   LIE GROUP AND ITS ACTION ON A MANIFOLD

With the definition of a smooth manifold, we first define the concept of Lie group as a continuous group with good properties.

**Definition C.1.1.** A **Lie group** is a smooth manifold $\mathcal{G}$ that is also a group with the property that the multiplication map $\mathcal{G} \times \mathcal{G} \to \mathcal{G}, (g, h) \mapsto g \cdot h$ and the inversion map $\mathcal{G} \to \mathcal{G}, g \mapsto g^{-1}$ are both smooth.

Define the left multiplication mapping $L_g(h) = gh$, which is introduced to differentiate $g$ as a Lie-group element and as an action on a group element. A vector field $\mathbf{v}$ on $\mathcal{G}$ is said to be left-invariant if it is invariant under all left multiplications, *i.e.* $(L_g)_{*g'}(\mathbf{v}_{g'}) = \mathbf{v}_{gg'}$.

Certain vector fields on a Lie group have an algebraic structure called the Lie algebra. We first give the general axiomatic definition.

**Definition C.1.2.** A **Lie algebra** is a vector space $\mathfrak{g}$ endowed with a map called the Lie bracket $[\cdot, \cdot] : \mathfrak{g} \times \mathfrak{g} \to \mathfrak{g}$ that satisfies the following properties for all $X, Y, Z \in \mathfrak{g}$:

- Bilinearity: $\forall a, b \in \mathbb{R}$,
$$[aX + bY, Z] = a[X, Z] + b[Y, Z], [Z, aX + bY] = a[Z, X] + b[Z, Y];$$

- Antisymmetry: $[X, Y] = -[Y, X]$;

- Jacobi Identity: $[X, [Y, Z]] + [Y, [Z, X]] + [Z, [X, Y]] = 0$.

All the smooth left-invariant vector fields on a Lie group $\mathcal{G}$ form a Lie algebra $\mathfrak{g}$, called the **Lie algebra of** $\mathcal{G}$. It has the same dimension as $\mathcal{G}$. Importantly, left-invariant vector fields are isomorphic to the tangent space at the identity, $T_e\mathcal{G}$, which is also identified as the Lie algebra $\mathfrak{g}$ of the Lie group $\mathcal{G}$.

**Example C.1.3.** The Lie algebra of the group $\mathrm{SO}(3)$, denoted by $\mathfrak{so}(3)$, is given by all the 3-dimensional antisymmetric matrices $\mathfrak{so}(3) = \{\mathbf{A} \in \mathbb{R}^{3 \times 3} \mid \mathbf{A} + \mathbf{A}^\top = \mathbf{0}\}$.

A Lie group typically represents a set of transformations on a state space $\mathcal{M}$ of interest. We formally define this as the group action. In the following, assume that the total space $\mathcal{M}$ is a Riemannian manifold and $\mathcal{G}$ is a compact Lie group.

**Definition C.1.4.** Let $\mathcal{G}$ be a group and $\mathcal{M}$ is a Riemannian manifold. A (left) **group action** of $\mathcal{G}$ on $\mathcal{M}$ is a map $\mathcal{G} \times \mathcal{M} \to \mathcal{M}$, $(g, \mathbf{x}) \mapsto g \cdot \mathbf{x}$, satisfying $g_1 \cdot (g_2 \cdot \mathbf{x}) = (g_1 g_2) \cdot \mathbf{x}$ and $e \cdot \mathbf{x} = \mathbf{x}, \forall g_1, g_2 \in \mathcal{G}, \mathbf{x} \in \mathcal{M}$. An action is smooth if its defining map $\mathcal{G} \times \mathcal{M} \to \mathcal{M}$ is smooth. We also reload the notation $L_g(\mathbf{x}) := g \cdot \mathbf{x}$ for distinguishing $g$ as an element from as a transformation on the manifold.

In the case where the Lie group acts on a Riemannian manifold, to draw meaningful conclusions, we would expect some compatibility between group action and the Riemannian metric, which is the concept of an isometric action. Moreover, to ensure the topological structure of the quotient space so as to define useful constructions on the quotient space, concepts of a free action and proper action are introduced.

**Definition C.1.5.** **(1)** A smooth action is said to be an **isometric** action if the map $L_g : \mathcal{M} \to \mathcal{M}, \mathbf{x} \mapsto g \cdot \mathbf{x}$ is an isometry for any $g \in \mathcal{G}$, *i.e.*,

$$\langle \mathbf{u}, \mathbf{v} \rangle_{\mathbf{x}} = \langle (L_g)_{*\mathbf{x}}(\mathbf{u}), (L_g)_{*\mathbf{x}}(\mathbf{v}) \rangle_{g \cdot \mathbf{x}}. \tag{C.1.1}$$

**(2)** A smooth action is said to be **free** if for any $\mathbf{x} \in \mathcal{M}$, $g \cdot \mathbf{x} = \mathbf{x}$ indicates $g = e$. **(3)** A smooth action is said to be **proper** if the map $\mathcal{G} \times \mathcal{M} \to \mathcal{M} \times \mathcal{M}, (g, \mathbf{x}) \mapsto (g \cdot \mathbf{x}, \mathbf{x})$ is a proper map, meaning that the preimage of every compact set is compact.

For the properness, there is a convenient characterization.

**Proposition C.1.6.** *(Lee, 2018, Prop. C.15) Assume $\mathcal{G}$ is a Lie group acting smoothly on the smooth manifold $\mathcal{M}$. The action is proper if and only if the following condition is satisfied: if $\{p^{(n)}\}_n$ is a sequence in $\mathcal{M}$ and $\{g^{(n)}\}_n$ is a sequence in $\mathcal{G}$ such that both $\{p^{(n)}\}_n$ and $\{g^{(n)} \cdot p^{(n)}\}_n$ converge, then a subsequence of $\{g^{(n)}\}_n$ converges. Particularly, every smooth action by a compact Lie group on a smooth manifold is proper.*

## C.2 CONSTRUCTION OF THE QUOTIENT SPACE

The group action typically represents a symmetry in the sense that points that can be transformed to each other by a group action are regarded as symmetric, *i.e.*, they are equivalent. Therefore, we can define an equivalence relation $\sim$ on $\mathcal{M}$ as $\mathbf{x} \sim \mathbf{x}'$ if $\exists g \in \mathcal{G}, \mathbf{x}' = g \cdot \mathbf{x}$. The equivalence class with representative $\mathbf{x}$ is defined as the set of all points that are equivalent to $\mathbf{x}$. The quotient space $\mathcal{Q} := \mathcal{M}/\mathcal{G}$ (as a set) is defined under this equivalence relation, which consists of equivalence classes under the relation $\sim$. The original space $\mathcal{M}$ is referred to as the total space. There is a natural mapping called projection that connects the total space and the quotient space, which maps any $\mathbf{x} \in \mathcal{M}$ to the equivalence class it represents. In this case where the equivalence is defined by a Lie group, the projection mapping can be written as:

$$\pi : \mathcal{M} \to \mathcal{Q}, \ \pi(\mathbf{x}) := \{g \cdot \mathbf{x} \mid g \in \mathcal{G}\}.$$

Due to this expression, the equivalence class in such a case is the orbit of the Lie group $\mathcal{G}$ at $\mathbf{x}$. Therefore, it can be understood that an equivalence class is a "representation" (literal meaning; not the mathematical concept) of the Lie group, hence can also adopt manifold structures of $\mathcal{G}$ under the mentioned "good" conditions. Also, the $(\mathcal{M}, \mathcal{Q}, \pi)$ structure forms a fiber bundle, in which context the equivalence class is also called a fiber at $\pi(\mathbf{x})$, and this special fiber bundle induced from a Lie group action is called a principal $\mathcal{G}$-bundle.

Moreover, under certain conditions, the quotient space inherits the Riemannian structure of the total space $\mathcal{M}$ through the projection mapping.

**Theorem C.2.1.** *(Lee, 2018, Cor. 2.29) Let $\mathcal{M}$ be a Riemannian manifold, and $\mathcal{G}$ be a Lie group acting smoothly, freely, properly, and isometrically on $\mathcal{M}$. Then the quotient space $\mathcal{Q} := \mathcal{M}/\mathcal{G}$ has a unique smooth manifold structure and Riemannian metric such that $\pi$ is a Riemannian submersion.*

We will assume the conditions, *i.e.*, $\mathcal{G}$ is a Lie group acting smoothly, freely, properly, and isometrically on $\mathcal{M}$, in the following development. Given that $\mathcal{Q}$ is a smooth manifold, the projection mapping induces a linear mapping $\pi_*$ between the tangent spaces of the two manifolds. It introduces more structures in the total space $\mathcal{M}$. In each tangent space $T_x \mathcal{M}$, we can define a subspace of it, called the **vertical space**, by the kernel of $\pi_*$:

$$\mathcal{V}_{\mathbf{x}} := \mathrm{Ker} \ \pi_{*\mathbf{x}}.$$

By this definition, tangent vectors in the vertical space can be understood that it does not move $\mathbf{x}$ in a way that alters the projection onto $\mathcal{Q}$ by $\pi$, so the movement stays within the equivalence class. The

vertical space can then be understood as the tangent space of the equivalence class. As mentioned above, in this case where the quotient space is induced from the Lie group $\mathcal{G}$, the equivalence class is a "representation" of the Lie group, hence the vertical space is a "mirror" of the tangent space of the Lie group, which is in turn isomorphic to the Lie algebra $\mathfrak{g}$ of the Lie group $\mathcal{G}$.

To complete the whole tangent space, a concept of horizontal space $\mathcal{H}_\mathbf{x}$ is expected. In general, the horizontal space $\mathcal{H}_\mathbf{x}$ is a linear subspace of $T_\mathbf{x}\mathcal{M}$ that makes up $T_\mathbf{x}\mathcal{M}$ by direct sum with $\mathcal{V}_\mathbf{x}$:

$$T_\mathbf{x}\mathcal{M} = \mathcal{V}_\mathbf{x} \oplus \mathcal{H}_\mathbf{x}.$$

Under this direct-sum construction, any tangent vector $\mathbf{v} \in T_\mathbf{x}\mathcal{M}$ can then be *uniquely* decomposed into the vertical and horizontal components, $\mathbf{v} = \mathbf{v}^\mathcal{V} + \mathbf{v}^\mathcal{H}$. Correspondingly, a vector field on $\mathcal{M}$ is called a vertical/horizontal vector field if it takes a vertical/horizontal tangent vector at every point. Every smooth vector field $\mathbf{v}$ on $\mathcal{M}$ can be expressed uniquely in the form $\mathbf{v} = \mathbf{v}^\mathcal{V} + \mathbf{v}^\mathcal{H}$, where both the vertical and horizontal vector fields are smooth (Lee, 2018, Prop. 2.25). For future reference, we assign a convenient notation to the horizontal projection within $T_\mathbf{x}\mathcal{M}$ itself:

$$P_\mathbf{x}(\mathbf{v}) := \mathbf{v}^\mathcal{H}, \quad \forall \mathbf{v} \in T_\mathbf{x}\mathcal{M}.$$

Nevertheless, the horizontal space $\mathcal{H}_\mathbf{x}$ as a subspace that makes up the tangent space $T_\mathbf{x}\mathcal{M}$ by the direct sum with $\mathcal{V}_\mathbf{x}$ is not unique. Therefore, a smooth correspondence from $\mathbf{x}$ to such an $\mathcal{H}_\mathbf{x}$ is an independent structure, referred to as the "connection" in the fiber-bundle context. In the current specific case where $\mathcal{M}$ endows a Riemannian structure, we can uniquely define the horizontal space as the orthogonal complement under the inner product in the tangent space:

$$\mathcal{H}_\mathbf{x} := \mathcal{V}_\mathbf{x}^{\perp (T_\mathbf{x}\mathcal{M}, \langle \cdot, \cdot \rangle_\mathbf{x})}, \tag{C.2.1}$$

which gives a canonical "connection".

As would be expected, in contrast to vertical tangent vectors, a horizontal tangent vector represents a movement through different equivalence classes, corresponding to a movement on the quotient space $\mathcal{Q}$. Therefore, we can construct the concept of horizontal lift which establishes a correspondence from a vector field on $\mathcal{Q}$ to a horizontal vector field on $\mathcal{M}$.

**Definition C.2.2.** Given a vector field $\mathbf{u}$ on $\mathcal{Q}$, a vector field $\tilde{\mathbf{u}}$ on $\mathcal{M}$ is called a **horizontal lift** of $\mathbf{u}$, if $\tilde{\mathbf{u}}$ is a horizontal vector field, *i.e.*, $\tilde{\mathbf{u}}_\mathbf{x} \in \mathcal{H}_\mathbf{x}$ for all $\mathbf{x} \in \mathcal{M}$, and $\tilde{\mathbf{u}}$ is $\pi$-related to $\mathbf{u}$ by $\pi_{*\mathbf{x}}(\tilde{\mathbf{u}}_\mathbf{x}) = \mathbf{u}_{\pi(\mathbf{x})}$.

**Proposition C.2.3.** *(Lee, 2018, Prop. 2.25) Given a smooth connection $\mathbf{x} \mapsto \mathcal{H}_\mathbf{x}$ and assuming $\pi : \mathcal{M} \to \mathcal{Q}$ is a smooth submersion, every smooth vector field on $\mathcal{Q}$ always has a unique smooth horizontal lift to $\mathcal{M}$.*

If the connection is induced from the Riemannian structure of $\mathcal{M}$ by Eq. (C.2.1) and if the group action is isometric, then a nice compatibility can be derived. For a quotient-space tangent vector $\mathbf{u} \in T_\mathbf{y}\mathcal{Q}$ at some $\mathbf{y} \in \mathcal{Q}$, consider two ways to construct a tangent vector at some point $\mathbf{x} \in \pi^{-1}(\mathbf{y})$ in the equivalence class. The first way is directly by the horizontal lift, which gives $\tilde{\mathbf{u}}_\mathbf{x}$, which is the unique horizontal tangent vector such that $\pi_{*\mathbf{x}}(\tilde{\mathbf{u}}_\mathbf{x}) = \mathbf{u}$. The other way is to first horizontal-lift $\mathbf{u}$ to another point $\mathbf{x}' \in \pi^{-1}(\mathbf{y})$ in the equivalence class, then push it forward to the tangent space at $\mathbf{x}$ by a transformation that maps $\mathbf{x}'$ to $\mathbf{x}$. Since both points lie in the same equivalence class and the Lie group acts on the manifold freely, there exists a unique group action $g \in \mathcal{G}$ such that $\mathbf{x} = g \cdot \mathbf{x}' = L_g(\mathbf{x}')$, so the resulting tangent vector is $(L_g)_{*\mathbf{x}'}(\tilde{\mathbf{u}}_{\mathbf{x}'})$. Noting that $(L_g)_{*\mathbf{x}'}$ preserves the metric between $T_{\mathbf{x}'}\mathcal{M}$ and $T_\mathbf{x}\mathcal{M}$ (see Eq. (C.1.1)), we know that it also preserves the horizontal spaces, *i.e.*, $(L_g)_{*\mathbf{x}'}(\mathcal{H}_{\mathbf{x}'}) = \mathcal{H}_\mathbf{x}$, so $(L_g)_{*\mathbf{x}'}(\tilde{\mathbf{u}}_{\mathbf{x}'}) \in \mathcal{H}_\mathbf{x}$. Moreover, $\pi_{*\mathbf{x}}\left((L_g)_{*\mathbf{x}'}(\tilde{\mathbf{u}}_{\mathbf{x}'})\right) = (\pi \circ L_g)_{*\mathbf{x}'}(\tilde{\mathbf{u}}_{\mathbf{x}'}) = \pi_{*\mathbf{x}'}(\tilde{\mathbf{u}}_{\mathbf{x}'}) = \mathbf{u}$ also projects to the quotient-space tangent vector $\mathbf{u}$ (noting that $\pi \circ L_g = \pi$ for any $g \in \mathcal{G}$, and recalling the definition of horizontal lift), by the uniqueness of the horizontal tangent vector that projects to $\mathbf{u}$, we have $\tilde{\mathbf{u}}_\mathbf{x} = (L_g)_{*\mathbf{x}'}(\tilde{\mathbf{u}}_{\mathbf{x}'})$, or equivalently,

$$\tilde{\mathbf{u}}_{g \cdot \mathbf{x}} = (L_g)_{*\mathbf{x}}(\tilde{\mathbf{u}}_\mathbf{x}), \quad \forall \mathbf{x} \in \pi^{-1}(\mathbf{y}), g \in \mathcal{G}. \tag{C.2.2}$$

The unique existence of the correspondence from $T_{\pi(\mathbf{x})}\mathcal{Q}$ back to $T_\mathbf{x}\mathcal{M}$ by horizontal lift (Prop. C.2.3) allows us to introduce a Riemannian structure on $\mathcal{Q}$ from that on $\mathcal{M}$. For any $\mathbf{y} \in \mathcal{Q}$ and $\mathbf{u}^{(1)}, \mathbf{u}^{(2)} \in T_\mathbf{y}\mathcal{Q}$, define:

$$\langle \mathbf{u}^{(1)}, \mathbf{u}^{(2)} \rangle_\mathbf{y}^\mathcal{Q} := \langle \tilde{\mathbf{u}}_\mathbf{x}^{(1)}, \tilde{\mathbf{u}}_\mathbf{x}^{(2)} \rangle_\mathbf{x}^\mathcal{M}, \tag{C.2.3}$$

for any $\mathbf{x} \in \pi^{-1}(\mathbf{y})$. This is well-defined since the right hand side is independent the choice of $\mathbf{x}$ due to the horizontal-lift–push-forward compatibility (Eq. (C.2.2)) and isometry (Eq. (C.1.1)):
$$\langle \tilde{\mathbf{u}}_{g\cdot\mathbf{x}}^{(1)}, \tilde{\mathbf{u}}_{g\cdot\mathbf{x}}^{(2)} \rangle_{g\cdot\mathbf{x}}^{\mathcal{M}} = \langle (L_g)_{*\mathbf{x}}(\tilde{\mathbf{u}}_{\mathbf{x}}^{(1)}), (L_g)_{*\mathbf{x}}(\tilde{\mathbf{u}}_{\mathbf{x}}^{(2)}) \rangle_{g\cdot\mathbf{x}}^{\mathcal{M}} = \langle \tilde{\mathbf{u}}_{\mathbf{x}}^{(1)}, \tilde{\mathbf{u}}_{\mathbf{x}}^{(2)} \rangle_{\mathbf{x}}^{\mathcal{M}}.$$

Subsequent constructions on the quotient space $\mathcal{Q}$ can be induced from this Riemannian structure. Due to its compatibility with the original Riemannian manifold $\mathcal{M}$, these constructions have direct connections to their counterparts on $\mathcal{M}$. Particularly, the Levi-Civita connections on $\mathcal{Q}$ and $\mathcal{M}$ follow the relation below.

**Proposition C.2.4.** *(Lee, 2018, Exercise. 5.6) Let $\nabla^{\mathcal{M}}$ and $\nabla^{\mathcal{Q}}$ denote the Levi-Civita connections on $\mathcal{M}$, $\mathcal{Q}$, respectively, where $\nabla^{\mathcal{Q}}$ is constructed from the Riemannian metric induced from that of $\mathcal{M}$ by Eq. (C.2.3). Then for any vector fields $\mathbf{u}^{(1)}, \mathbf{u}^{(2)}$ on $\mathcal{Q}$, denoting their horizontal lifts to $\mathcal{M}$ as $\tilde{\mathbf{u}}^{(1)}, \tilde{\mathbf{u}}^{(2)}$, we have:*

$$\nabla_{\tilde{\mathbf{u}}^{(1)}}^{\mathcal{M}} \tilde{\mathbf{u}}^{(2)} = \widetilde{\nabla_{\mathbf{u}^{(1)}}^{\mathcal{Q}} \mathbf{u}^{(2)}} + \frac{1}{2} \Big( \mathcal{L}_{\tilde{\mathbf{u}}^{(1)}} \tilde{\mathbf{u}}^{(2)} \Big)^{\mathcal{V}},$$

*where $\widetilde{\nabla_{\mathbf{u}^{(1)}}^{\mathcal{Q}} \mathbf{u}^{(2)}}$ denotes the horizontal lift of the vector field $\nabla_{\mathbf{u}^{(1)}}^{\mathcal{Q}} \mathbf{u}^{(2)}$ on $\mathcal{Q}$, and $\mathcal{L}_{\tilde{\mathbf{u}}^{(1)}} \tilde{\mathbf{u}}^{(2)}$ is the Lie derivative (or commutator) of vector field $\tilde{\mathbf{u}}^{(2)}$ under $\tilde{\mathbf{u}}^{(1)}$, defined as $\frac{d}{dt}\big|_{t=0} (\Phi_t)_* \tilde{\mathbf{u}}^{(2)}$ where $\Phi_t$ is the flow of vector field $\tilde{\mathbf{u}}^{(1)}$. The Lie derivative adopts an explicit expression $\mathcal{L}_{\tilde{\mathbf{u}}^{(1)}} \tilde{\mathbf{u}}^{(2)}(f) = \tilde{\mathbf{u}}^{(1)}(\tilde{\mathbf{u}}^{(2)}(f)) - \tilde{\mathbf{u}}^{(2)}(\tilde{\mathbf{u}}^{(1)}(f))$. Particularly,*

$$\widetilde{\nabla_{\mathbf{u}^{(1)}}^{\mathcal{Q}} \mathbf{u}^{(2)}} = (\nabla_{\tilde{\mathbf{u}}^{(1)}}^{\mathcal{M}} \tilde{\mathbf{u}}^{(2)})^{\mathcal{H}}.$$

## C.3 Specifications for the Shape Space $\mathbb{R}^{3N}/\mathrm{SE}(3)$

For a concrete and practically highly concerned example, we consider the shape space $\mathbb{R}^{3N}/\mathrm{SE}(3)$. In this example, each $\mathbb{R}^{3N}$ element is structured as:

$$\mathbf{x} = [\vec{x}^{(1)}, \vec{x}^{(2)}, \cdots, \vec{x}^{(N)}] \in \mathbb{R}^{3N}, \quad \text{with each } \vec{x}^{(n)} \in \mathbb{R}^3,$$

which represents the 3-dimensional coordinates of $N$ points in $\mathbb{R}^3$ (point cloud). The $\mathrm{SE}(3)$ group is composed of the 3-dimensional translation group $\mathbb{T}(3)$ and the 3-dimensional rotation group $\mathrm{SO}(3)$, altogether representing the set of rigid-body movements of the $N$ points. Since the translation group $\mathbb{T}(3)$ is not compact, there does not exist a probability distribution that is translation invariant. We (as well as many others (Yim et al., 2023b; Lin et al., 2024a)) hence represent the quotient space w.r.t this group by suppressing these equivalent DoFs by choosing a canonical translational position by anchoring the center of mass (CoM) of the point cloud at the origin, and consider the resulting CoM-free subspace $\mathbb{R}_{\mathrm{CoM}}^{3N} := \{\mathbf{x} \in \mathbb{R}^{3N} \mid \frac{1}{N} \sum_{n=1}^{N} \vec{x}^{(n)} = \vec{0}\}$ [4] and consider the $\mathrm{SO}(3)$ action on it. Since the constraint is linear, this space $\mathbb{R}_{\mathrm{CoM}}^{3N}$ is a linear subspace of $\mathbb{R}^{3N}$, and it is naturally a Riemannian manifold with the standard inner product of $\mathbb{R}^{3N}$. An element of the $\mathrm{SO}(3)$ group is given by a $3 \times 3$ rotation matrix for which we reload the notation $g$. The natural action of $g$ on $\mathbf{x}$ is defined as $g \cdot \mathbf{x} = [g\vec{x}^{(1)}, g\vec{x}^{(2)}, \cdots, g\vec{x}^{(N)}]$ ($g$ here represents the rotation matrix), *i.e.* the rotation is applied on each point of the system.

Unfortunately, $\mathrm{SO}(3)$ does not act freely (see Def. C.1.5) on $\mathbb{R}_{\mathrm{CoM}}^{3N}$ in some degenerate cases, *e.g.* all the points lie on a straight line. So we define the subset $\mathcal{D} \subset \mathbb{R}_{\mathrm{CoM}}^{3N}$ that $\mathrm{SO}(3)$ does not have free action on it; *i.e.*, for any $\mathbf{x} \in \mathcal{D}$, there exists a nontrivial action $g \neq e \in \mathrm{SO}(3)$ such that $g \cdot \mathbf{x} = \mathbf{x}$, indicating that $\mathcal{D}$ contains points that have a higher symmetry beyond $\mathrm{SO}(3)$. For a converging sequence $\{\mathbf{x}^{(n)}\}_n$ in $\mathcal{D}$ which converges to $\mathbf{x} \in \mathbb{R}_{\mathrm{CoM}}^{3N}$ as $n \to \infty$, there exists a sequence $\{g^{(n)}\}_n$ in $\mathcal{G}$ such that $g^{(n)} \cdot \mathbf{x}^{(n)} = \mathbf{x}^{(n)}$. Since the group $\mathrm{SO}(3)$ is compact and the group action is continuous, $\{g^{(n)}\}_n$ has a convergent subsequence that converges to $g \in \mathcal{G}$, which satisfies $g \cdot \mathbf{x} = \mathbf{x}$. Hence $\mathbf{x} \in \mathcal{D}$, and therefore, $\mathcal{D}$ is closed. Subsequently, $\mathbb{R}_{\mathrm{CoM}^\circ}^{3N} := \mathbb{R}_{\mathrm{CoM}}^{3N} \setminus \mathcal{D}$ is still a smooth manifold. As $\mathcal{D}$ is measure-zero in $\mathbb{R}_{\mathrm{CoM}}^{3N}$ (since any $g \in \mathrm{SO}(3)$ is non-singular, the equation $g \cdot \mathbf{x} = \mathbf{x}$ reduces degrees of freedom of $\mathbf{x}$), it is unlikely for a real simulation in $\mathbb{R}_{\mathrm{CoM}}^{3N}$ to hit the set $\mathcal{D}$, making negligible difference algorithmically.

---

[4]Here we choose a simple form of CoM by treating atoms equally weighted to avoid unnecessary notation complexity. In fact, any choice to determine one point in $\mathbb{R}^3$ from the $N$ points suffices the reduction of the translation DoFs (as long as proper permutational invariance is guaranteed).

By removing the degenerate set $\mathcal{D}$, $\mathrm{SO}(3)$ can now act freely and smoothly on $\mathbb{R}^{3N}_{\mathrm{CoM}\circ}$. Moreover, since the $\mathrm{SO}(3)$ action is isometric in the Euclidean space and $\mathbb{R}^{3N}_{\mathrm{CoM}\circ}$ inherits the same metric, $\mathrm{SO}(3)$ also acts isometrically on $\mathbb{R}^{3N}_{\mathrm{CoM}\circ}$. Since $\mathrm{SO}(3)$ is a compact group, by Prop. C.1.6, the action is also proper. Now that the action is smooth, free, proper, and isometric, by Thm. C.2.1, the quotient space $\mathcal{Q} := \mathbb{R}^{3N}_{\mathrm{CoM}\circ}/\mathrm{SO}(3)$ is a Riemannian manifold and the projection $\pi : \mathbb{R}^{3N}_{\mathrm{CoM}\circ} \to \mathcal{Q}$ is a Riemannian submersion. This $\mathcal{Q}$ is the concrete construction for the $\mathbb{R}^{3N}/\mathrm{SE}(3)$ quotient space. Since the each element in this quotient space $\mathcal{Q}$ is an equivalence class containing equivalent point-cloud configurations, we refer to this space $\mathcal{Q}$ as the "shape space".

By the projection mapping $\pi : \mathbb{R}^{3N}_{\mathrm{CoM}\circ} \to \mathcal{Q}$, the vertical space $\mathcal{V}_{\mathbf{x}} := \mathrm{Ker}\,\pi_{*\mathbf{x}}$ can already be defined, which reflects the infinitesimal movements in $\mathbb{R}^{3N}_{\mathrm{CoM}\circ}$ by group actions, which amounts to movements within the equivalence class $\pi(\mathbf{x})$ (Appx. B). Since $\mathbb{R}^{3N}_{\mathrm{CoM}\circ}$ is a Riemannian manifold (tangent space inner product is inherited from the standard Euclidean inner product), we can define the horizontal space $\mathcal{H}_{\mathbf{x}} := (\mathcal{V}_{\mathbf{x}})^{\perp_{T_{\mathbf{x}}\mathbb{R}^{3N}_{\mathrm{CoM}\circ}}}$ as the orthogonal complement of $\mathcal{V}_{\mathbf{x}}$ in $T_{\mathbf{x}}\mathbb{R}^{3N}_{\mathrm{CoM}\circ}$. Since $\mathcal{V}_{\mathbf{x}}$ and $\mathcal{H}_{\mathbf{x}}$ recover $T_{\mathbf{x}}\mathbb{R}^{3N}_{\mathrm{CoM}\circ}$ by direct sum, any tangent vector $\mathbf{v} \in T_{\mathbf{x}}\mathbb{R}^{3N}_{\mathrm{CoM}\circ}$ can thus be uniquely decomposed as the addition of a vertical component and horizontal component.

On this concrete example, the vertical and horizontal spaces can be expressed explicitly. Since the vertical space is induced from group action, which acts freely, so this space is isomorphic to the tangent space of the Lie group, *i.e.*, the Lie algebra. For $\mathcal{G} = \mathrm{SO}(3)$, the Lie algebra $\mathfrak{so}(3)$ is the set of antisymmetric $3 \times 3$ matrices. So the vertical space is given by:

$$\mathcal{V}_{\mathbf{x}} = \{[\mathbf{A}\vec{x}^{(1)},\ \mathbf{A}\vec{x}^{(2)},\ \cdots,\ \mathbf{A}\vec{x}^{(N)}] \mid \mathbf{A} \in \mathfrak{so}(3)\}.$$

Using the 3-dimensional representation $\vec{\omega} = [\vec{\omega}^1, \vec{\omega}^2, \vec{\omega}^3]^\top \in \mathbb{R}^3$ for $\mathfrak{so}(3)$, any antisymmetric $3 \times 3$ matrix can be represented as:

$$\mathbf{A} = \begin{bmatrix} 0 & -\vec{\omega}^3 & \vec{\omega}^2 \\ \vec{\omega}^3 & 0 & -\vec{\omega}^1 \\ -\vec{\omega}^2 & \vec{\omega}^1 & 0 \end{bmatrix} = \sum_{i=1}^{3} \vec{\omega}^i \mathbf{J}_i, \tag{C.3.1}$$

$$\text{where} \quad \mathbf{J}_1 := \begin{bmatrix} 0 & 0 & 0 \\ 0 & 0 & -1 \\ 0 & 1 & 0 \end{bmatrix}, \quad \mathbf{J}_2 := \begin{bmatrix} 0 & 0 & 1 \\ 0 & 0 & 0 \\ -1 & 0 & 0 \end{bmatrix}, \quad \mathbf{J}_3 := \begin{bmatrix} 0 & -1 & 0 \\ 1 & 0 & 0 \\ 0 & 0 & 0 \end{bmatrix}. \tag{C.3.2}$$

The $\vec{\omega}$ vector is the *Euler vector* (or *rotation vector*) representation of $\mathfrak{so}(3)$. Its direction represents the axis of rotation, and its length represents the rate of rotation; therefore, the $\vec{\omega}$ vector can be thought of as an *angular velocity vector*. Eq. (C.3.1) suggests that $\{\mathbf{J}_1, \mathbf{J}_2, \mathbf{J}_3\}$ forms a basis for $\mathfrak{so}(3)$. Following this representation, we have $\mathbf{A}\vec{x}^{(n)} = \vec{\omega} \times \vec{x}^{(n)}$, where "$\times$" denotes the usual cross product on $\mathbb{R}^3$, so the vertical space can also be written as:

$$\mathcal{V}_{\mathbf{x}} = \{[\vec{\omega} \times \vec{x}^{(1)},\ \vec{\omega} \times \vec{x}^{(2)},\ \cdots,\ \vec{\omega} \times \vec{x}^{(N)}] \mid \vec{\omega} \in \mathbb{R}^3\}. \tag{C.3.3}$$

This can be thought of as the (linear) velocity on each atom induced from the common angular velocity $\vec{\omega}$ as if all the atoms are on a rigid body which rotates following the angular velocity $\vec{\omega}$. As a rigid-body movement, this (linear) velocity does not deform the shape of the body, hence the "shape" of the point cloud is reserved, and the resulting configuration would still be within the equivalence class. This is the intuition behind the concept of the vertical space.

For the horizontal space, which is the orthogonal complement of the vertical space, is given by

$$\mathcal{H}_{\mathbf{x}} = \left\{ \mathbf{v} = [\vec{v}^{(1)}, \cdots, \vec{v}^{(N)}] \in \mathbb{R}^{3N} \ \Big|\ \sum_{n=1}^{N} \vec{v}^{(n)} = \vec{0},\ \sum_{n=1}^{N} \vec{x}^{(n)} \times \vec{v}^{(n)} = \vec{0} \right\}, \tag{C.3.4}$$

since the $\vec{v}^{(n)}$ vectors should keep the CoM fixed, and as the orthogonal complement, they should also satisfy $\sum_{n=1}^{N} \vec{v}^{(n)} \cdot (\vec{\omega} \times \vec{x}^{(n)}) = \vec{\omega} \cdot (\sum_{n=1}^{N} \vec{x}^{(n)} \times \vec{v}^{(n)}) = 0$ for any $\vec{\omega} \in \mathbb{R}^3$. Intuitively, the first constraint $\sum_{n=1}^{N} \vec{v}^{(n)} = \vec{0}$ means the total (linear) momentum of the system is zero, meaning that the movement keeps the CoM of the point cloud, *i.e.*, the total/rigid-body translational degree of freedom is fixed. The second constraint $\sum_{n=1}^{N} \vec{x}^{(n)} \times \vec{v}^{(n)} = \vec{0}$ means the total angular momentum of the system is zero, meaning that there is no net rotation as a whole, *i.e.*, the total/rigid-body rotational degree of freedom is fixed. Therefore, vectors in the horizontal space correspond to movements that does not move within an equivalence class, but purely across equivalence classes.

Given the construction of the horizontal space (*i.e.*, a "connection"), the horizontal lift for a quotient-space tangent vector (and vector field) can be derived. As $SO(3)$ acts smoothly, freely, properly, and isometrically on $\mathbb{R}^{3N}_{\text{CoM}^\circ}$, a Riemannian structure can be induced for $\mathcal{Q}$ from that of $\mathbb{R}^{3N}_{\text{CoM}^\circ}$, which is inherited from the standard Euclidean metric. Since the horizontal-lifted vector field is horizontal, by the above intuition, it induces a movement purely across equivalence classes, making it a perfect correspondence to a vector field on the quotient space, and avoiding "unnecessary" movements that are within an equivalence class.

## D  PROOFS

### D.1  PROOF OF THEOREM 1

**Theorem 1'.** Assume $\{\mathbf{x}_t\}_{t\in[0,T]}$ is a diffusion process on $\mathcal{M}$, specified by the following SDE:

$$\mathrm{d}\mathbf{x}_t = \mathbf{f}_t(\mathbf{x}_t)\,\mathrm{d}t + \sigma_t\,\mathrm{d}\mathbf{w}_t, \quad \mathbf{x}_0 \sim p_{\text{prior}}, \tag{6'}$$

where $\mathbf{f}_t$ is a $\mathcal{G}$-equivariant vector field on $\mathcal{M}$, $\mathbf{w}_t$ is the Wiener process on $\mathcal{M}$,[5] and $p_{\text{prior}}$ is a $\mathcal{G}$-invariant distribution. Then the projected process $\{\mathbf{y}_t := \pi(\mathbf{x}_t)\}_{t\in[0,T]}$ onto the quotient space $\mathcal{Q} := \mathcal{M}/\mathcal{G}$ is specified by the following SDE:

$$\mathrm{d}\mathbf{y}_t = \left( (\pi_* \mathbf{f}_t)(\mathbf{y}_t) - \frac{\sigma_t^2}{2} \mathbf{h}(\mathbf{y}_t) \right)\mathrm{d}t + \sigma_t\,\mathrm{d}\boldsymbol{\omega}_t, \quad \mathbf{y}_0 \sim \pi_\# p_{\text{prior}}, \tag{7'}$$

where: **(1)** $\pi_* \mathbf{f}_t$ is the pushed-forward vector field of $\mathbf{f}_t$ induced by $\pi$, *i.e.*, $(\pi_* \mathbf{f}_t)(\mathbf{y}_t) := \pi_{*\mathbf{x}_t} \mathbf{f}_t(\mathbf{x}_t)$, which is the same for any $\mathbf{x}_t \in \pi^{-1}(\mathbf{y}_t)$ due to the $\mathcal{G}$-equivariance of $\mathbf{f}_t$; **(2)** $\mathbf{h}(\mathbf{y}_t) := \pi_{*\mathbf{x}_t}(\sum_{i=M-G+1}^{M} \nabla_{\mathbf{e}_i}\mathbf{e}_i)$ for any $\mathbf{x}_t \in \pi^{-1}(\mathbf{y}_t)$ is the mean curvature vector at $\mathbf{y}_t$, where $\{\mathbf{e}_i\}_{i=1}^{M}$ is an orthonormal basis of $T_{\mathbf{x}_t}\mathcal{M}$ such that $\mathcal{V}_{\mathbf{x}_t} = \text{span}\{\mathbf{e}_i\}_{i=M-G+1}^{M}$; **(3)** $\boldsymbol{\omega}_t$ is the Wiener process on $\mathcal{Q}$; and **(4)** $\pi_\# p_{\text{prior}}$ is the pushed-forward distribution of $p_{\text{prior}}$, *i.e.*, its samples can be produced by $\mathbf{y}_0 = \pi(\mathbf{x}_0)$ where $\mathbf{x}_0 \sim p_{\text{prior}}$.

*Proof.* As $\mathbf{x}_t$ is a diffusion process on $\mathcal{M}$ given by the the SDE $\mathrm{d}\mathbf{x}_t = \mathbf{f}_t(\mathbf{x}_t)\,\mathrm{d}t + \sigma_t\,\mathrm{d}\mathbf{w}_t$, by Prop. B.2.9 and Def. B.2.4, $\mathbf{x}_t$ is an $\mathcal{L}_t$-diffusion and the generator is

$$\mathcal{L}_t = \mathbf{f}_t + \frac{\sigma_t^2}{2}\Delta^{\mathcal{M}}.$$

For any $\mathbf{x} \in \mathcal{M}$, let $\{\mathbf{e}_i\}_{i=1}^{M}$ be an orthonormal basis of $T_{\mathbf{x}}\mathcal{M}$ such that $\mathcal{H}_{\mathbf{x}} = \text{span}\{\mathbf{e}_i\}_{i=1}^{M-G}$, and $\mathcal{V}_{\mathbf{x}} = \text{span}\{\mathbf{e}_i\}_{i=M-G+1}^{M}$. Then by the Riemannian submersion construction of $\pi : \mathcal{M} \to \mathcal{Q}$ (see Appx. C), $\{\bar{\mathbf{e}}_i := \pi_{*\mathbf{x}}\mathbf{e}_i\}_{i=1}^{M-G}$ is an orthonormal basis of $T_{\pi(\mathbf{x})}\mathcal{Q}$. Such a basis as a smooth function of $\mathbf{x}$ always exists in a neighborhood, so locally each $\mathbf{e}_i$ can be viewed as a vector field. Let $\nabla^{\mathcal{M}}$ and $\nabla^{\mathcal{Q}}$ be the Levi-Civita connections on $\mathcal{M}, \mathcal{Q}$, respectively, where $\nabla^{\mathcal{Q}}$ is induced from the Riemannian metric inherited from $\mathcal{M}$. Using the local expression of the Laplace-Beltrami operator (Def. B.1.11), the generator is given by

$$\mathcal{L}_t = \mathbf{f}_t + \frac{\sigma_t^2}{2}\Delta^{\mathcal{M}} = \mathbf{f}_t + \frac{\sigma_t^2}{2}\sum_{i=1}^{M}(\mathbf{e}_i(\mathbf{e}_i(\cdot)) - \nabla_{\mathbf{e}_i}^{\mathcal{M}}\mathbf{e}_i) = \left( \mathbf{f}_t - \frac{\sigma_t^2}{2}\sum_{i=1}^{M}\nabla_{\mathbf{e}_i}^{\mathcal{M}}\mathbf{e}_i \right) + \frac{\sigma_t^2}{2}\sum_{i=1}^{M}\mathbf{e}_i^2.$$

Then the process is the solution to the following Stratonovitch SDE

$$\mathrm{d}\mathbf{x}_t = \mathbf{v}^{(0)}(\mathbf{x}_t, t)\mathrm{d}t + \sum_{i=1}^{M}\mathbf{v}^{(i)}(\mathbf{x}_t, t) \circ \mathrm{d}\mathbf{w}_t^i,$$

$$\text{where} \quad \mathbf{v}^{(0)} := \mathbf{f}_t - \frac{\sigma_t^2}{2}\sum_{i=1}^{M}\nabla_{\mathbf{e}_i}^{\mathcal{M}}\mathbf{e}_i, \quad \text{and } \mathbf{v}^{(i)} := \sigma_t\mathbf{e}_i \text{ for } i = 1, \cdots, M.$$

By Def. B.2.8, for all $f \in C^\infty(\mathcal{M})$,

$$f(\mathbf{x}_t) = f(\mathbf{x}_0) + \int_0^t \left( \mathbf{v}^{(0)}(f)(\mathbf{x}_s, s)\mathrm{d}s + \sum_{i=1}^{M}\mathbf{v}^{(i)}(f)(\mathbf{x}_s, s) \circ \mathrm{d}\mathbf{w}_s^i \right).$$

---

[5]The Wiener process $\mathbf{w}_t$ is defined under the Riemannian metric (Def. B.2.4). The transition probability it induces is $\mathcal{G}$-invariant when the group $\mathcal{G}$ acts on the Riemannian manifold $\mathcal{M}$ isometrically (Def. C.1.5**(1)**).

Let $\bar{f} \in C^\infty(\mathcal{Q})$, then $f := \bar{f} \circ \pi \in C^\infty(\mathcal{M})$, then

$$
\bar{f}(\pi(\mathbf{x}_t)) = \bar{f}(\pi(\mathbf{x}_0)) + \int_0^t \left( \mathbf{v}^{(0)}(\bar{f} \circ \pi)(\mathbf{x}_s, s)\mathrm{d}s + \sum_{i=1}^M \mathbf{v}^{(i)}(\bar{f} \circ \pi)(\mathbf{x}_s, s) \circ \mathrm{d}\mathbf{w}_s^i \right),
$$

$$
= \bar{f}(\pi(\mathbf{x}_0)) + \int_0^t \left( (\pi_*\mathbf{v}^{(0)})(\bar{f})(\pi(\mathbf{x}_s), s)\mathrm{d}s + \sum_{i=1}^M (\pi_*\mathbf{v}^{(i)})(\bar{f})(\pi(\mathbf{x}_s), s) \circ \mathrm{d}\mathbf{w}_s^i \right),
$$

by Def. B.1.5. Since $\bar{f}$ is arbitrary, by Def. B.2.8, $\mathbf{y}_t := \pi(\mathbf{x}_t)$ is the solution to:

$$
\mathrm{d}\mathbf{y}_t = \pi_*\mathbf{v}^{(0)}(\mathbf{y}_t, t)\mathrm{d}t + \sum_{i=1}^M \pi_*\mathbf{v}^{(i)}(\mathbf{y}_t, t) \circ \mathrm{d}\mathbf{w}_t^i.
$$

We first need to check that the projected vector field is well defined. In fact, we only need to check that $\pi_*\mathbf{f}$ is well defined. Since $\mathbf{f}$ is $\mathcal{G}$-equivariant, then for any $g \in \mathcal{G}$, $(L_g)_*\mathbf{f}_t(\mathbf{x}) = \mathbf{f}_t(g \cdot \mathbf{x})$. Then $\pi_*(\mathbf{f}_t(g \cdot \mathbf{x})) = \pi_*((L_g)_*\mathbf{f}_t(\mathbf{x})) = (\pi \circ L_g)_*(\mathbf{f}_t(\mathbf{x})) = \pi_*(\mathbf{f}_t(\mathbf{x}))$, where we have used the chain rule in the second-last step, and the last step holds since $\pi \circ L_g$ and $\pi$ projects to the same equivalence class ($\mathbf{x}$ and $g \cdot \mathbf{x}$ are in the same equivalence class). By a notational equivalence that $\pi_*(\mathbf{f}_t(\mathbf{x})) = \pi_*\mathbf{f}_t(\pi(\mathbf{x}))$, we know that $\pi_*\mathbf{f}_t(\mathbf{y})$ is the same on the equivalence class regardless of the choice of $\mathbf{x}$ in $\pi^{-1}(\mathbf{y})$, which implies that the projected vector field $\pi_*\mathbf{f}_t$ is well defined.

Next, we calculate the expression of the projected vector field. Since $\mathcal{H}_\mathbf{x} = \mathrm{span}\{\mathbf{e}_1, \cdots, \mathbf{e}_{M-G}\}$, $\mathcal{V}_\mathbf{x} = \mathrm{span}\{\mathbf{e}_{M-G+1}, \cdots, \mathbf{e}_M\}$, we have

$$
\pi_{*\mathbf{x}}\mathbf{e}_i = \begin{cases} \bar{\mathbf{e}}_i, & \text{if } i \leqslant M - G, \\ \mathbf{0}, & \text{if } i \geqslant M - G + 1, \end{cases}
$$

so $\pi_{*\mathbf{x}}(\mathbf{v}^{(i)}) = \sigma_t\bar{\mathbf{e}}_i$ for $i = 1, \cdots, M - G$ and $\pi_{*\mathbf{x}}(\mathbf{v}^{(i)}) = \mathbf{0}$ for $i \geqslant M - G + 1$. Moreover, since each $\mathbf{e}_i$ for $i \in \{1, \cdots, M - G\}$ lies in the horizontal space and $\pi_*\mathbf{e}_i = \bar{\mathbf{e}}$ by definition, by the uniqueness of horizontal lift, we know $\tilde{\bar{\mathbf{e}}}_i = \mathbf{e}_i$. By Prop. C.2.4, which means $\pi_*\left(\widetilde{\nabla_{\mathbf{u}^{(1)}}^\mathcal{Q} \mathbf{u}^{(2)}}\right) = \nabla_{\mathbf{u}^{(1)}}^\mathcal{Q}\mathbf{u}^{(2)} = \pi_*\left((\nabla_{\tilde{\mathbf{u}}^{(1)}}^\mathcal{M} \tilde{\mathbf{u}}^{(2)})^\mathcal{H}\right) = \pi_*(\nabla_{\tilde{\mathbf{u}}^{(1)}}^\mathcal{M} \tilde{\mathbf{u}}^{(2)})$, we know that $\pi_*(\nabla_{\mathbf{e}_i}^\mathcal{M}\mathbf{e}_i) = \nabla_{\bar{\mathbf{e}}_i}^\mathcal{Q}\bar{\mathbf{e}}_i$. Therefore, for the drift term, we have:

$$
\pi_*\mathbf{v}^{(0)}(\mathbf{y}, t) = \pi_*\mathbf{f}_t(\mathbf{y}) - \frac{\sigma_t^2}{2}\sum_{i=1}^M \pi_*(\nabla_{\mathbf{e}_i}^\mathcal{M}\mathbf{e}_i)
$$

$$
= \pi_*\mathbf{f}_t(\mathbf{y}) - \frac{\sigma_t^2}{2}\sum_{i=1}^{M-G} \pi_*(\nabla_{\mathbf{e}_i}^\mathcal{M}\mathbf{e}_i) - \frac{\sigma_t^2}{2}\sum_{i=M-G+1}^M \pi_*(\nabla_{\mathbf{e}_i}^\mathcal{M}\mathbf{e}_i)
$$

$$
= \pi_*\mathbf{f}_t(\mathbf{y}) - \frac{\sigma_t^2}{2}\sum_{i=1}^{M-G} \nabla_{\bar{\mathbf{e}}_i}^\mathcal{Q}\bar{\mathbf{e}}_i - \frac{\sigma_t^2}{2}\sum_{i=M-G+1}^M \pi_*(\nabla_{\mathbf{e}_i}^\mathcal{M}\mathbf{e}_i)
$$

$$
= \pi_*\mathbf{f}_t(\mathbf{y}) - \frac{\sigma_t^2}{2}\sum_{i=1}^{M-G} \nabla_{\bar{\mathbf{e}}_i}^\mathcal{Q}\bar{\mathbf{e}}_i - \frac{\sigma_t^2}{2}\mathbf{h}(\mathbf{y}).
$$

So the generator of the process $\mathbf{y}_t$ is

$$
\mathcal{L}_s = \pi_*\mathbf{f}_t - \frac{\sigma_t^2}{2}\sum_{i=1}^{M-G} \nabla_{\bar{\mathbf{e}}_i}^\mathcal{Q}\bar{\mathbf{e}}_i - \frac{\sigma_t^2}{2}\mathbf{h} + \frac{\sigma_t^2}{2}\sum_{i=1}^{M-G} \bar{\mathbf{e}}_i^2
$$

$$
= \left(\pi_*\mathbf{f}_t - \frac{\sigma_t^2}{2}\mathbf{h}\right) + \frac{\sigma_t^2}{2}\left(\sum_{i=1}^{M-G} \bar{\mathbf{e}}_i^2 - \sum_{i=1}^{M-G} \nabla_{\bar{\mathbf{e}}_i}^\mathcal{Q}\bar{\mathbf{e}}_i\right)
$$

$$
= \left(\pi_*\mathbf{f}_t - \frac{\sigma_t^2}{2}\mathbf{h}\right) + \frac{\sigma_t^2}{2}\Delta^\mathcal{Q}. \tag{D.1.1}
$$

Then we can conclude that the projected process $\mathbf{y}_t := \pi(\mathbf{x}_t)$ is the solution to the following SDE

$$
\mathrm{d}\mathbf{y}_t = \left((\pi_*\mathbf{f}_t)(\mathbf{y}_t) - \frac{\sigma_t^2}{2}\mathbf{h}(\mathbf{y}_t)\right)\mathrm{d}t + \sigma_t\,\mathrm{d}\boldsymbol{\omega}_t,
$$

where $\pi_* \mathbf{f}_t$ is the push-forward vector field, $\mathbf{h}(\mathbf{y}_t)$ is the mean curvature vector at $\mathbf{y}_t$ and $\boldsymbol{\omega}_t$ is the standard Wiener process on the quotient space $\mathcal{Q}$. □

## D.2    PROOF OF THEOREM 2

In Def. C.2.2, we define the horizontal lift of a vector field that generates a deterministic flow. In fact, for a stochastic process on $\mathcal{Q}$, we can define the horizontal lift for it similarly. First, we need to define the stochastic line integral, which is the integration of a one-form along the trajectory of a stochastic process.

**Definition D.2.1.** (Hsu, 2002, Prop. 2.4.2) Let $\Theta$ be a 1-form (Def. B.1.7) on $\mathcal{M}$ and $\mathbf{x}_t$ the solution to the equation

$$\mathrm{d}\mathbf{x}_t = \mathbf{v}^{(0)}(\mathbf{x}_t, t)\mathrm{d}t + \sum_{i=1}^{D} \mathbf{v}^{(i)}(\mathbf{x}_t, t) \circ \mathrm{d}\mathbf{w}_t^i.$$

Then

$$\int_0^t \Theta_{\mathbf{x}_s} \, \mathrm{d}s = \int_0^t \Theta(\mathbf{v}^{(0)})(\mathbf{x}_s) \, \mathrm{d}s + \int_0^t \sum_{i=1}^{D} \Theta(\mathbf{v}^{(i)})(\mathbf{x}_s) \circ \mathrm{d}\mathbf{w}_s^i.$$

**Definition D.2.2.** (Baudoin et al., 2024, Def. 3.1.9) A semimartingale $(\mathbf{x}_t)_t$ on $\mathcal{M}$ is called horizontal if for every 1-form $\Theta$ on $\mathcal{M}$ whose kernel contains the horizontal space $\mathcal{H}$, one has $\int_0^t \Theta_{\mathbf{x}_s} \, \mathrm{d}s = 0$, for all $t \geqslant 0$. Let $(\mathbf{y}_t)_t$ be a semimartingale on $\mathcal{Q}$ such that $\mathbf{y}_0$ is a point of $\mathcal{Q}$. Then for a given starting point $\mathbf{x}_0 \in \pi^{-1}(\mathbf{y}_0)$, there exists a unique horizontal semimartingale $\mathbf{x}_t$ on $\mathcal{M}$ such that $\mathbf{x}_t$ starts from $\mathbf{x}_0$ and $\pi(\mathbf{x}_t) = \mathbf{y}_t$ for all $t \geqslant 0$. This process $(\mathbf{x}_t)_t$ is called the horizontal lift of $(\mathbf{y}_t)_t$ from $\mathbf{x}_0$.

**Theorem 2'.** The horizontal lift of Eq. (7) has the following explicit expression:

$$\mathrm{d}\tilde{\mathbf{x}}_t = \left( P_{\tilde{\mathbf{x}}_t}(\mathbf{f}_t(\tilde{\mathbf{x}}_t)) - \frac{\sigma_t^2}{2}\tilde{\mathbf{h}}(\tilde{\mathbf{x}}_t) \right) \mathrm{d}t + \sigma_t \, \mathrm{d}\tilde{\mathbf{w}}_t, \quad \tilde{\mathbf{x}}_0 \sim p_{\mathrm{prior}}, \tag{8'}$$

where $P_{\mathbf{x}}(\mathbf{v}) := \mathbf{v}^{\mathcal{H}}$ is the horizontal projection in the tangent space of $\mathcal{M}$, $\tilde{\mathbf{h}}$ is the horizontal lift of the mean curvature vector, and $\tilde{\mathbf{w}}_t$ is the horizontal lift of the Wiener process of $\mathcal{Q}$.

***Proof.*** We only need to check the definition of the horizontal lift (Def. D.2.2). Again, for any $\mathbf{x} \in \mathcal{M}$, let $\{\mathbf{e}_i\}_{i=1}^{M}$ be an orthonormal basis of $T_{\mathbf{x}}\mathcal{M}$ such that $\mathcal{H}_{\mathbf{x}} = \mathrm{span}\{\mathbf{e}_i\}_{i=1}^{M-G}$, and $\mathcal{V}_{\mathbf{x}} = \mathrm{span}\{\mathbf{e}_i\}_{i=M-G+1}^{M}$. Then by the Riemannian submersion construction of $\pi : \mathcal{M} \to \mathcal{Q}$ (see Appx. C), $\{\bar{\mathbf{e}}_i := \pi_{*\mathbf{x}}\mathbf{e}_i\}_{i=1}^{M-G}$ is an orthonormal basis of $T_{\pi(\mathbf{x})}\mathcal{Q}$. Such a basis as a smooth function of $\mathbf{x}$ always exists in a neighborhood, so each $\mathbf{e}_i$ can be viewed as a vector field in each neighborhood. Let $\nabla^{\mathcal{M}}$ and $\nabla^{\mathcal{Q}}$ be the Levi-Civita connections on $\mathcal{M}, \mathcal{Q}$, respectively, where $\nabla^{\mathcal{Q}}$ is induced from the Riemannian metric inherited from $\mathcal{M}$.

Now we calculate the generator of the SDE in Eq. (8'). From Eq. (D.1.1), the Wiener process $\boldsymbol{\omega}_t$ on the quotient space $\mathcal{Q}$ has generator $\frac{1}{2}\Delta^{\mathcal{Q}} = \frac{1}{2}\sum_{i=1}^{M-G}(\bar{\mathbf{e}}_i^2 - \nabla^{\mathcal{Q}}_{\bar{\mathbf{e}}_i}\bar{\mathbf{e}}_i)$. Since each $\mathbf{e}_i$ for $i \in \{1, \cdots, M - G\}$ lies in the horizontal space and $\pi_*\mathbf{e}_i = \bar{\mathbf{e}}$ by definition, by the uniqueness of horizontal lift, we know $\tilde{\bar{\mathbf{e}}}_i = \mathbf{e}_i$. By Prop. C.2.4, we know that the generator of the horizontal-lifted Wiener process $\tilde{\mathbf{w}}_t$ has generator $\frac{1}{2}\sum_{i=1}^{M-G}(\mathbf{e}_i^2 - (\nabla^{\mathcal{M}}_{\mathbf{e}_i}\mathbf{e}_i)^{\mathcal{H}})$. So the generator of the horizontal lifted process in Eq. (8') is:

$$\tilde{\mathcal{L}}_t = \left( \mathbf{f}_t^{\mathcal{H}} - \frac{\sigma_t^2}{2}\tilde{\mathbf{h}} \right) + \frac{\sigma_t^2}{2} \sum_{i=1}^{M-G} \left( \mathbf{e}_i^2 - (\nabla^{\mathcal{M}}_{\mathbf{e}_i}\mathbf{e}_i)^{\mathcal{H}} \right). \tag{D.2.1}$$

Its projection under $\pi_*$ is given by:

$$\pi_*\tilde{\mathcal{L}}_t = \left( \pi_*\mathbf{f}_t - \frac{\sigma_t^2}{2}\mathbf{h} \right) + \frac{\sigma_t^2}{2} \sum_{i=1}^{M-G} \left( \bar{\mathbf{e}}_i^2 - \nabla^{\mathcal{Q}}_{\bar{\mathbf{e}}_i}\bar{\mathbf{e}}_i \right),$$

which coincides with the generator of Eq. (7) (see Eq. (D.1.1)). So the process $\pi(\tilde{\mathbf{x}}_t)$ is the same diffusion process as $\mathbf{y}_t$, where $\mathbf{y}_t$ is defined in Eq. (7).

Let $\Theta$ be a 1-form on $\mathcal{M}$ whose kernel contains the horizontal space $\mathcal{H}$ everywhere. From Eq. (D.2.1), $\tilde{\mathbf{x}}_t$ is the following SDE

$$d\mathbf{x}_t = \mathbf{v}^{(0)}(\mathbf{x}_t, t)dt + \sum_{i=1}^{M} \mathbf{v}^{(i)}(\mathbf{x}_t, t) \circ d\mathbf{w}_t^i,$$

$$\text{where } \mathbf{v}^{(0)} = \left( \mathbf{f}_t^{\mathcal{H}} - \frac{\sigma_t^2}{2}\tilde{\mathbf{h}} \right) - \frac{\sigma_t^2}{2} \sum_{i=1}^{M-G} (\nabla_{\mathbf{e}_i}^{\mathcal{M}} \mathbf{e}_i)^{\mathcal{H}}, \quad \mathbf{v}^{(i)} = \sigma_t \mathbf{e}_i.$$

Then the line integral

$$\int_0^t \Theta_{\mathbf{x}_s} \, ds = \int_0^t \sum_{i=0}^{M} \Theta(\mathbf{v}^{(i)})(\tilde{\mathbf{x}}_s) \circ d\mathbf{w}_s^i = 0,$$

since $\mathbf{v}^{(i)} \in \mathcal{H}$ and $\Theta(\mathbf{v}^{(i)}) = 0$. So we can conclude that $\tilde{\mathbf{x}}_t$ is the horizontal lift of $\mathbf{y}_t$. □

**Corollary 3'.** **(1)** The resulting random variable $\tilde{\mathbf{x}}_1$ by the lifted diffusion process Eq. (8) and the resulting $\mathbf{x}_1$ by the original diffusion process Eq. (6) follow the same distribution in the total space: $p_{\tilde{\mathbf{x}}_1} = p_{\mathbf{x}_1} = p_{\text{target}}$. **(2)** When $\sigma_t \equiv 0$ and starting from the same $\mathbf{x}_0 \in \mathcal{M}$, Eq. (8) leads to a shorter trajectory than Eq. (6) does:

$$\int_0^1 \langle P_{\tilde{\mathbf{x}}_t}(\mathbf{f}_t(\tilde{\mathbf{x}}_t)), P_{\tilde{\mathbf{x}}_t}(\mathbf{f}_t(\tilde{\mathbf{x}}_t)) \rangle \, dt \leqslant \int_0^1 \langle \mathbf{f}_t(\mathbf{x}_t), \mathbf{f}_t(\mathbf{x}_t) \rangle \, dt.$$

**Proof.** **(1)** Since $\mathbf{x}_0$ and $\tilde{\mathbf{x}}_0$ follow the same $\mathcal{G}$-invariant distribution $p_{\text{prior}}$, and by Thm. 1 and that the lifted process projects to the same quotient-space process before the lift, we know that $\pi(\mathbf{x}_t)$ follows the same diffusion process as $\pi(\mathbf{x}_t)$ on $\mathcal{Q}$. Since $\mathbf{f}_t$ is $\mathcal{G}$-equivariant and the Wiener process is $\mathcal{G}$-invariant, we know that the transition kernel $p(\mathbf{x}_1 \mid \mathbf{x}_0)$ is $\mathcal{G}$-equivariant. Together with that $p_{\text{prior}}$ is $\mathcal{G}$-invariant, we know that $p(\mathbf{x}_1)$ is $\mathcal{G}$-invariant (Xu et al., 2022, Prop. 1). On the side of the lifted quotient-space diffusion process, since the lifted process does not have a vertical movement, *i.e.*, the process effectively does not apply any group action, we know that $p(\tilde{\mathbf{x}}_1 \mid \tilde{\mathbf{x}}_0)$ is also $\mathcal{G}$-equivariant, and together with the $\mathcal{G}$-invariance of $p_{\text{prior}}$, we know that $p(\tilde{\mathbf{x}}_1)$ is $\mathcal{G}$-invariant as well. Since $p(\mathbf{x}_1)$ and $p(\tilde{\mathbf{x}}_1)$ project to the same distribution on $\mathcal{Q}$ and they are $\mathcal{G}$-invariant, we know that they are the same distribution on the total space $\mathcal{M}$.

**(2)** When $\sigma_t \equiv 0$, the process becomes deterministic and solves the ODEs $\frac{d\mathbf{x}_t}{dt} = \mathbf{f}_t(\mathbf{x}_t)$ and $\frac{d\tilde{\mathbf{x}}_t}{dt} = P_{\tilde{\mathbf{x}}_t}(\mathbf{f}_t(\tilde{\mathbf{x}}_t))$. Since they start from the same point $\mathbf{x}_0 = \tilde{\mathbf{x}}_0$, and the vector fields $\mathbf{f}_t$ and $P_{\tilde{\mathbf{x}}_t}(\mathbf{f}_t(\tilde{\mathbf{x}}_t))$ differ only in a vertical movement, *i.e.*, movement by a group action, we know that there exists $g_t \in \mathcal{G}$ such that $\mathbf{x}_t = g_t \cdot \tilde{\mathbf{x}}_t$. By leveraging the $\mathcal{G}$-equivariance of $\mathbf{f}_t$ and the isometry of group action, the lengths of the two processes can then be related by:

$$\int_0^1 \langle \mathbf{f}_t(\mathbf{x}_t), \mathbf{f}_t(\mathbf{x}_t) \rangle_{\mathbf{x}_t} dt = \int_0^1 \langle \mathbf{f}_t(g_t \cdot \tilde{\mathbf{x}}_t), \mathbf{f}_t(g_t \cdot \tilde{\mathbf{x}}_t) \rangle_{\mathbf{x}_t} dt$$

$$= \int_0^1 \langle (L_{g_t})_{*\tilde{\mathbf{x}}_t} \mathbf{f}_t(\tilde{\mathbf{x}}_t), (L_{g_t})_{*\tilde{\mathbf{x}}_t} \mathbf{f}_t(\tilde{\mathbf{x}}_t) \rangle_{g_t \cdot \tilde{\mathbf{x}}_t} dt$$

$$= \int_0^1 \langle \mathbf{f}_t(\tilde{\mathbf{x}}_t), \mathbf{f}_t(\tilde{\mathbf{x}}_t) \rangle_{\tilde{\mathbf{x}}_t} dt = \int_0^1 \left( \langle \mathbf{f}_t(\tilde{\mathbf{x}}_t)^{\mathcal{H}}, \mathbf{f}_t(\tilde{\mathbf{x}}_t)^{\mathcal{H}} \rangle_{\tilde{\mathbf{x}}_t} + \langle \mathbf{f}_t(\tilde{\mathbf{x}}_t)^{\mathcal{V}}, \mathbf{f}_t(\tilde{\mathbf{x}}_t)^{\mathcal{V}} \rangle_{\tilde{\mathbf{x}}_t} \right) dt$$

$$\geqslant \int_0^1 \langle \mathbf{f}_t(\tilde{\mathbf{x}}_t)^{\mathcal{H}}, \mathbf{f}_t(\tilde{\mathbf{x}}_t)^{\mathcal{H}} \rangle_{\tilde{\mathbf{x}}_t} dt = \int_0^1 \langle P_{\tilde{\mathbf{x}}_t}(\mathbf{f}_t(\tilde{\mathbf{x}}_t)), P_{\tilde{\mathbf{x}}_t}(\mathbf{f}_t(\tilde{\mathbf{x}}_t)) \rangle_{\tilde{\mathbf{x}}_t} dt.$$

□

## D.3 PROOF OF THEOREM 4

**Theorem 4'.** Assume $\mathbf{x}_t$ is a diffusion process in a manifold $\mathcal{M}$ that can be embedded into a Euclidean space, specified by the SDE: $d\mathbf{x}_t = \mathbf{f}_t(\mathbf{x}_t) \, dt + \sigma_t \, d\mathbf{w}_t$, where $\mathbf{f}_t(\mathbf{x}_t)$ is $\mathcal{G}$-equivariant and $\mathbf{x}_0 \sim p_{\text{prior}}$ is $\mathcal{G}$-invariant. Then the lifted quotient-space diffusion process on $\mathcal{Q} := \mathcal{M}/\mathcal{G}$ onto the total space $\mathcal{M}$ is given by:

$$d\tilde{\mathbf{x}}_t = \left( P_{\tilde{\mathbf{x}}_t}(\mathbf{f}_t(\tilde{\mathbf{x}}_t)) - \frac{\sigma_t^2}{2}\tilde{\mathbf{h}}(\tilde{\mathbf{x}}_t) \right) dt + \sigma_t P_{\tilde{\mathbf{x}}_t} \, d\mathbf{w}_t, \quad \tilde{\mathbf{x}}_0 \sim p_{\text{prior}}, \tag{9'}$$

where $P_{\tilde{\mathbf{x}}_t} \, \mathrm{d}\mathbf{w}_t$ can be simulated by projecting the infinitesimal Gaussian sample using $P_{\tilde{\mathbf{x}}_t}$ in each step. Particularly, for the shape space $\mathcal{Q} := \mathbb{R}^{3N}/\mathrm{SE}(3) = \mathbb{R}^{3N}_{\mathrm{CoM}\circ}/\mathrm{SO}(3)$, the horizontal projection $P$ and the $\tilde{\mathbf{h}}$ vector field have the following explicit expressions: for any $\mathbf{x} = [\vec{x}^{(n)}]_n \in \mathbb{R}^{3N}_{\mathrm{CoM}\circ}$, which is CoM-free, and any $\mathbf{v} = [\vec{v}^{(n)}]_n \in T_{\mathbf{x}}\mathbb{R}^{3N}_{\mathrm{CoM}\circ}$, which is momentum-free,

$$P_{\mathbf{x}}(\mathbf{v}) = \left[\vec{v}^{(n)} - \mathbf{K}(\mathbf{x})^{-1}\left(\sum_{n'=1}^{N} \vec{x}^{(n')} \times \vec{v}^{(n')}\right) \times \vec{x}^{(n)}\right]_n, \text{ and} \tag{10'}$$

$$\tilde{\mathbf{h}}(\mathbf{x}) = \left[\mathbf{K}(\mathbf{x})^{-1}\vec{x}^{(n)} - \mathrm{tr}(\mathbf{K}(\mathbf{x})^{-1})\vec{x}^{(n)}\right]_n, \tag{D.3.1}$$

$$\text{where } \mathbf{K}(\mathbf{x}) := \sum_{n=1}^{N}\|\vec{x}^{(n)}\|^2\mathbf{I} - \sum_{n=1}^{N}\vec{x}^{(n)}\vec{x}^{(n)^\top} \in \mathbb{R}^{3\times 3}, \tag{D.3.2}$$

and "$\times$" denotes the cross product in $\mathbb{R}^3$.

***Proof.*** We unroll the proof by first deriving the general expression Eq. (9') in the case of Euclidean total space, then deriving the expression for the projection operator $P_{\mathbf{x}}(\mathbf{v})$ for $\mathcal{G} = \mathrm{SO}(3)$, and the horizontal-lifted mean curvature vector field $\tilde{\mathbf{h}}(\mathbf{x})$ for $\mathcal{G} = \mathrm{SO}(3)$. Expressions for the SO(3) case are expressed in the total space $\mathcal{M}$ defined in Appx. C.3, which can be embedded in the Euclidean space $\mathbb{R}^{3N}$.

**The general expression Eq. (9').** Again, for any $\mathbf{x} \in \mathcal{M}$, let $\{\mathbf{e}_i\}_{i=1}^M$ be an orthonormal basis of $T_{\mathbf{x}}\mathcal{M}$ such that $\mathcal{H}_{\mathbf{x}} = \mathrm{span}\{\mathbf{e}_i\}_{i=1}^{M-G}$, and $\mathcal{V}_{\mathbf{x}} = \mathrm{span}\{\mathbf{e}_i\}_{i=M-G+1}^M$. Then by the Riemannian submersion construction of $\pi : \mathcal{M} \to \mathcal{Q}$ (see Appx. C), $\{\bar{\mathbf{e}}_i := \pi_{*\mathbf{x}}\mathbf{e}_i\}_{i=1}^{M-G}$ is an orthonormal basis of $T_{\pi(\mathbf{x})}\mathcal{Q}$. Such a basis as a smooth function of $\mathbf{x}$ always exists in a neighborhood, so each $\mathbf{e}_i$ can be viewed as a vector field in each neighborhood. Let $\nabla^{\mathcal{M}}$ and $\nabla^{\mathcal{Q}}$ be the Levi-Civita connection on $\mathcal{M}, \mathcal{Q}$, respectively, where $\nabla^{\mathcal{Q}}$ is induced from the Riemannian metric inherited from $\mathcal{M}$.

As shown in Appx. D.2, Eq. (D.2.1), the horizontal lift of Eq. (8) has the generator

$$\mathcal{L}_t = \left(\mathbf{f}_t^{\mathcal{H}} - \frac{\sigma_t^2}{2}\tilde{\mathbf{h}}\right) + \frac{\sigma_t^2}{2}\sum_{i=1}^{M-G}\left(\mathbf{e}_i^2 - (\nabla_{\mathbf{e}_i}^{\mathcal{M}}\mathbf{e}_i)^{\mathcal{H}}\right).$$

Since $\mathcal{L}_{\mathbf{e}_i}\mathbf{e}_i = \mathbf{e}_i(\mathbf{e}_i(\cdot)) - \mathbf{e}_i(\mathbf{e}_i(\cdot)) = 0$, by Prop. C.2.4, we know that $(\nabla_{\mathbf{e}_i}^{\mathcal{M}}\mathbf{e}_i)^{\mathcal{V}} = \mathbf{0}$ for $i = \{1,\cdots,M-G\}$, so

$$\mathcal{L}_t = \left(\mathbf{f}_t^{\mathcal{H}} - \frac{\sigma_t^2}{2}\tilde{\mathbf{h}}\right) + \frac{\sigma_t^2}{2}\sum_{i=1}^{M-G}\left(\mathbf{e}_i^2 - (\nabla_{\mathbf{e}_i}^{\mathcal{M}}\mathbf{e}_i)\right).$$

Since $\mathcal{M}$ is embedded in a Euclidean space, we have $\nabla_{\mathbf{e}_i}^{\mathcal{M}}\mathbf{e}_i = \sum_{j=1}^M \mathbf{e}_i(\mathbf{e}_i^j)\partial_j$, where $\mathbf{e}_i^j$ is the $j$-th component of $\mathbf{e}_i$ and $\partial_j := \partial/\partial x_j$ is the derivative w.r.t a standard coordinate system of the Euclidean space (*i.e.*, these $\{\partial_j\}_{j=1}^M$ are a standard orthonormal basis frame for tangent spaces of the Euclidean space, which are isometrically isomorphic to the Euclidean space itself). Since $\mathbf{f}_t^{\mathcal{H}}(\mathbf{x}) = P_{\mathbf{x}}\mathbf{f}_t(\mathbf{x})$, then the generator becomes

$$\mathcal{L}_t = \left(\mathbf{f}_t^{\mathcal{H}}(\mathbf{x}) - \frac{\sigma_t^2}{2}\tilde{\mathbf{h}}(\mathbf{x})\right) + \frac{\sigma_t^2}{2}\sum_{i=1}^{M-G}\mathbf{e}_i^2 - (\nabla_{\mathbf{e}_i}^{\mathcal{M}}\mathbf{e}_i)$$

$$= \left(P_{\mathbf{x}}\mathbf{f}_t(\mathbf{x}) - \frac{\sigma_t^2}{2}\tilde{\mathbf{h}}(\mathbf{x})\right) + \frac{\sigma_t^2}{2}\sum_{i=1}^{M-G}\sum_{j,k=1}^M \mathbf{e}_i^j(\partial_j\mathbf{e}_i^k)\partial_k + \mathbf{e}_i^j\mathbf{e}_i^k\partial_j\partial_k - \mathbf{e}_i^j(\partial_j\mathbf{e}_i^k)\partial_k$$

$$= \left(P_{\mathbf{x}}\mathbf{f}_t(\mathbf{x}) - \frac{\sigma_t^2}{2}\tilde{\mathbf{h}}(\mathbf{x})\right) + \frac{\sigma_t^2}{2}\sum_{i=1}^{M-G}\sum_{j,k=1}^M \mathbf{e}_i^j\mathbf{e}_i^k\partial_j\partial_k$$

$$= \left(P_{\mathbf{x}}\mathbf{f}_t(\mathbf{x}) - \frac{\sigma_t^2}{2}\tilde{\mathbf{h}}(\mathbf{x})\right) + \frac{\sigma_t^2}{2}\sum_{j,k=1}^M (P_{\mathbf{x}})^{jk}\partial_j\partial_k$$

$$= \left(P_{\mathbf{x}}\mathbf{f}_t(\mathbf{x}) - \frac{\sigma_t^2}{2}\tilde{\mathbf{h}}(\mathbf{x})\right) + \frac{\sigma_t^2}{2}\sum_{j,k=1}^M (P_{\mathbf{x}}P_{\mathbf{x}}^\top)^{jk}\partial_j\partial_k,$$

where we have defined $P_{\mathbf{x}} := \sum_{i=1}^{M-G} \mathbf{e}_i \mathbf{e}_i^\top$ as the projection operator. Then $\mathcal{L}_t$ is the generator of

$$d\tilde{\mathbf{x}}_t = \left( P_{\tilde{\mathbf{x}}_t} (\mathbf{f}_t(\tilde{\mathbf{x}}_t)) - \frac{\sigma_t^2}{2} \tilde{\mathbf{h}}(\tilde{\mathbf{x}}_t) \right) dt + \sigma_t P_{\tilde{\mathbf{x}}_t} d\mathbf{w}_t.$$

Now we deduce the explicit expressions for the case when $\mathcal{G} = \mathrm{SO}(3)$.

**The expression for the projection operator $P_{\mathbf{x}}(\mathbf{v})$ for $\mathcal{G} = \mathrm{SO}(3)$.** Recall from Appx. C.3, the tangent space $T_{\mathbf{x}}\mathcal{M}$ of $\mathcal{M}$ at $\mathbf{x}$ can be explicitly expressed as (Eqs. (C.3.3, C.3.4) ):

- The vertical tangent space $\mathcal{V}_{\mathbf{x}}$:

$$\mathcal{V}_{\mathbf{x}} = \{ [\vec{\omega} \times \vec{x}^{(1)}, \ \vec{\omega} \times \vec{x}^{(2)}, \ \cdots, \ \vec{\omega} \times \vec{x}^{(N)}] \mid \vec{\omega} \in \mathbb{R}^3 \}.$$

- The horizontal space $\mathcal{H}_{\mathbf{x}}$, which is the orthogonal complement of the vertical space:

$$\mathcal{H}_{\mathbf{x}} = \left\{ \mathbf{v} = [\vec{v}^{(1)}, \cdots, \vec{v}^{(N)}] \in \mathbb{R}^{3N} \ \Big| \ \sum_{n=1}^N \vec{v}^{(n)} = \vec{0}, \ \sum_{n=1}^N \vec{x}^{(n)} \times \vec{v}^{(n)} = \vec{0} \right\}.$$

The horizontal projection mapping is defined by $P_{\mathbf{x}}(\mathbf{v}) = \mathbf{v}^{\mathcal{H}} = \mathbf{v} - \mathbf{v}^{\mathcal{V}}, \forall \mathbf{v} \in T_{\mathbf{x}}\mathcal{M}$, and we can find an explicit expression of it. By definition, $\sum_{n=1}^N \vec{x}^{(n)} \times \vec{v}^{(n)\mathcal{H}} = \vec{0}$, then

$$\sum_{n=1}^N \vec{x}^{(n)} \times \vec{v}^{(n)} = \sum_{n=1}^N \vec{x}^{(n)} \times \vec{v}^{(n)\mathcal{V}}.$$

Assume $\mathbf{v}^{\mathcal{V}} = [\vec{\omega} \times \vec{x}^{(1)}, \ \vec{\omega} \times \vec{x}^{(2)}, \ \cdots, \ \vec{\omega} \times \vec{x}^{(N)}]$, then

$$\sum_{n=1}^N \vec{x}^{(n)} \times \vec{v}^{(n)} = \sum_{n=1}^N \vec{x}^{(n)} \times \vec{v}^{(n)\mathcal{V}} = \sum_{n=1}^N \vec{x}^{(n)} \times (\vec{\omega} \times \vec{x}^{(n)})$$

$$= \sum_{n=1}^N \left\langle \vec{x}^{(n)}, \vec{x}^{(n)} \right\rangle \vec{\omega} - \left\langle \vec{x}^{(n)}, \vec{\omega} \right\rangle \vec{x}^{(n)} = \left( \sum_{n=1}^N \|\vec{x}^{(n)}\|^2 \mathbf{I} - \sum_{n=1}^N \vec{x}^{(n)} \vec{x}^{(n)\top} \right) \vec{\omega},$$

where we have used the identity $\vec{x}^{(n)} \times (\vec{\omega} \times \vec{x}^{(n)}) = \left\langle \vec{x}^{(n)}, \vec{x}^{(n)} \right\rangle \vec{\omega} - \left\langle \vec{x}^{(n)}, \vec{\omega} \right\rangle \vec{x}^{(n)}$. By denoting

$$\mathbf{K}(\mathbf{x}) := \sum_{n=1}^N \|\vec{x}^{(n)}\|^2 \mathbf{I} - \sum_{n=1}^N \vec{x}^{(n)} \vec{x}^{(n)\top}, \tag{D.3.3}$$

we have $\vec{\omega} = \mathbf{K}(\mathbf{x})^{-1} (\sum_{n=1}^N \vec{x}^{(n)} \times \vec{v}^{(n)})$, and

$$\mathbf{v}^{\mathcal{V}} = [\vec{\omega} \times \vec{x}^{(n)}]_n = \left[ \mathbf{K}(\mathbf{x})^{-1} \left( \sum_{n'=1}^N \vec{x}^{(n')} \times \vec{v}^{(n')} \right) \times \vec{x}^{(n)} \right]_n,$$

where the last cross-product is applied on each 3-dimensional coordinate vector $\vec{x}^{(n)}$. Henceforth, we have:

$$P_{\mathbf{x}}(\mathbf{v}) = \mathbf{v} - \mathbf{v}^{\mathcal{V}} = \left[ \vec{v}^{(n)} - \mathbf{K}(\mathbf{x})^{-1} \left( \sum_{n'=1}^N \vec{x}^{(n')} \times \vec{v}^{(n')} \right) \times \vec{x}^{(n)} \right]_n, \quad \forall \mathbf{v} \in T_{\mathbf{x}}\mathcal{M}.$$

**The expression for the horizontal-lifted mean curvature vector field $\tilde{\mathbf{h}}(\mathbf{x})$ for $\mathcal{G} = \mathrm{SO}(3)$.**
**(a)** We start with a characterization of the horizontal-lifted mean curvature vector field $\tilde{\mathbf{h}}(\mathbf{x})$ in the general case. As we have mentioned, $\tilde{\mathbf{h}}(\mathbf{x})$ reflects the effect of the change of the volume of the equivalence class, which helps us derive an explicit expression of it. The volume can be defined by a Riemannian structure. As each equivalence class can be seen to be induced by the Lie group, we first establish a relation between the two spaces. When the Lie group $\mathcal{G}$ acts smoothly, freely, properly, and isometrically on $\mathcal{M}$, we can define a mapping $\Phi^{\mathbf{x}} : \mathcal{G} \to \mathcal{M}, g \mapsto g \cdot \mathbf{x}$ which identifies the Lie group to the equivalence class $\pi(\mathbf{x})$ that $\mathbf{x}$ is in. This $\Phi^{\mathbf{x}}$ is a bijection since freeness indicates that if $g \cdot \mathbf{x} = g' \cdot \mathbf{x}$, then $g = g'$.

Using this bijection, we can induce structures on $\mathcal{G}$ from those on $\mathcal{M}$. Firstly, a Riemannian structure can be defined. Let $\{g^i\}_{i=1}^{G}$ be a coordinate chart of $\mathcal{G}$, and $\{\mathbf{g}_i\}_{i=1}^{G}$ be the induced frame on $\mathcal{G}$, $i.e.$, $\{\mathbf{g}_i\}_{i=1}^{G}$ is a basis set of $T_g\mathcal{G}$. Then a Riemannian metric can be defined by:

$$\mathbf{G}_{ij}^{\mathbf{x}}(g) := \langle \mathbf{g}_i, \mathbf{g}_j \rangle_g^{\mathcal{G}, \mathbf{x}} := \langle \Phi_{*g}^{\mathbf{x}} \mathbf{g}_i, \Phi_{*g}^{\mathbf{x}} \mathbf{g}_j \rangle_{\Phi^{\mathbf{x}}(g)}^{\mathcal{M}}. \tag{D.3.4}$$

The Riemannian structure defines a measure on $\mathcal{G}$ in the form of the volume form ($i.e.$, top-ranked (rank-$G$) exterior form on $\mathcal{G}$), which can be explicitly expressed by:

$$\mathrm{Vol}^{\mathbf{x}}(g) := \sqrt{\det(\mathbf{G}^{\mathbf{x}}(g))} \, \mathrm{d}g^1 \wedge \cdots \wedge \mathrm{d}g^G,$$

where $\{\mathrm{d}g^i\}_{i=1}^{G}$ is a basis set of $T_g^*\mathcal{G}$ and $\mathrm{d}g^i(\mathbf{g}_j) = \mathbf{g}_j(g^i) = \delta_j^i$. Through the embedding mapping $\Phi^{\mathbf{x}}$, the volume form can also be seen as a measure on the equivalence class $\pi(\mathbf{x})$ through $\mathbf{x}$.

The relation between the horizontal-lifted mean curvature vector field and the volume of equivalence class lies in the effect under time variation. For this, we first define how a spatial function can be extended to vary in time (unnecessarily uniquely).

**Definition D.3.1.** Let $\Phi : \mathcal{G} \to \mathcal{M}$ be an immersion. A **smooth variation** of $\Phi$ is a smooth mapping $\Phi_{\cdot}(\cdot) : (-\epsilon, \epsilon) \times \mathcal{G} \to \mathcal{M}$ satisfying:

- For any $t \in (-\epsilon, \epsilon)$, $\Phi_t(\cdot)$ is an immersion;

- $\Phi_0(\cdot) = \Phi(\cdot)$.

The relation is explicitly given by the following conclusion:

**Proposition D.3.2.** *(First variation of volume (Chavel, 1995, Exercise. III.14)) Let $\Phi_t^{\mathbf{x}}$ be a smooth variation of $\Phi^{\mathbf{x}}$. Define the corresponding time-dependent Riemannian metric tensor $\mathbf{G}_{t,ij}^{\mathbf{x}}(g) := \langle (\Phi_t^{\mathbf{x}})_{*g} \mathbf{g}_i, (\Phi_t^{\mathbf{x}})_{*g} \mathbf{g}_j \rangle_{\Phi_t^{\mathbf{x}}(g)}^{\mathcal{M}}$, and $\mathrm{Vol}_t^{\mathbf{x}}(g) := \sqrt{\det(\mathbf{G}_t^{\mathbf{x}}(g))} \, \mathrm{d}g^1 \wedge \cdots \wedge \mathrm{d}g^G$ the corresponding volume form. Then the mean curvature vector $\tilde{\mathbf{h}}(\mathbf{x})$ satisfies the following formula:*

$$\frac{\mathrm{d}}{\mathrm{d}t}\bigg|_{t=0} \int_{\mathcal{G}} \mathrm{Vol}_t^{\mathbf{x}}(g) = -\int_{\mathcal{G}} \langle \tilde{\mathbf{h}}(\Phi_0^{\mathbf{x}}(g)), \mathbf{v}_{\Phi_0^{\mathbf{x}}(g)} \rangle \mathrm{Vol}_0^{\mathbf{x}}(g), \tag{D.3.5}$$

*where $\mathbf{v}_{\Phi_0^{\mathbf{x}}(g)} := \dot{\gamma}_0$ for which $(\gamma_t)_t$ is the curve $\gamma_t = \Phi_t^{\mathbf{x}}(g)$.*

This proposition formalizes the intuition that the mean curvature vector represents the effect of volume change of equivalence class. The left hand side represents the change rate of $\int_{\mathcal{G}} \mathrm{Vol}_t^{\mathbf{x}}(g)$, $i.e.$, the volume of the equivalence class through $\mathbf{x}$, and the equality on the right hand side indicates that this change is given by the projection of the equivalence-class movement $\mathbf{v}$ in the direction of the mean curvature vector $\tilde{\mathbf{h}}(\mathbf{x})$. Particularly, as the mean curvature vector $\tilde{\mathbf{h}}(\mathbf{x})$ is horizontal everywhere, if the movement $\mathbf{v}$ is vertical (almost) everywhere, then the volume does not change, which matches the intuition.

**(b)** Before diving into the $\mathcal{G} = \mathrm{SO}(3)$ case, we first derive a general explicit expression for $\tilde{\mathbf{h}}(\mathbf{x})$ under a common construction that common Lie groups have, including $\mathrm{SO}(3)$. Let the coordinate system of $\mathcal{G}$ be defined in a left-invariant way, $i.e.$,

$$\mathbf{g}_i|_g = (L_g)_{*e} \mathbf{g}_i|_e.$$

Then consider the mapping $\psi_g^{\mathbf{x}}(g') := (gg') \cdot \mathbf{x}$, which leads to two equivalent expressions from its differential map. The first leverages $\psi_g^{\mathbf{x}}(g') = \Phi^{\mathbf{x}}(L_g(g'))$ and gives $(\psi_g^{\mathbf{x}})_{*g'} \mathbf{g} = \Phi_{*gg'}^{\mathbf{x}}(L_g)_{*g'} \mathbf{g}$ for any $\mathbf{g} \in T_{g'}\mathcal{G}$. The second leverages $\psi_g^{\mathbf{x}}(g') = g \cdot (g' \cdot \mathbf{x}) = L_g(\Phi^{\mathbf{x}}(g'))$ and gives $(\psi_g^{\mathbf{x}})_{*g'} \mathbf{g} = (L_g)_{*(g' \cdot \mathbf{x})} \Phi_{*g'}^{\mathbf{x}} \mathbf{g}$, which indicate that:

$$\Phi_*^{\mathbf{x}}(L_g)_* = (L_g)_* \Phi_*^{\mathbf{x}}.$$

By further noting the isometry of the group action on $\mathcal{M}$, the metric defined in Eq. (D.3.4) can be simplified as:

$$\mathbf{G}_{ij}^{\mathbf{x}}(g) = \langle \Phi_{*g}^{\mathbf{x}} \mathbf{g}_i, \Phi_{*g}^{\mathbf{x}} \mathbf{g}_j \rangle_{\Phi^{\mathbf{x}}(g)}^{\mathcal{M}} = \langle \Phi_{*g}^{\mathbf{x}}(L_g)_{*e} \mathbf{g}_i|_e, \Phi_{*g}^{\mathbf{x}}(L_g)_{*e} \mathbf{g}_j|_e \rangle_{g \cdot \mathbf{x}}^{\mathcal{M}}$$

$$= \langle (L_g)_{*\mathbf{x}} \Phi_{*e}^{\mathbf{x}} \mathbf{g}_i|_e, (L_g)_{*\mathbf{x}} \Phi_{*e}^{\mathbf{x}} \mathbf{g}_j|_e \rangle_{g \cdot \mathbf{x}}^{\mathcal{M}} = \langle \Phi_{*e}^{\mathbf{x}} \mathbf{g}_i|_e, \Phi_{*e}^{\mathbf{x}} \mathbf{g}_j|_e \rangle_{\mathbf{x}}^{\mathcal{M}} = \mathbf{G}_{ij}^{\mathbf{x}}(e), \tag{D.3.6}$$

which turns out to be constant over $\mathcal{G}$. This turns the left-hand-side of Eq. (D.3.5) simplified to:

$$\int_{\mathcal{G}} \partial_{t=0} \sqrt{\det(\mathbf{G}_t^{\mathbf{x}})} \, \mathrm{d}g^1 \wedge \cdots \wedge \mathrm{d}g^G = \int_{\mathcal{G}} \partial_{t=0} \log \sqrt{\det(\mathbf{G}_t^{\mathbf{x}})} \sqrt{\det(\mathbf{G}_0^{\mathbf{x}})} \, \mathrm{d}g^1 \wedge \cdots \wedge \mathrm{d}g^G$$

$$= \partial_{t=0} \log \sqrt{\det(\mathbf{G}_t^{\mathbf{x}})} \int_{\mathcal{G}} \mathrm{Vol}_0^{\mathbf{x}}(g) = V_0^{\mathbf{x}} \partial_{t=0} \log \sqrt{\det(\mathbf{G}_t^{\mathbf{x}})},$$

where $V_t^{\mathbf{x}} := \int_{\mathcal{G}} \mathrm{Vol}_t^{\mathbf{x}}(g)$ is the volume of the equivalence class $\pi(\mathbf{x})$ induced from $\Phi_t^{\mathbf{x}}$.

For the time-dependent embedding $\Phi_t^{\mathbf{x}}(g)$, we consider a special type that is induced from an left-invariant vector field $\mathbf{v}$ on the total space $\mathcal{M}$:

$$\Phi_t^{\mathbf{x}}(g) := \mathrm{Exp}_{g \cdot \mathbf{x}}(t \mathbf{v}_{g \cdot \mathbf{x}}),$$

for $t$ around zero. This definition indicates that $\dot\gamma_0$ in Eq. (D.3.5) is:

$$\partial_{t=0} \Phi_t^{\mathbf{x}}(g) = \mathbf{v}_{\Phi_0^{\mathbf{x}}(g)} = \mathbf{v}_{g \cdot \mathbf{x}} = (L_g)_{*\mathbf{x}} \mathbf{v}_{\mathbf{x}}, \tag{D.3.7}$$

which is left-invariant by definition. Moreover, as a horizontal-lifted vector field, $\tilde{\mathbf{h}}(\mathbf{x})$ is naturally left-invariant (see Eq. (C.2.2)). Together with the isometry of the group action on $\mathcal{M}$, this indicates that the right-hand-side of Eq. (D.3.5) can be simplified to:

$$- \int_{\mathcal{G}} \langle \tilde{\mathbf{h}}(\Phi_0^{\mathbf{x}}(g)), \mathbf{v}_{\Phi_0^{\mathbf{x}}(g)} \rangle_{\Phi_0^{\mathbf{x}}(g)} \mathrm{Vol}_0^{\mathbf{x}}(g)$$

$$= - \int_{\mathcal{G}} \langle (L_g)_{*\mathbf{x}} \tilde{\mathbf{h}}(\mathbf{x}), (L_g)_{*\mathbf{x}} \mathbf{v}_{\mathbf{x}} \rangle_{g \cdot \mathbf{x}} \mathrm{Vol}_0^{\mathbf{x}}(g) = - \int_{\mathcal{G}} \langle \tilde{\mathbf{h}}(\mathbf{x}), \mathbf{v}_{\mathbf{x}} \rangle_{\mathbf{x}} \mathrm{Vol}_0^{\mathbf{x}}(g)$$

$$= - \langle \tilde{\mathbf{h}}(\mathbf{x}), \mathbf{v}_{\mathbf{x}} \rangle_{\mathbf{x}} \int_{\mathcal{G}} \mathrm{Vol}_0^{\mathbf{x}}(g) = - V_0^{\mathbf{x}} \langle \tilde{\mathbf{h}}(\mathbf{x}), \mathbf{v}_{\mathbf{x}} \rangle_{\mathbf{x}}.$$

In this case, Eq. (D.3.5) can be simplified to $V_0^{\mathbf{x}} \partial_{t=0} \log \sqrt{\det(\mathbf{G}_t^{\mathbf{x}})} = - V_0^{\mathbf{x}} \langle \tilde{\mathbf{h}}(\mathbf{x}), \mathbf{v}_{\mathbf{x}} \rangle_{\mathbf{x}}$, which indicates:

$$\langle \tilde{\mathbf{h}}(\mathbf{x}), \mathbf{v}_{\mathbf{x}} \rangle_{\mathbf{x}} = -\frac{1}{2} \partial_{t=0} \log \det(\mathbf{G}_t^{\mathbf{x}}), \quad \forall \mathbf{x} \in \mathcal{M}.$$

To further simplify the expression, next we will prove the following conversion from the time derivative to spatial derivative, *i.e.*, $\partial_{t=0} \log \det(\mathbf{G}_t^{\mathbf{x}}) = \langle \nabla_{\mathbf{x}} \log \det(\mathbf{G}_0^{\mathbf{x}}), \mathbf{v}_{\mathbf{x}} \rangle_{\mathbf{x}}$, henceforth $\tilde{\mathbf{h}}(\mathbf{x}) = -\frac{1}{2} \nabla_{\mathbf{x}} \log \det(\mathbf{G}^{\mathbf{x}})$. Firstly, using Jacobi's formula in matrix calculus, we have $\partial_{t=0} \log \det(\mathbf{G}_t^{\mathbf{x}}) = \mathrm{tr}\big(\mathbf{G}_t^{\mathbf{x}-1} \partial_{t=0} \mathbf{G}_t^{\mathbf{x}}\big)$, and by Eq. (D.3.6), we have:

$$\partial_{t=0}(\mathbf{G}_t^{\mathbf{x}})_{ij} = \langle \partial_{t=0}(\Phi_t^{\mathbf{x}})_{*e} \mathbf{g}_i, (\Phi_0^{\mathbf{x}})_{*e} \mathbf{g}_j \rangle_{\mathbf{x}}^{\mathcal{M}} + \langle (\Phi_0^{\mathbf{x}})_{*e} \mathbf{g}_i, \partial_{t=0}(\Phi_t^{\mathbf{x}})_{*e} \mathbf{g}_j \rangle_{\mathbf{x}}^{\mathcal{M}}.$$

We next convert the time derivative $\partial_{t=0}(\Phi_t^{\mathbf{x}})_{*e} \mathbf{g}_i$ into spatial derivative by leveraging the left-invariance of $\mathbf{v}$ that defines $\Phi_t^{\mathbf{x}}$. Consider a general $g \in \mathcal{G}$ and $\mathbf{g} \in T_g \mathcal{G}$. From the definition of a tangent vector, for any test function $f$ around $\Phi_0^{\mathbf{x}}(g) = g \cdot \mathbf{x}$, we have:

$$\big((\Phi_t^{\mathbf{x}})_{*g} \mathbf{g}\big)(f) = \mathbf{g}\big(f(\Phi_t^{\mathbf{x}}(\cdot))|_g\big) = \mathbf{g}^i \partial_{g^i} f(\Phi_t^{\mathbf{x}}(\cdot))|_g$$

$$= \mathbf{g}^i (\partial_{\mathbf{x}^\alpha} f|_{g \cdot \mathbf{x}}) \partial_{g^i} \Phi_t^{\mathbf{x}}(g)^\alpha = \big(\mathbf{g}^i \partial_{g^i} \Phi_t^{\mathbf{x}}(g)^\alpha \partial_{\mathbf{x}^\alpha}\big)(f),$$

where we have used Einstein's summation convention where summations over indices repeated in the superscript and subscript are implied. Taking the derivative w.r.t $t$ and by leveraging Eq. (D.3.7), we have:

$$\partial_{t=0}(\Phi_t^{\mathbf{x}})_{*g} \mathbf{g} = \mathbf{g}^i \partial_{g^i} (\partial_{t=0} \Phi_t^{\mathbf{x}}(g)^\alpha) \partial_{\mathbf{x}^\alpha}$$

$$= \mathbf{g}^i \partial_{g^i} \mathbf{v}_{g \cdot \mathbf{x}}^\alpha \partial_{\mathbf{x}^\alpha} = \mathbf{g}^i \partial_{g^i} \big((L_g)_{*\mathbf{x}} \mathbf{v}_{\mathbf{x}}\big)^\alpha \partial_{\mathbf{x}^\alpha}$$

$$= \mathbf{g}^i \partial_{g^i} \frac{\partial L_g^\alpha}{\partial \mathbf{x}^\beta}\bigg|_{\mathbf{x}} \mathbf{v}_{\mathbf{x}}^\beta \partial_{\mathbf{x}^\alpha} = \mathbf{g}^i \partial_{\mathbf{x}^\beta} \frac{\partial L_g^\alpha}{\partial g^i}\bigg|_{\mathbf{x}} \mathbf{v}_{\mathbf{x}}^\beta \partial_{\mathbf{x}^\alpha}$$

$$= \mathbf{v}_{\mathbf{x}}^\beta \partial_{\mathbf{x}^\beta} \frac{\partial L_g^\alpha}{\partial g^i}\bigg|_{\mathbf{x}} \mathbf{g}^i \partial_{\mathbf{x}^\alpha} = \mathbf{v}_{\mathbf{x}}^\beta \partial_{\mathbf{x}^\beta} \big((\Phi_0^{\mathbf{x}})_{*g} \mathbf{g}\big)^\alpha \partial_{\mathbf{x}^\alpha},$$

which converts the time derivative to spatial derivatives by exchanging the order of differentiating $g \cdot \mathbf{x} = L_g(\mathbf{x}) = \Phi_0^{\mathbf{x}}(g)$ w.r.t $\mathbf{x}$ and $g$. By leveraging the linearity of inner product, we know $\langle \partial_{t=0}(\Phi_t^{\mathbf{x}})_{*e} \mathbf{g}_i, (\Phi_0^{\mathbf{x}})_{*e} \mathbf{g}_j \rangle_{\mathbf{x}}^{\mathcal{M}} = \langle \mathbf{v}_{\mathbf{x}}^\beta \partial_{\mathbf{x}^\beta}(\Phi_0^{\mathbf{x}})_{*e} \mathbf{g}_i, (\Phi_0^{\mathbf{x}})_{*e} \mathbf{g}_j \rangle_{\mathbf{x}}^{\mathcal{M}}$, so:

$$\partial_{t=0}(\mathbf{G}_t^{\mathbf{x}})_{ij} = \langle \mathbf{v}_{\mathbf{x}}^\beta \partial_{\mathbf{x}^\beta}(\Phi_0^{\mathbf{x}})_{*e} \mathbf{g}_i, (\Phi_0^{\mathbf{x}})_{*e} \mathbf{g}_j \rangle_{\mathbf{x}}^{\mathcal{M}} + \langle (\Phi_0^{\mathbf{x}})_{*e} \mathbf{g}_i, \mathbf{v}_{\mathbf{x}}^\beta \partial_{\mathbf{x}^\beta}(\Phi_0^{\mathbf{x}})_{*e} \mathbf{g}_j \rangle_{\mathbf{x}}^{\mathcal{M}}$$

$$= \mathbf{v}_{\mathbf{x}}^\beta \partial_{\mathbf{x}^\beta} \langle \Phi_{*e}^{\mathbf{x}} \mathbf{g}_i, \Phi_{*e}^{\mathbf{x}} \mathbf{g}_j \rangle_{\mathbf{x}} = \mathbf{v}_{\mathbf{x}}^\beta \partial_{\mathbf{x}^\beta}(\mathbf{G}_0^{\mathbf{x}})_{ij},$$

hence $\partial_{t=0} \log \det(\mathbf{G}_t^{\mathbf{x}}) = \mathrm{tr}\big(\mathbf{G}_0^{\mathbf{x}-1}\partial_{t=0}\mathbf{G}_t^{\mathbf{x}}\big) = \mathrm{tr}\big(\mathbf{G}_0^{\mathbf{x}-1}\mathbf{v}_{\mathbf{x}}^{\beta}\partial_{\mathbf{x}^{\beta}}\mathbf{G}_0^{\mathbf{x}}\big) = \mathbf{v}_{\mathbf{x}}^{\beta}\,\mathrm{tr}\big(\mathbf{G}_0^{\mathbf{x}-1}\partial_{\mathbf{x}^{\beta}}\mathbf{G}_0^{\mathbf{x}}\big) = \mathbf{v}_{\mathbf{x}}^{\beta}\partial_{\mathbf{x}^{\beta}} \log \det(\mathbf{G}_0^{\mathbf{x}}) = \langle \nabla_{\mathbf{x}} \log \det(\mathbf{G}_0^{\mathbf{x}}), \mathbf{v}_{\mathbf{x}} \rangle_{\mathbf{x}}$. Therefore, the mentioned expression indeed holds:

$$\tilde{\mathbf{h}}(\mathbf{x}) = -\frac{1}{2}\nabla_{\mathbf{x}} \log \det(\mathbf{G}^{\mathbf{x}}). \tag{D.3.8}$$

**(c)** Finally, we derive the expression for the specific case of $\mathcal{G} = \mathrm{SO}(3)$. Its embedding into $\mathcal{M}$ through $\mathbf{x} \in \mathcal{M}$ is given by $\Phi^{\mathbf{x}}(g) := g \cdot \mathbf{x} = g\mathbf{x}$, where in the last expression, $g$ is meant to be its $3 \times 3$ rotation-matrix form, and $g\mathbf{x} := [g\vec{x}^{(n)}]_n$ denotes a set of matrix-vector products. Its push-forward mapping is given explicitly as $\Phi_{*g}^{\mathbf{x}}\mathbf{g} = \mathbf{g}\mathbf{x}$, where $\mathbf{g} \in T_g\mathcal{G}$ also takes its $3 \times 3$ matrix form. As indicated by Eq. (C.3.1), $\{\mathbf{J}_1, \mathbf{J}_2, \mathbf{J}_3\}$ as defined in Eq. (C.3.2) forms a basis for $\mathfrak{so}(3)$, *i.e.*, the tangent space at $e = \mathbf{I}$, which also induces a basis $\{g\mathbf{J}_1, g\mathbf{J}_2, g\mathbf{J}_3\}$ for $T_g\mathcal{G}$, and this basis frame is left invariant, satisfying the construction above. Therefore, the asked basis frame $\{\mathbf{g}_i\}_i$ at $g$ can be constructed by $\mathbf{g}_i = \frac{1}{\sqrt{2}}g\mathbf{J}_i$. Then the Riemannian metric tensor $\mathbf{G}$ is given explicitly by:

$$\mathbf{G}_{ij}^{\mathbf{x}}(g) = \langle \mathbf{g}_i\mathbf{x}, \mathbf{g}_j\mathbf{x} \rangle_{g\mathbf{x}}^{\mathcal{M}} = \frac{1}{2}(g\mathbf{J}_i\mathbf{x})^{\top}(g\mathbf{J}_j\mathbf{x}) = \frac{1}{2}\mathbf{x}^{\top}\mathbf{J}_i^{\top}\mathbf{J}_j\mathbf{x} = \frac{1}{2}\sum_{n=1}^{N}\vec{x}^{(n)}\mathbf{J}_i^{\top}\mathbf{J}_j\vec{x}^{(n)}.$$

To proceed, one can easily verify that $\mathbf{J}_i^{\top}\mathbf{J}_j = \delta_{ij}\mathbf{I} - \vec{1}_j\vec{1}_i^{\top}$, where $\vec{1}_i$ denotes the 3-dimensional one-hot vector where 1 is placed at the $i$-th dimension. Leveraging this expression, we have:

$$\mathbf{G}^{\mathbf{x}}(g) = \frac{1}{2}\mathbf{K}(\mathbf{x}) =: \mathbf{G}^{\mathbf{x}},$$

where $\mathbf{K}(\mathbf{x}) := \sum_{n=1}^{N} \|\vec{x}^{(n)}\|^2\mathbf{I} - \sum_{n=1}^{N} \vec{x}^{(n)}\vec{x}^{(n)\top}$ is the same as defined in Eq. (D.3.3). Note that this expression indeed does not depend on $g$. From Eq. (D.3.8), we know:

$$\tilde{\mathbf{h}}(\mathbf{x}) = -\frac{1}{2}\nabla_{\mathbf{x}} \log \det \mathbf{G}^{\mathbf{x}} = -\frac{1}{2}\nabla_{\mathbf{x}} \log \det \mathbf{K}(\mathbf{x}). \tag{D.3.9}$$

Using Jacobi's formula $\mathrm{d}\log\det\mathbf{G} = \mathrm{tr}(\mathbf{K}^{-1}\mathrm{d}\mathbf{K})$ again, we have:

$$\mathbf{K}(\mathbf{x}) := \sum_{n=1}^{N} \|\vec{x}^{(n)}\|^2\mathbf{I} - \sum_{n=1}^{N} \vec{x}^{(n)}\vec{x}^{(n)\top}, \quad \frac{\partial\mathbf{K}}{\partial\vec{x}^{(n)i}} = 2\vec{x}^{(n)i}\mathbf{I} - (\vec{1}^i\vec{x}^{(n)\top} + \vec{x}^{(n)}\vec{1}^{i\top}),$$

where $\vec{1}^i \in \mathbb{R}^3$ is a one-hot vector at $i$. Therefore,

$$\mathrm{tr}\left(\mathbf{K}^{-1}\frac{\partial\mathbf{K}}{\partial\vec{x}^{(n)i}}\right) = 2\,\mathrm{tr}(\mathbf{K}^{-1})\vec{x}^{(n)i} - 2\vec{1}^{i\top}\mathbf{K}^{-1}\vec{x}^{(n)},$$

and finally we get the expression for the mean curvature vector field for $\mathrm{SO}(3)$:

$$\tilde{\mathbf{h}}(\mathbf{x}) = \left[-\frac{1}{2}\nabla_{\vec{x}^{(n)}} \log\det\mathbf{G}^{\mathbf{x}}\right]_n = \left[(\mathbf{K}(\mathbf{x})^{-1} - \mathrm{tr}(\mathbf{K}(\mathbf{x})^{-1})\mathbf{I})\vec{x}^{(n)}\right]_n.$$

$\square$

**Physical interpretations of the expressions.** As already assumed when defining the center of mass (CoM) of the system, the $N$ points in the system are treated as possessing the same mass, which simplifies calculation without hurting the utility to reduce the equivalent degrees of freedom. Under this assumption, the $\mathbf{K}(\mathbf{x})$ matrix defined in Eq. (D.3.2) is the *inertia tensor* of the $N$-point cloud as a rigid body. Given a total/rigid-body angular velocity $\vec{\omega}$, the angular momentum of the system is $\mathbf{K}(\mathbf{x})\vec{\omega}$. On the other hand, $\sum_{n'=1}^{N} \vec{x}^{(n')} \times \vec{v}^{(n')}$ is the total/rigid-body angular momentum of the system by definition (no matter whether $[\vec{v}^{(n')}]_{n'}$ deforms the point cloud / molecule). So the angular momentum is $\vec{\omega} = \mathbf{K}(\mathbf{x})^{-1}\big(\sum_{n'=1}^{N} \vec{x}^{(n')} \times \vec{v}^{(n')}\big)$. The linear velocity it induces on each point, altogether only contributing a rigid-body rotation, is $[\vec{\omega} \times \vec{x}^{(n)}]_n$. Therefore, subtracting this velocity from the original one, which gives the expression of the horizontal projection operator in Eq. (10), leaves a linear velocity that only deforms the point cloud / molecule, without any rigid-body rotation. This is in analogy with the treatment for the $\mathbb{T}(3)$ translational group where the total/rigid-body (linear) momentum is removed.

As for the lifted mean curvature vector field in Eq. (D.3.1), it can be reformulated as $\tilde{\mathbf{h}}(\mathbf{x}) = -\frac{1}{2}\nabla_{\mathbf{x}} \log\det \mathbf{K}(\mathbf{x})$ (as also shown in the proof, Eqs. (D.3.8, D.3.9) ). To understand this, recall

that on the quotient space, rotationally equivalent structures are compressed as one point, neglecting the volume of the equivalence class which differs with the shape. This also induces a neglect of the volume of angular momentum in the original space, which also differs with the shape. Indeed, as the angular momentum induced by an angular velocity $\vec{\omega}$ on a point cloud in state $\mathbf{x}$ is $\vec{L} = \mathbf{K}(\mathbf{x})\vec{\omega}$, where $\mathbf{K}(\mathbf{x})$ is the inertia tensor of the point cloud. Therefore, although the volume of the angular velocity $\omega$ up to a given scale is independent of the configuration of the point cloud, the volume of the angular momentum $\vec{L}$ does: $\mathrm{d}\vec{L} = (\det \mathbf{K}(\mathbf{x})) \, \mathrm{d}\vec{\omega}$ ($\mathrm{d}\vec{L}$ and $\mathrm{d}\vec{\omega}$ here represent the respective volume form; *i.e.*, infinitesimal volume element). So configurations with a larger $\det \mathbf{K}(\mathbf{x})$ occupies a larger portion of the original space, so in the purely deformation process, configurations with a larger $\det \mathbf{K}(\mathbf{x})$ would be easier to be hit than it actually should be. Therefore, the $\tilde{\mathbf{h}}(\mathbf{x})$ vector field comes to counteract this effect, by adding a steering force towards configurations with a smaller $\det \mathbf{K}(\mathbf{x})$. Only with this correction can the generation process recover the same target distribution.

Finally, we verify that this $\tilde{\mathbf{h}}(\mathbf{x})$ vector field is indeed horizontal. From Eq. (C.3.4), this amounts to showing $\sum_{n=1}^{N} \vec{x}^{(n)} \times \tilde{\mathbf{h}}^{(n)}(\mathbf{x}) = \vec{0}$. Since:

$$\sum_{n=1}^{N} \vec{x}^{(n)} \times \tilde{\mathbf{h}}^{(n)}(\mathbf{x}) = \sum_{n=1}^{N} \vec{x}^{(n)} \times \left( \mathbf{K}(\mathbf{x})^{-1} \vec{x}^{(n)} - \mathrm{tr}(\mathbf{K}(\mathbf{x})^{-1}) \vec{x}^{(n)} \right) = \sum_{n=1}^{N} \vec{x}^{(n)} \times \left( \mathbf{K}(\mathbf{x})^{-1} \vec{x}^{(n)} \right)$$

$$= \sum_{n=1}^{N} \frac{1}{\det \mathbf{K}(\mathbf{x})} \mathbf{K}(\mathbf{x}) \left( \left( \mathbf{K}(\mathbf{x}) \vec{x}^{(n)} \right) \times \vec{x}^{(n)} \right),$$

we only need to verify $\sum_{n=1}^{N} \left( \mathbf{K}(\mathbf{x}) \vec{x}^{(n)} \right) \times \vec{x}^{(n)} = \vec{0}$. Leveraging the expression for $\mathbf{K}(\mathbf{x})$ in Eq. (D.3.2), we have: $\sum_{n=1}^{N} \left( \mathbf{K}(\mathbf{x}) \vec{x}^{(n)} \right) \times \vec{x}^{(n)}$

$$= \sum_{n=1}^{N} \left( \sum_{n'=1}^{N} \|\vec{x}^{(n')}\|^2 \right) \vec{x}^{(n)} \times \vec{x}^{(n)} - \sum_{n=1}^{N} \left( \left( \sum_{n'=1}^{N} \vec{x}^{(n')} \vec{x}^{(n')\top} \right) \vec{x}^{(n)} \right) \times \vec{x}^{(n)}$$

$$= -\sum_{n=1}^{N} \left( \sum_{n'=1}^{N} \left( \vec{x}^{(n')\top} \vec{x}^{(n)} \right) \vec{x}^{(n')} \right) \times \vec{x}^{(n)} = -\sum_{n=1}^{N} \sum_{n'=1}^{N} \left( \vec{x}^{(n')\top} \vec{x}^{(n)} \right) \left( \vec{x}^{(n')} \times \vec{x}^{(n)} \right).$$

Noting that $\vec{x}^{(n')\top} \vec{x}^{(n)} = \vec{x}^{(n)\top} \vec{x}^{(n')}$ while $\vec{x}^{(n')} \times \vec{x}^{(n)} = -\vec{x}^{(n)} \times \vec{x}^{(n')}$, we can couple the $(n, n')$ pair with the $(n', n)$ pair in the summation, which all cancels out. Therefore, the result is indeed zero.

## E  TRAINING AND SAMPLING METHODS IN MORE GENERAL CASES

**Training objective.** The diffusion model on the total space $\mathcal{M}$ is trained by the denoising score matching objective. Since the vertical components of the velocity are not strictly needed, we propose to supervise the model only on the horizontal components and allow arbitrary vertical output of the model. Recall that the horizontal projection operator $P_{\mathbf{x}}$ projects a vector to its horizontal component, *i.e.*, $P_{\mathbf{x}}(\mathbf{v}) = \mathbf{v}^{\mathcal{H}}$. Thus the improved training objective is given by

$$\mathcal{L}(\theta) := \mathbb{E}_{p(t)} w(t) \mathbb{E}_{(\mathbf{x}_0, \mathbf{x}_1) \sim p_{\mathrm{joint}}, \boldsymbol{\epsilon} \sim \mathcal{N}(0, \mathbf{I})} \left\| P_{\mathbf{x}_t} \left( \mathbf{v}_\theta(\mathbf{x}_t, t) - (\dot{\alpha}_t \mathbf{x}_0 + \dot{\beta}_t \mathbf{x}_1 + \dot{\gamma}_t \boldsymbol{\epsilon}) \right) \right\|^2.$$

**ODE sampler.** After the training stage, $P_{\mathbf{x}_t}(\mathbf{v}_\theta(\mathbf{x}_t, t))$ is an approximation of the ground truth vector field in the horizontal subspace. For the deterministic sampler, we need to simulate the horizontal lift of the projected ODE, which is given by

$$\frac{\mathrm{d}\mathbf{x}_t}{\mathrm{d}t} = P_{\mathbf{x}_t}(\mathbf{v}(\mathbf{x}_t, t)).$$

In practice, the ODE process is approximated by numerical solvers, *e.g.* the Euler method and Runge-Kutta methods.

**SDE sampler.** For the stochastic sampler, we need to simulate the horizontal lift of the projected original SDE in Eq. (3). According to Thm. 1 and Thm. 4, the lifted process is given by

$$\mathrm{d}\mathbf{x}_t = P_{\mathbf{x}_t} \left( \mathbf{v}_\theta(\mathbf{x}_t, t) + \eta_t \mathbf{s}_\theta(\mathbf{x}_t, t) \right) \mathrm{d}t + \eta_t \tilde{\mathbf{h}}(\mathbf{x}_t) \, \mathrm{d}t + \sqrt{2\eta_t} P_{\mathbf{x}_t} \, \mathrm{d}\mathbf{w}_t,$$

The training and sampling algorithm is summarized in Algorithm 2 and 3.

---

**Algorithm 1** Training for $p_{\text{prior}} = \mathcal{N}(0, \mathbf{I})$

1: **repeat**
2: $\quad (\mathbf{x}_0, \mathbf{x}_1) \sim p_{\text{joint}}$
3: $\quad t \sim p_t$
4: $\quad \mathbf{x}_t = \hat{\alpha}_t \mathbf{x}_0 + \beta_t \mathbf{x}_1$
5: $\quad$ Take a gradient descent step on
$\qquad \nabla_\theta \, w(t) \, \|P_{\mathbf{x}_t} \left(\mathbf{D}_\theta(\mathbf{x}_t, t) - \mathbf{x}_1\right)\|^2$
6: **until** converged

---

**Algorithm 2** Training for general $p_{\text{prior}}$

1: **repeat**
2: $\quad (\mathbf{x}_0, \mathbf{x}_1) \sim p_{\text{joint}}, \; \boldsymbol{\epsilon} \sim \mathcal{N}(\mathbf{0}, \mathbf{I})$
3: $\quad t \sim p_t$
4: $\quad \mathbf{x}_t = \alpha_t \mathbf{x}_0 + \beta_t \mathbf{x}_1 + \gamma_t \boldsymbol{\epsilon}$
5: $\quad \mathbf{v}_t = \dot{\alpha}_t \mathbf{x}_0 + \dot{\beta}_t \mathbf{x}_1 + \dot{\gamma}_t \boldsymbol{\epsilon}$
6: $\quad$ Take a gradient descent step on
$\qquad \nabla_\theta \, w(t) \, \|P_{\mathbf{x}_t} \left(\mathbf{v}_\theta(\mathbf{x}_t, t) - \mathbf{v}_t\right)\|^2$
7: **until** converged

---

**Algorithm 3** Sampling

1: $\mathbf{x}_0 \sim p_{\text{prior}}$
2: **for** $i = 0$ **to** $K - 1$ **do**
3: $\quad \Delta t_i = t_{i+1} - t_i$
4: $\quad$ **if** ODE sampling **then**
5: $\qquad \mathbf{x}_{t_{i+1}} = \mathbf{x}_{t_i} + P_{\mathbf{x}_{t_i}} \mathbf{v}_\theta(\mathbf{x}_{t_i}, t_i) \, \Delta t_i$
6: $\quad$ **end if**
7: $\quad$ **if** SDE sampling **then**
8: $\qquad \mathbf{d}_i = P_{\mathbf{x}_{t_i}} \left(\mathbf{v}_\theta(\mathbf{x}_{t_i}, t_i) + \eta_{t_i} \mathbf{s}_\theta(\mathbf{x}_{t_i}, t_i)\right) + \eta_{t_i} \tilde{\mathbf{h}}(\mathbf{x}_{t_i})$
9: $\qquad \boldsymbol{\epsilon} \sim \mathcal{N}(\mathbf{0}, \mathbf{I})$
10: $\qquad \mathbf{x}_{t_{i+1}} = \mathbf{x}_{t_i} + \mathbf{d}_i \Delta t_i + \sqrt{2 \eta_{t_i} \Delta t_i} P_{\mathbf{x}_{t_i}} \boldsymbol{\epsilon}$
11: $\quad$ **end if**
12: **end for**

---

# F EXPERIMENTAL DETAILS

## F.1 MOLECULAR STRUCTURE GENERATION

This appendix summarizes our experimental setup, which strictly follows that of ET-Flow (Hassan et al., 2024). We detail the datasets, model architecture, training, sampling, and evaluation. For a more comprehensive discussion of each component, we refer the reader to the appendices of their original paper.

### F.1.1 DATASET

First, we evaluate our framework on the molecule structure generation task. In this scenario, our goal is to generate the 3D coordinates of a molecule given the graph structure of the molecule. We conduct the experiments on the GEOM datasets (Axelrod & Gomez-Bombarelli, 2022), which provide structure ensembles generated by metadynamics in CREST (Pracht et al., 2024), and we focus on the GEOM-QM9 and GEOM-DRUGS datasets. Following the data processing and splits from Hassan et al. (2024), we use the random splits with train/validation/test of 243473/30433/1000 for GEOM-DRUGS and 106586/13323/1000 for GEOM-QM9. In addition, data with disconnected molecule graphs are removed for GEOM-DRUGS (Hassan et al., 2024). Our reproduction is based on the modified data-processing pipeline following the released configurations thus different from the results reported in the original paper. In evaluation, we use the same sampling steps of 50 NFEs for all ET-Flow series models in Table 2 for fair comparison.

### F.1.2 SETTINGS

We primarily follow the setting in Hassan et al. (2024). We set the Gaussian distribution as the prior distribution on GEOM-QM9 and use the harmonic prior for GEOM-DRUGS (Volk et al., 2023). Following Jing et al. (2022); Xu et al. (2022), we report the RMSD-based metrics, *e.g.*, Coverage and Average Minimum RMSD (AMR) between generated and ground truth structure ensembles. We parameterize $\mathbf{v}_\theta$ by using equivariant graph transformer architectures from ET-Flow (Hassan et al., 2024), including the O(3) and SO(3) equivariant variants, which also serves as a verification that

our framework is compatible with different backbone models. For training, we use AdamW as the optimizer, and set the hyper-parameter $\epsilon$ to 1e-8 and $(\beta_1, \beta_2)$ to (0.9,0.999). We use the dynamic gradient clipping as (Hassan et al., 2024; Hoogeboom et al., 2022b). The peak learning rate is set to 5e-4 for GEOM-DRUGS and 7e-4 for GEOM-QM9. The batch size is set to 48 for GEOM-DRUGS and 128 for GEOM-QM9. The weight decay is set to 1e-8. The model is trained for 1000 epochs for both datasets. The noise scale $\sigma$ is set to 0.1. We also use 50 time steps with the Euler solver for sampling. All models are trained on 8 NVIDIA A100 GPUs.

### F.1.3 BASELINES

Following (Hassan et al., 2024), we choose strong baselines trained on GEOM-DRUGS and GEOM-QM9 for a challenging comparison. We report the performance of CGCF (Xu et al., 2021), GeoMol (Ganea et al., 2021), GeoDiff (Xu et al., 2022), Torsional Diffusion (Jing et al., 2022), and MCF (Wang et al., 2023).

## F.2 PROTEIN STRUCTURE GENERATION

This appendix summarizes our experimental setup, which strictly follows that of Proteína (Geffner et al., 2025). We detail the datasets, model architecture, training, sampling, and evaluation. For a more comprehensive discussion of each component, we refer the reader to the appendices of their original paper.

### F.2.1 DATASET

For training, we utilize the FoldSeek AFDB clusters ($\mathcal{D}_{FS}$) dataset as curated and described in the Proteína. This dataset is a high-quality, non-redundant subset of the AlphaFold Database (AFDB), containing 588,318 cluster-representative protein structures with lengths between 32 and 256 residues. The dataset is annotated with hierarchical CATH labels, which are leveraged during training. Our data processing and handling strictly follow the pipeline detailed in Appx. M of Geffner et al. (2025).

### F.2.2 BASELINES

The representative references in Table 3 primarily focus on generating the full backbone of proteins, whereas the Proteína setting evaluated here focuses only on generating the alpha carbon atoms ($C_\alpha$) of the residues. These baselines can be categorized according to their structure-generation strategies. FrameDiff (Yim et al., 2023b) and RFDiffusion (Watson et al., 2023) are diffusion-based backbone generators built on residue-frame representations. FrameDiff (Yim et al., 2023b) trains a standalone SE(3)-aware denoising model for unconditional backbone generation, whereas RFDiffusion (Watson et al., 2023) adapts the RoseTTAFold structure prediction architecture into a denoising diffusion model and is particularly effective for conditional design and motif scaffolding. FoldFlow (Bose et al., 2024) and FrameFlow (Yim et al., 2023a) retain a similar frame-based geometric representation but replace the score-based diffusion with flow matching, with the goal of achieving more stable dynamics and faster sampling. FoldFlow (Bose et al., 2024) further introduces deterministic, optimal-transport, and stochastic variants. Proteus (Wang et al., 2024) also falls within the backbone-frame diffusion family, but improves the denoising architecture with graph triangle blocks and multi-track interactions, enhancing designability and sampling efficiency without relying on large pretrained structure-prediction models. Chroma (Ingraham et al., 2023) adopts a different geometry-aware diffusion strategy, generating protein backbones through correlated coordinate diffusion and subsequently completing sequence and side-chain design with its design network. Genie2 (Lin et al., 2024b) is closer to the Proteína setting in terms of its forward process, as it adds Gaussian noise only to the $C_\alpha$ coordinates, while still adopting the residue frame representation and SE(3)-equivariant modules. ESM3 (Hayes et al., 2025) is methodologically distinct from these structure-centric baselines: it is a multimodal protein language model operating over sequence, structure, and function tokens for broad conditional protein generation. In Table 3, results of these baselines under the metrics are taken directly from the Proteína paper (Geffner et al., 2025).

### F.2.3 MODEL ARCHITECTURE AND TRAINING

We adopt the efficient $\mathcal{M}_{FS}^{small}$ architecture from the original Proteína work (Geffner et al., 2025), which is a 60M-parameter non-equivariant Transformer-based architecture that forgoes the use of computationally expensive triangle update layers and pair representation updates. Key aspects of the

training protocol from Proteína are preserved, including their novel Beta-Uniform mixture for the time-sampling distribution $p(t)$, the use of self-conditioning, and data augmentation with random rotations. All model and training hyperparameters, such as embedding dimensions, number of layers, attention heads, and optimizer settings, are kept consistent with hyperparameters saved in their released checkpoint $\mathcal{M}_{\text{FS}}^{\text{small}}$. The hyperparameters for the $\mathcal{M}_{\text{FS}}^{\text{small}}$ model are detailed in Table F.2.1, in comparison with the larger models from the original Proteína paper.

Table F.2.1: Hyperparameters for Proteína model.

| Hyperparameter | $\mathcal{M}_{\text{FS}}$ | $\mathcal{M}_{\text{FS}}^{\text{no-tri}}$ | $\mathcal{M}_{\text{FS}}^{\text{small}}$ |
|---|---|---|---|
| **Proteína Architecture** | | | |
| sequence repr dim | 768 | 768 | 512 |
| # registers | 10 | 10 | 10 |
| sequence cond dim | 512 | 512 | 128 |
| $t$ sinusoidal enc dim | 256 | 256 | 196 |
| idx. sinusoidal enc dim | 128 | 128 | 196 |
| fold emb dim | 256 | 256 | 196 |
| pair repr dim | 512 | 512 | 196 |
| seq separation dim | 128 | 128 | 128 |
| pair distances dim ($x_t$) | 64 | 64 | 64 |
| pair distances dim ($\tilde{x}(x_t)$) | 128 | 128 | 128 |
| pair distances min (Å) | 1 | 1 | 1 |
| pair distances max (Å) | 30 | 30 | 30 |
| # attention heads | 12 | 12 | 12 |
| # tranformer layers | 15 | 15 | 12 |
| # triangle layers | 5 | — | — |
| # trainable parameters | 200M | 200M | 60M |
| **Proteína Training** | | | |
| # steps | 200K | 360K | 150K |
| batch size per GPU | 4 | 10 | 5 |
| # GPUs | 128 | 96 | 16 |
| # grad. acc. steps | 1 | 1 | 1 |

### F.2.4   SAMPLING

To facilitate a direct comparison with the publicly available Proteína checkpoints, we trained our model with an identical hierarchical fold class conditioning mechanism. However, to ensure a fair assessment of foundational generative capabilities, all experiments reported in our main text were performed in a strictly unconditional setting. We applied the same sampling protocol across all models, using 400 sampling steps and enabling self-conditioning, which consistently improved performance. No other guidance techniques, such as autoguidance, were utilized. We use deterministic ODE sampling to assess distributional fidelity and SDE sampling to explore the designability-diversity trade-off. We adapt the SDE formulation and its Euler-Maruyama numerical scheme, detailed in Appx. I of Geffner et al. (2025), for our quotient space framework, while retaining all other configurations, such as the sampling scheduler and $g(t)$, from the original paper.

### F.2.5   EVALUATION

We evaluate our models rigorously adheres to the metrics established and validated in the Proteína paper. We assess model performance using the standard suite of metrics in protein backbone structure design:

- **Designability.** Quantified by the self-consistency RMSD (scRMSD) protocol, using ProteinMPNN for inverse folding and ESMFold for structure prediction, with a success threshold of scRMSD less than 2 Å.

- **Diversity.** Measured in two ways: by the average pairwise TM-score among designable samples, and by the number of distinct structural clusters identified by FoldSeek at a TM-score threshold of 0.5.

- **Novelty.** Assessed by calculating the maximum TM-score of each designable sample against reference structures in the PDB and AFDB databases.

We also adopt the novel probabilistic metrics introduced by Geffner et al. (2025), to measure how well our model captures the true distribution of protein structures:

- **FPSD.** Measured the distributional similarity between generated and reference structures in the feature space of a pre-trained fold class predictor.

- **fS.** Evaluated both the quality and diversity of samples based on the confidence and entropy of fold class predictions.

- **fJSD.** Quantified the similarity between the categorical fold class distributions of generated and reference sets.

It is noteworthy that we have omitted the Diversity and Novelty metrics from our main text to avoid comparisons with potentially inaccurate results in the literature. This decision is based on a bug recently identified in the alntmscore output of FoldSeek versions prior to v10 (release 10-941cd33), which renders many previously reported TM-based metrics incorrect (also found in (Daras et al., 2025)). To provide a controlled and accurate benchmark, we conducted our own analysis using the FoldSeek v10 (release 10-941cd33). We limited this re-evaluation to the released small Proteína model and our corresponding model trained in the quotient space. The full results of this comparison are summarized in Table F.2.2.

Table F.2.2: Complete performance comparison of the released Proteína checkpoints against models using quotient-space diffusion. Best results are marked in **bold**.

| Model | Designability (%) ↑ | Diversity | | Novelty ↓ vs. | | FPSD ↓ vs. | | fS ↑ in | fJSD ↓ vs. | |
|---|---|---|---|---|---|---|---|---|---|---|
| | | Cluster ↑ | TM-Sc. ↓ | PDB | AFDB | PDB | AFDB | C/A/T | PDB | AFDB |
| **SDE Sampling** | | | | | | | | | | |
| Proteína $\mathcal{M}_{\mathrm{FS}}^{\mathrm{small}}, \gamma = 0.35$ | 96.0 | 0.44 (209) | 0.50 | 0.86 | 0.91 | 386.5 | 378.2 | 1.77/4.97/17.78 | 2.17 | 1.73 |
| **+ Quotient-space diffusion** | **97.6** | 0.40 (197) | 0.48 | 0.86 | 0.91 | 274.7 | 277.1 | 2.24/6.69/20.99 | 1.68 | 1.55 |
| Proteína $\mathcal{M}_{\mathrm{FS}}^{\mathrm{small}}, \gamma = 0.45$ | 92.2 | 0.55 (253) | 0.49 | 0.84 | 0.90 | 332.9 | 320.4 | 1.83/5.01/20.22 | 1.93 | 1.49 |
| **+ Quotient-space diffusion** | 92.6 | 0.51 (253) | 0.47 | 0.85 | 0.90 | 244.5 | 246.3 | 2.24/6.68/23.47 | 1.43 | 1.28 |
| Proteína $\mathcal{M}_{\mathrm{FS}}^{\mathrm{small}}, \gamma = 0.50$ | 89.2 | 0.57 (255) | 0.48 | 0.83 | 0.89 | 306.2 | 290.8 | 1.86/4.92/21.15 | 1.81 | 1.36 |
| **+ Quotient-space diffusion** | 90.2 | 0.51 (231) | 0.47 | 0.84 | 0.90 | 228.0 | 228.7 | 2.25/6.59/25.24 | 1.32 | 1.17 |
| **ODE Sampling** | | | | | | | | | | |
| Proteína $\mathcal{M}_{\mathrm{FS}}^{\mathrm{small}}$ | 13.8 | **0.90 (62)** | **0.43** | **0.80** | 0.87 | 83.18 | 21.93 | 2.45/5.63/31.76 | 0.58 | 0.12 |
| **+ Quotient-space diffusion** | 15.6 | 0.87 (68) | **0.43** | **0.80** | 0.86 | 69.94 | 17.56 | **2.57/6.40/32.14** | 0.41 | 0.11 |

# G  ADDITIONAL EXPERIMENTAL RESULTS AND DISCUSSIONS

## G.1  EFFICIENCY AND COMPLEXITY ANALYSIS

**Complexity analysis.** In this subsection, we give a detailed discussion on the computational cost of our method. As mentioned in Thm. 4, we need to compute the inversion of the matrix $\mathbf{K}$ and the cross product for the horizontal projection operator $P_{\mathbf{x}}$ and the mean curvature vector $\tilde{\mathbf{h}}(\mathbf{x})$. For the calculation of $\mathbf{K}^{-1}$, notice that $\mathbf{K}$ is always a $3 \times 3$ matrix, so construction cost of $\mathbf{K}^{-1}$ is only linear $O(N)$, where $N$ is the number of atoms (linear $O(N)$ cost for constructing $\mathbf{K}$, and constant $O(1)$ cost for inversion). The cross product is conducted atom-wise, so its computational cost is also linear $O(N)$. So we can conclude that the overall computational complexity is $O(N)$ for both $P_{\mathbf{x}}$ and $\tilde{\mathbf{h}}(\mathbf{x})$.

We would like to mention that the alignment operation adopted in the heuristic alignment-based diffusion strategies also has the same complexity. To see this, for aligning $\mathbf{x} \in \mathbb{R}^{3 \times N}$ towards $\mathbf{y} \in \mathbb{R}^{3 \times N}$, the Kabsch-Umeyama algorithm constructs the optimal rotation matrix as $(\mathbf{H}^{\top}\mathbf{H})^{\frac{1}{2}}\mathbf{H}^{-1}$, where $\mathbf{H} := \mathbf{y}\mathbf{x}^{\top} \in \mathbb{R}^{3 \times 3}$ requires a linear $O(N)$ cost. In practice, the $O(N)$ computational cost is negligible compared to the cost of gradient back-propagation through the neural network. A comparison of practical training times is shown in the following table.

All the results are tested on a single Nvidia A100 GPU. From the results, we can see that the additional computational cost brought by the alignment and projection is negligible.

Table G.1.1: Training speed comparison on QM9.

| Methods | Original diffusion | GeoDiff alignment | AF3 alignment | Quotient-space diffusion |
|---|---|---|---|---|
| training speed (iters/s) | 4.19 | 4.07 | 4.08 | 4.10 |

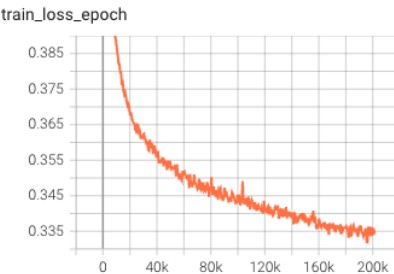

Figure G.1.1: Training loss vs. training epochs. We find that our training is stable in practice.

**Numerical stability.** In our quotient-space diffusion model framework, we need to calculate the matrix inversion of $\mathbf{K}$, which may have numerical issues for near-collinear systems of points. In practice, we add an $\epsilon\mathbf{I}$ term before conducting matrix inversion, that is, we calculate $(\epsilon\mathbf{I} + \mathbf{K})^{-1}$ in practice, where $\mathbf{I}$ is the $3 \times 3$ identity matrix. This treatment is widely adopted in algorithms facing similar situations, *e.g.*, the practical implementation of the Kabsch-Umeyama algorithm for alignment. Our typical choice of $\epsilon$ is 1e-8, and we found that the training process is stable under this setting. We have shown the training curve of the model on the protein structure generation task in Fig. G.1.1, which indicates no numerical issues arise during the training process.

### G.2 THE IMPLEMENTATION OF $\mathcal{G}$-EQUIVARIANT VECTOR FIELD

In Thm. 4, we require that the vector field is $\mathrm{SO}(3)$-equivariant. In practice, this can be implemented by using a $\mathrm{SO}(3)$-equivariant network architecture or applying data augmentation. In this subsection, we justify that both of these choices are valid, such that the diffusion model can generate a $\mathrm{SO}(3)$-invariant distribution.

**Diffusion model with data augmentation.** The optimal solution of the Euclidean diffusion model is given by $\mathbf{D}_\theta^*(\mathbf{x}_t) = \mathbb{E}[\mathbf{x}_1|\mathbf{x}_t]$ (Song et al., 2021; Karras et al., 2022). When the data distribution is augmented by random rotation, the data distribution becomes $\mathrm{SO}(3)$-invariant. Thus, the optimal diffusion model can recover the $\mathrm{SO}(3)$-invariant data distribution. When the transition density $p(\mathbf{x}_t|\mathbf{x}_1)$ is $\mathrm{SO}(3)$-equivariant, *i.e.* $p(\mathbf{x}_t|\mathbf{x}_1) = p(g \cdot \mathbf{x}_t|g \cdot \mathbf{x}_1), \forall g \in \mathrm{SO}(3)$, the optimal network is $\mathrm{SO}(3)$-equivariant. To see this, let $g \in \mathrm{SO}(3)$ be an arbitrary rotation matrix. Since $\mathbf{D}_\theta^*(g \cdot \mathbf{x}_t) = \mathbb{E}[\mathbf{x}_1|g \cdot \mathbf{x}_t]$, by the Bayes formula,

$$\mathbb{E}[\mathbf{x}_1|g \cdot \mathbf{x}_t] = \frac{\mathbb{E}_{p_{\text{target}}(\mathbf{x}_1)}[\mathbf{x}_1 p(g \cdot \mathbf{x}_t|\mathbf{x}_1)]}{\mathbb{E}_{p_{\text{target}}(\mathbf{x}_1)}[p(g \cdot \mathbf{x}_t|\mathbf{x}_1)]} = \frac{\mathbb{E}_{p_{\text{target}}(\mathbf{x}_1)}[\mathbf{x}_1 p(\mathbf{x}_t|g^{-1} \cdot \mathbf{x}_1)]}{\mathbb{E}_{p_{\text{target}}(\mathbf{x}_1)}[p(\mathbf{x}_t|g^{-1}\mathbf{x}_1)]}$$

$$= \frac{g \cdot \mathbb{E}_{p_{\text{target}}(g^{-1}\mathbf{x}_1)}[g^{-1}\mathbf{x}_1 p(\mathbf{x}_t|g^{-1} \cdot \mathbf{x}_1)]}{\mathbb{E}_{p_{\text{target}}(g^{-1}\mathbf{x}_1)}[p(\mathbf{x}_t|g^{-1}\mathbf{x}_1)]} = g \cdot \mathbb{E}[\mathbf{x}_1|\mathbf{x}_t],$$

where we use the equivariance property of the transition density to get the second equality and the invariance property of $p_{\text{target}}$ to get the third equality. Thus, we can conclude that the optimal solution under these conditions is $\mathrm{SO}(3)$-equivariant. Geffner et al. (2025) also gives an empirical validation that a well-trained neural network becomes nearly equivariant even if its architecture is not equivariant.

**Equivariant architecture.** When the model is required to be $\mathrm{SO}(3)$-equivariant, the optimal solution of the diffusion model is not $\mathbb{E}[\mathbf{x}_1|\mathbf{x}_t]$. To figure out the optimal solution, we consider the training loss at time $t$. The loss function at $t$ is given by

$$\mathcal{L}_t(\theta) = \mathbb{E}\|\mathbf{D}_\theta(\mathbf{x}_t, t) - \mathbf{x}_1\|^2$$

$$= \int \mathrm{d}^{3N}\mathbf{x}_1 \int \mathrm{d}^{3N}\mathbf{x}_t \, p(\mathbf{x}_1, \mathbf{x}_t) \left( \|\mathbf{D}_\theta(\mathbf{x}_t, t)\|^2 + \|\mathbf{x}_1\|^2 - 2\langle \mathbf{D}_\theta(\mathbf{x}_t, t), \mathbf{x}_1 \rangle \right).$$

The optimal solution satisfies

$$\mathbf{D}_\theta^*(\mathbf{x}_t, t) = \operatorname*{argmin}_{\mathbf{D}_\theta \text{ is SO(3)-equivariant}} \mathcal{L}_t(\theta).$$

The training loss can be simplified using the equivariant constraint: $\mathcal{L}_t(\theta)$

$$= \int \mathrm{d}^{3N}\mathbf{x}_1 \int \mathrm{d}^{3N}\mathbf{x}_t \, p(\mathbf{x}_1, \mathbf{x}_t) \left( \|\mathbf{D}_\theta(\mathbf{x}_t)\|^2 + \|\mathbf{x}_1\|^2 - 2\langle \mathbf{D}_\theta(\mathbf{x}_t), \mathbf{x}_1 \rangle \right)$$

$$= \int_{\mathbb{R}^{3N}_{\mathrm{CoM}\circ}/\mathrm{SO}(3)} \mathrm{d}\mathbf{r}_t \int_{\mathrm{SO}(3)} \mathrm{d}g \int \mathrm{d}^{3N}\mathbf{x}_1 \, p(\mathbf{x}_1, g \cdot \mathbf{r}_t) \left( \|\mathbf{D}_\theta(g \cdot \mathbf{r}_t)\|^2 + \|\mathbf{x}_1\|^2 - 2\langle \mathbf{D}_\theta(g \cdot \mathbf{r}_t), \mathbf{x}_1 \rangle \right),$$

where $\mathbb{R}^{3N}_{\mathrm{CoM}\circ}$ is defined in Appx. C.3, and $\mathbf{r}_t \in \mathbb{R}^{3N}$ is a representation of a quotient-space element in the original Euclidean space. Since $\mathbf{D}_\theta$ is SO(3)-equivariant, $\mathbf{D}_\theta(g \cdot \mathbf{r}_t) = g \cdot \mathbf{D}_\theta(\mathbf{r}_t)$, then we have: $\mathcal{L}_t(\theta)$

$$= \int_{\mathbb{R}^{3N}_{\mathrm{CoM}\circ}/\mathrm{SO}(3)} \mathrm{d}\mathbf{r}_t \int_{\mathrm{SO}(3)} \mathrm{d}g \int \mathrm{d}^{3N}\mathbf{x}_1 \, p(\mathbf{x}_1, g \cdot \mathbf{r}_t) \left( \|\mathbf{D}_\theta(\mathbf{r}_t)\|^2 + \|\mathbf{x}_1\|^2 - 2\langle g \cdot \mathbf{D}_{1\theta}(\mathbf{r}_t), \mathbf{x}_1 \rangle \right).$$

Define $p(\mathbf{r}_t) = \int_{\mathrm{SO}(3)} \mathrm{d}g \int \mathrm{d}^{3N}\mathbf{x}_1 \, p(\mathbf{x}_1, g \cdot \mathbf{r}_t)$, and $p(\mathbf{x}_1, g \mid \mathbf{r}_t) = \frac{p(\mathbf{x}_1, g \cdot \mathbf{r}_t)}{p(\mathbf{r}_t)}$. Then we have: $\mathcal{L}_t(\theta)$

$$= \int_{\mathbb{R}^{3N}_{\mathrm{CoM}\circ}/\mathrm{SO}(3)} \mathrm{d}\mathbf{r}_t \left[ p(\mathbf{r}_t) \|\mathbf{D}_\theta(\mathbf{r}_t)\|^2 - 2\langle \mathbf{D}_\theta(\mathbf{r}_t), \int_{\mathrm{SO}(3)} \mathrm{d}g \int \mathrm{d}^{3N}\mathbf{x}_1 \, p(\mathbf{x}_1, g \cdot \mathbf{r}_t) g^{-1} \cdot \mathbf{x}_1 \rangle \right]$$

$$+ \int_{\mathbb{R}^{3N}_{\mathrm{CoM}\circ}/\mathrm{SO}(3)} \mathrm{d}\mathbf{r}_t \int_{\mathrm{SO}(3)} \mathrm{d}g \int \mathrm{d}^{3N}\mathbf{x}_1 \, p(\mathbf{x}_1, g\mathbf{r}_t) \|\mathbf{x}_1\|^2.$$

So we can conclude that

$$\mathbf{D}_\theta^*(\mathbf{r}_t, t) = \int_{\mathrm{SO}(3)} \mathrm{d}g \int \mathrm{d}^{3N}\mathbf{x}_1 \, p(\mathbf{x}_1, g \mid \mathbf{r}_t) \, g^{-1} \cdot \mathbf{x}_1,$$

$$\mathbf{D}_\theta^*(g' \cdot \mathbf{r}_t) = \int_{\mathrm{SO}(3)} \mathrm{d}g \int \mathrm{d}^{3N}\mathbf{x}_1 \, p(\mathbf{x}_1, g \mid \mathbf{r}_t) \, g' \cdot g^{-1} \cdot \mathbf{x}_1, \forall g \in \mathrm{SO}(3).$$

Notice that

$$\mathbf{D}_\theta^*(\mathbf{r}_t) = \int_{\mathrm{SO}(3)} \mathrm{d}g \int \mathrm{d}^{3N}\mathbf{x}_1 \, p(\mathbf{x}_1, g \mid \mathbf{r}_t) \, g^{-1} \cdot \mathbf{x}_1$$

$$= \frac{\int_{\mathrm{SO}(3)} \mathrm{d}g \int \mathrm{d}^{3N}\mathbf{x}_1 \, p(g \cdot \mathbf{x}_1) p(g \cdot \mathbf{r}_t \mid g \cdot \mathbf{x}_1) \mathbf{x}_1}{\int_{\mathrm{SO}(3)} \mathrm{d}g \int \mathrm{d}^{3N}\mathbf{x}_1 \, p(g \cdot \mathbf{x}_1) p(g \cdot \mathbf{r}_t \mid g \cdot \mathbf{x}_1)}$$

$$= \frac{\int_{\mathrm{SO}(3)} \mathrm{d}g \int \mathrm{d}^{3N}\mathbf{x}_1 \, p(g \cdot \mathbf{x}_1) p(\mathbf{r}_t \mid \mathbf{x}_1) \mathbf{x}_1}{\int_{\mathrm{SO}(3)} \mathrm{d}g \int \mathrm{d}^{3N}\mathbf{x}_1 \, p(g \cdot \mathbf{x}_1) p(\mathbf{r}_t \mid \mathbf{x}_1)},$$

which is equivalent to the case $p_{\mathrm{target}} = \int_{\mathrm{SO}(3)} \mathrm{d}g \, p(g \cdot \mathbf{x}_1)$, *i.e.* using the augmentation by random SO(3) rotation.

### G.3 TRAINING AND SAMPLING ACCELERATION

In this subsection, we study the training and sampling convergence speed of different methods. For the training convergence speed comparison, we plot the generation performance measured by the precision AMR median metric with respect to the training epochs for previous heuristic alignment methods and our quotient-space diffusion model in Fig. G.3.1(Left). We only focus on the first 100 epochs for all the methods. These models are trained with the same architecture ET-Flow(SO(3)) and training configurations on the GEOM-DRUGS dataset. The results indicate that our method achieves a similar convergence speed to the AF3 heuristic method, because both methods reduce the learning difficulty of the model, and our method leads to a faster convergence than the GeoDiff alignment method since the latter only reduces the variance in learning a target in the equivalence class but has not removed this learning task, as shown in Table 1. We also notice that the AF3

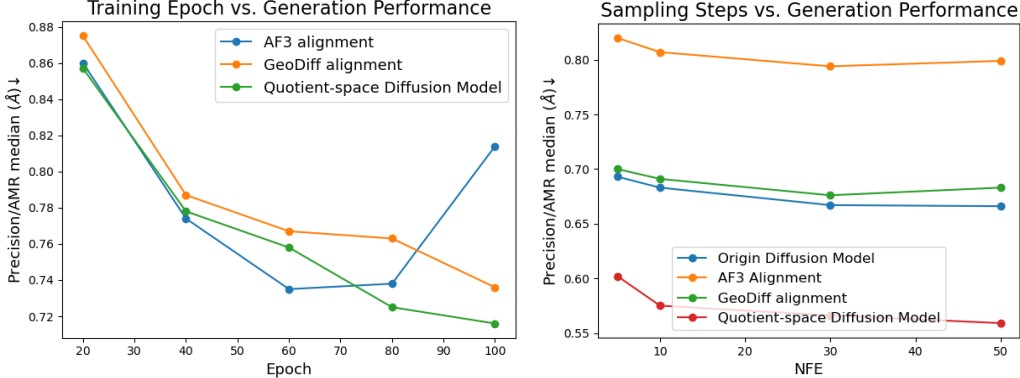

Figure G.3.1: Training and sampling convergence speed comparison among different symmetry training strategies using the ET-Flow(SO(3)) architecture on the GEOM-DRUGS dataset. **(Left)** The relationship between training epochs and generation performance measured by the precision AMR median metric. **(Right)** The relationship between the number of function evaluations (NFE) for sampling and generation performance measured by the precision AMR median metric.

alignment method starts to get worse generation performance after 80 training epochs. This happens due to the incompatibility between the modified learning target and the original sampling process.

For the sampling convergence speed comparison, we plot the generation performance measured by the precision AMR median metric with respect to the number of function evaluations (NFE) for the sampling process in Fig. G.3.1(Right). For all these methods trained on the GEOM-DRUGS dataset, we use the Flow Matching ODE sampler (Lipman et al., 2023) with Euler discretization. From the results, we can observe that models trained with different strategies exhibit similar convergence trends (performance gradually degrades as NFE decreases), our quotient-space diffusion framework consistently outperforms all baselines across every NFE setting.

### G.4    QUOTIENT SPACE BEYOND $\mathbb{R}^{3N}/\mathrm{SE}(3)$

Our framework can generalize to quotient spaces generated by symmetry groups beyond the special Euclidean group $\mathrm{SE}(3)$. Possible examples include the $\mathrm{U}(1)$ symmetry in quantum wavefunctions, the $\mathrm{SU}(2)$ symmetry in particle physics, and the $\mathrm{SO}(3)$ symmetry in higher ($> 3$) representation spaces for tasks including the mean-field electron Hamiltonian matrix prediction. In this work, we focus on the $\mathrm{SE}(3)$ case for its significant relevance to scientific research (Abramson et al., 2024). Applications of our framework on the mentioned more diverse systems above are left as future work.

### G.5    DISCUSSION ON TRANSITION PROBABILITY FOR REWARD-GUIDED GENERATION

Calculating transition probability is required in sampling from a reward-tilted distribution. It is a common need with significant practical relevance, *e.g.*, inference-time scaling for image generation (Singhal et al., 2025) and for protein structure generation (Wohlwend et al., 2025). We treat it also as an important future extension of our quotient-diffusion framework.

Regarding the transition probability calculation corresponding to the stochastic process in Eq. (9), we would like to first point out that an infinitesimal update step under say, Euler-Maruyama discretization, happens on a linear subspace at the current position defined by the projection operator $P_{\tilde{\mathbf{x}}_t}$ (note that $\tilde{\mathbf{h}}(\tilde{\mathbf{x}}_t)$ already lives in the subspace by definition); specifically, the Gaussian noise rising from the Wiener process is projected onto the subspace. A consequence of this operation is that the transition probability density function would be ill-defined if viewed as a distribution on the total space $\mathcal{M}$ (*i.e.*, it is not absolutely continuous with respect to the Lebesgue measure of the total space; it is infinite on the linear subspace and is zero outside). For an appropriate formulation in sequential-Monte-Carlo/reweighting-based tilting sampling, the distribution should be viewed on the subspace, and its density function be defined with respect to the Lebesgue measure on the subspace.

More concretely, using *simplified symbols*, the one-step transition can be seen as projecting the Gaussian random variable in the $M$-dimensional total space, $\mathbf{x} \sim \mathcal{N}(\boldsymbol{\mu}, \boldsymbol{\Sigma})$, using the projection in matrix form $\mathbf{P}$. This operation can be seen as marginalizing out the component orthogonal to the projected linear subspace. To leverage this view, we can first conduct a coordinate transformation that decouples the two components. This can be done from the diagonalization $\mathbf{P} = \mathbf{Q}^{-1}\boldsymbol{\Lambda}\mathbf{Q}$. Since $\mathbf{P}$ is a projection matrix satisfying $\mathbf{P}^2 = \mathbf{P}$, $\boldsymbol{\Lambda}$ is a diagonal matrix consisting only 1 and 0; denote the number of 1's as $Q$, which is the dimensionality of the linear subspace. Then written under the new coordinate system transformed by $\mathbf{Q}$, the projected coordinates is $\boldsymbol{\Lambda}\mathbf{Q}^{-1}\mathbf{x}$, which essentially omitting the last $M - Q$ components corresponding to the coordinates orthogonal to the linear subspace, so the distribution of the projected vector viewed in the $Q$-dimensional linear subspace is the distribution of $\mathbf{Q}^{-1}\mathbf{x}$ marginalized over the last $M - Q$ coordinates. Explicitly, $\mathbf{Q}^{-1}\mathbf{x} \sim \mathcal{N}(\mathbf{Q}^{-1}\boldsymbol{\mu}, \mathbf{Q}^{-1}\boldsymbol{\Sigma}\mathbf{Q}^{-\top})$, and its marginalized distribution for the first $Q$ coordinates is a Gaussian with mean vector taking the first $Q$ components of $\mathbf{Q}^{-1}\boldsymbol{\mu}$, and the covariance matrix as the top-left $Q \times Q$ block of the $M \times M$ matrix $\mathbf{Q}^{-1}\boldsymbol{\Sigma}\mathbf{Q}^{-\top}$. This is how the transition probability is calculated.

We would like to mention that Thm. 2 and Cor. 3 guarantee that this generation process still produces the same target distribution at the end when viewed on the quotient space (which is sufficient since we do not need to care about the distribution on the equivalent class).

As for the application of sequential-Monte-Carlo/reweighting-based approaches, the proposal distribution (if different from the transition distribution) must also be a distribution on the linear subspace, whose probability density should also be written in that space. The step-wise importance weight would be relatively flexible to choose as long as they accumulate to the desired reward at the end.

