# OpenReview forum: "Quotient-Space Diffusion Models"
_ICLR.cc/2026/Conference — ICLR 2026 Oral_

### Official Review · Reviewer_srZ8 · 2025-10-31

**Soundness:** 3
**Presentation:** 3
**Contribution:** 3
**Rating:** 6
**Confidence:** 3

**Summary:**

This paper addresses the problem of learning diffusion models for distributions with specific group symmetries. The proposed framework learns the score function on the quotient space with respect to the group, avoiding the need to model components corresponding to group actions. They start by projecting the standard forward SDE onto the quotient space using a Riemannian Ito's lemma. They then observe that the vector field on the quotient space admits an orthonormal decomposition into horizontal and vertical components, where only the horizontal component is relevant as the vertical component corresponds to infinitesimal group actions. Given the non-trivial geometry of the quotient space, they re-define a process in the original space that retains only the horizontal movement, called horizontal lift. Experiments focus on molecular structure generation, which is characterized by the SE(3) symmetry. Results demonstrate that training with this approach outperforms two alternative heuristics for handling symmetries.

**Strengths:**

The paper tackles the problem of handling symmetries in a novel and principled manner in contrast to previous approaches that rely on heuristics to deal with the symmetry problem. It is the first work I have seen that attempts to constrain the diffusion process within the quotient space.The authors also provide intuitive explanations of the main differences between their proposed method and existing approaches, supported by Table 1 and Section 3.4. Additionally, they include a comprehensive background on Riemannian geometry and stochastic calculus in the appendix to help readers follow the proofs of the theoretical claims presented in the main text. Experimental results demonstrate that applying their framework to both molecular and protein data improves the performance of the model compared to the use of conventional heuristic methods for handling symmetries.

**Weaknesses:**

I believe that, in principle, the approach proposed in the paper can be applied to any diffusion model by keeping the backbone architecture and changing the training, if I am not wrong. Therefore, it would have been interesting to see whether the improvements observed for the selected model also hold across different architectures. In relation to this, the results table could have been more informative if, for each baseline, the specific heuristics used to handle symmetries were explicitly stated.

Additionally, while the theoretical advantages of the proposed model are clearly illustrated in Table 1, I would have expected these advantages to be examined empirically as well, rather than being evaluated only based on final performance metrics. For example, I would expect a similar convergence behaviour between the proposed approach and the AF3 heuristic, since they are both removing the unnecessary DOFs and also lowering the variance. With respect to GeoDiff,I would expect the proposed method to achieve faster convergence during training, as it removes the equivalent DOFs.

Finally, the proposed framework rely on a horizontal projection operator that involves an inverse of a matrix that depends on the input dimensions, i.e. number of points in the cloud, or atom in the molecule. It would have been useful to include a discussion on whether this operation could become a bottleneck when scaling the method to larger molecules, and whether the matrix inversion might introduce numerical instability.

**Questions:**

- Just to clarify my understanding, could the first paragraph of Section 3.4 be related to [1], where the authors highlight a mismatch between using a translation-invariant score network and a target score that is not translation-invariant? In other words, given $x_t$ and $g\cdot x_t$, which are equivalent, the network is not able to distinguish between them while the corresponding target scores differ, leading to high variance during training?
- By avoiding the modeling of these unnecessary degrees of freedom using the proposed method, do you observe that this leads to better results when sampling with fewer steps compared to models trained using the other heuristics?

- line 961 'up' instead of 'bp'

[1] "Equivariant Diffusion for Crystal Structure Prediction" Lin et al, ICML 2024

---

> ### Author Response · Authors · 2025-11-25
> **Rebuttal (Part 1/3)**
>
> Thank you for your effort in reviewing our paper and your acknowledgment on our contributions from both the theoretical and empirical sides. We are glad to know that you believe that our method is novel and principled. We are also grateful for your valuable suggestions and insightful questions, which help us improve the quality of our work. Here are our responses to your questions and suggestions.
>
> ## Different Architectures
> Thank you for your suggestion. As you correctly mentioned, our framework can be applied to any diffusion model and backbone architecture. Our experiments use the ET-Flow(O(3)), ET-Flow(SO(3)) backbone on molecule structure generation experiments, and use a non-equivariant transformer-based architecture (following [1,2]) for protein backbone generation. We will conduct more experiments to further support this argument in the final version.
>
> Baselines reported in Table 4 are previous landmark works on protein backbone generation. Most of them, including FrameDiff, FrameFLow, FoldFlow, ESM3, and RFDiffusion, construct diffusion models directly on the group $\text{SE}(3)$. Thus, no alignment methods are used in these frameworks. The AF3 alignment is first introduced by the Alphafold3[1] paper, which is designed for constructing diffusion models on Euclidean space directly. We have added the baselines using alignment methods on Prote\'ina $\mathcal{M}_{\text{FS}}^{\text{small}}$ model [2] in the protein backbone generation task, and on the ET-Flow model in the Molecule structure generation task. We will conduct more experiments and add more baselines with different alignment methods in the final version.
>
> [1]. Josh Abramson,  et al. Accurate structure prediction of biomolecular interactions with AlphaFold 3. Nature, , 2024.
>
> [2]. Geffner, Tomas, et al. "Proteina: Scaling flow-based protein structure generative models." arXiv preprint arXiv:2503.00710 (2025).
>
> ## Advantages of our method
> Thank you for your valuable suggestions. Following your guidelines, we report the performance along the training trajectory in our revised paper (Fig.5(Left)). For the training convergence speed comparison, we plot the generation performance measured by the precision AMR median metric with respect to the training epochs for previous heuristic alignment methods and our quotient-space diffusion model. We only focus on the first 100 epochs for all the methods. These models are trained with the same architecture ET-Flow($SO(3)$) and training configurations on the GEOM-DRUGS dataset. As you have expected, our method achieves a similar convergence speed to the AF3 heuristic method, because both methods reduce the learning difficulty of the model, as shown in Table 1. This theoretical benefit leads to faster convergence than the GeoDiff alignment method. We also notice that the AF3 alignment method starts to get worse generation performance after 80 training epochs. This happens due to the incompatibility between the training loss and the generation performance metric, as the AF3 method is originally designed for the protein structure prediction task, which is not evaluated by distributional metrics.
>
> ## Computational cost and numerical issues in our method
> Thanks for your valuable question. Such a question is also mentioned by Reviewer Th24 and k95T. We would first clarify that, the matrix to be inverted is always a $3\times3$ matrix. It is only in constructing the $3\times3$ matrix that requires traversing over the atoms in a molecule, whose cost scales linearly in the number of atoms, and is negligible compared to the gradient propagation of a training neural network.
>
> Specifically, the matrix to be inverted is the $\mathcal{I}$ matrix in Thm. 4. The matrix $\mathcal{I}$ is constructed as the sum of the self-outer product of the 3-dimensional coordinates of atoms in a molecule, together with a weighted identity $3\times3$ matrix, so it is a $3\times3$ matrix requiring a linear complexity to construct. The inversion is ways on a $3\times3$ matrix, so the cost is constant. For the record, this cost is similar to the alignment algorithms (e.g., the Kabsch-Umeyama algorithm) adopted in the heuristic alignment-based diffusion strategies, which also involve a linear-complexity construction of a $3\times3$ matrix then conducting an inversion on it.
>
> As empirical evidence that the computational cost is negligible, we show the actual training time using our strategy compared to the vanilla training strategy and alignment-based methods in the following table:
>
> |Methods| Original diffusion | Geodiff alignment | Af3 alignment | Quotient space diffusion (Ours)|
> |-|-|-|-|-|
> |training speed (iters/s on an A100 GPU)| 4.19 | 4.07 | 4.08 | 4.10 |
>
> This result indicates that the additional computational cost brought by the additional operation is negligible. We have added the discussions above in the revised paper (Appx.F.1).

---

> > ### Author Response · Authors · 2025-11-25
> > **Rebuttal (Part 2/3)**
> >
> > As for the numerical instability issue, the matrix inversion may indeed be unstable for near-collinear systems of points. In practice, we add an $\epsilon \mathbf{I}$ term before conducting matrix inversion (i.e., we calculate $(\epsilon \mathbf{I} + \mathcal{I})^{-1}$ in practice). This treatment is widely adopted in algorithms facing similar situations, e.g., the practical implementation of the Kabsch-Umeyama algorithm for alignment.
> > Our typical choice is $\epsilon=1\times 10^{-8}$ and found that the training is well stable under this setting. We have shown the training curve of the model on the protein backbone generation task in our revised paper (Fig. 4), which indicates no numerical issue arising during the training process.
> >
> > ## About the related work
> > Thank you for recommending the insightful work [3], which is also recommended by Reviewer mNjY.
> > The work [3] highlighted the intrinsic periodic-translation symmetry that has been omitted for a long time in the field of periodic crystalline structure generation.
> > This can be trivially implemented using data augmentation (uniformly randomly translating the fractional coordinates), but the work instead designed a modified diffusion process which induces a transition kernel $p(x_t | x_0)$ that is invariant under periodic translation of $x_t$. The resulting optimization problem, while keeping the simplicity of no data augmentation, leads to a learning target for the score model that is invariant under periodic translation (at least when the data distribution is a single-point delta). Such a score model then leads to a periodic-translation invariant data distribution through the diffusion generation process.
> >
> > Intriguingly, this treatment can be systematically understood under our formulation. In particular, the modification to the training objective in [3] effectively conducts a translation on the input $x_t$ to make its center of mass (CoM) (in the sense on a ring, or hypertorus) located at the periodic origin, an operation called "CoM-free". This operation can be understood as aligning $x_t$ to a fixed reference (the periodic origin "0/1"), an operation similar to the "GeoDiff alignment" in Table 1 and Section 3.4. More concretely, by choosing a fixed reference (the periodic origin "0/1") for CoM, the direct learning samples ($\nabla_{\bar{\mathbf{F}}} \log q(\bar{\mathbf{F}} | \mathbf{F}_0)$ or $\bar{\mathbf{\epsilon}}$ in the original paper of [3]) for one $\mathbf{F}_0$ data point does not manifest a difference in the location of CoM, i.e., in the equivalent degrees of freedom (DOFs). Hence, it indeed achieves "Removal of variance on equivalent DOFs" in Table 1. For a step further, the optimization problem still asks a neural network model to learn to predict a specific target in the equivalent DOFs, i.e., the fixed location of CoM.
> >
> > As you may have expected, our quotient-space diffusion formulation can indeed be applied to this situation, where the total space ($\mathcal{M}$ in our paper) is the hypertorus and the group ($\mathcal{G}$ in our paper) is the periodic translation group.
> > Intriguingly, from our framework, we can explain the translation invariance of the score model, formally described in Lines 186-191: as $x_t$ lives in the sample space while $s_t(x_t)$ lives in the tangent space of the sample space at this $x_t$, the equivariance is formally be given by $s_t(g \cdot x_t) = g_* s_t(x_t)$, where $g_*$ is the induced transformation from a tangent vector at $x_t$ to a tangent vector at $g \cdot x_t$.
> > In the above specific case, a periodic translation $g$ induces an **identity** $g_*$, so the specific equivariance manifests an invariance $s_t(g \cdot x_t) = s_t(x_t)$.
> > Moreover, in this case, the projection operator $P_{\mathbf{x}}(\mathbf{v})$ amounts to setting the CoM of $\mathbf{v}$ to the CoM of $\mathbf{x}$ (instead of the periodic origin 0/1), a subtle difference from [3]. But with this operator, the model does not need to learn to predict a specific target in the location of CoM, the equivalent DOFs, corresponding to the last line of Table 1 and Equation (10).
> >
> > This is how we understand the relation to the work [3]. We have cited the work [3] and added discussions in the revised paper (L792-797).

---

> > > ### Author Response · Authors · 2025-11-25
> > > **Rebuttal (Part 3/3)**
> > >
> > > Regarding your specific questions, we are not sure if we have understood them correctly. Here we try to give an explanation, and we'd be glad if you could further specify your questions.
> > >
> > > * "a mismatch between using a translation-invariant score network and a target score that is not translation-invariant":
> > > In conventional diffusion for fractional coordinates generation, neither the score model nor the learning target is invariant. The work [3] wanted to implement a diffusion model that guarantees producing a periodic-translation invariant distribution. To achieve this, the authors designed a proper diffusion process (the one that uses the CoM-free operation in the forward process $q(\mathbf{F}_t | \mathbf{F}_0)$ that guarantees this. This diffusion process defines a translation-invariant score $\nabla \log q(\mathbf{F}_t)$, hence asks for a translation-invariant score model (the authors designed proper model architecture to ensure this). In the learning objective under the modified diffusion process (Eq. (19) and (20) in [3]), the target score samples ($\nabla \log q(\bar{\mathbf{F}} | \mathbf{F}_0)$ and $\bar{\mathbf{\epsilon}}$ in Eq. (19) and (20) in [3]) are all translation invariant.
> > > Trying to make a summary, the mismatch is between the previous conventional treatment and the desideratum of a translation-invariant distribution. Under the delicate design of [3], translation invariance is an ask on the score model, and the training process provides translation-invariant score targets.
> > >
> > > * "given $x_t$ and $g \cdot x_t$, which are equivalent, the network is not able to distinguish between them while the corresponding target scores differ, leading to high variance during training?":
> > > As mentioned, "the network is not able to distinguish between them" is an ask to generate translation-invariant samples, and the model architecture is intentionally designed for this. Under the designed diffusion process, "the corresponding target scores" are also translation-invariant (i.e., CoM-free), so there is no mismatch here. There is no variance among "the corresponding target scores" in the equivalent DOFs (i.e., location of CoM). But when compared to our method, the model is still asked to learn to predict a specific target in the equivalent DOFs (i.e., the fixed location of CoM), which is obviated in our quotient-space diffusion framework.
> > >
> > > [3] Equivariant diffusion for crystal structure prediction, ICML 2024
> > >
> > >
> > > ## Acceleration of the sampling process
> > > Thank you for your insightful question. For the sampling convergence speed comparison, we plot the generation performance measured by the precision AMR median metric with respect to the number of function evaluations (NFE) for the sampling process in Fig.5(Right). For all the methods, we use the Flow Matching ODE sampler with Euler discretization. From the results, we can observe that models trained with different strategies exhibit similar convergence trends (performance gradually degrades as NFE decreases), our quotient-space diffusion framework consistently outperforms all baselines across every NFE setting.
> > >
> > >
> > >
> > > ## About the typo
> > > Thank you for your careful reading. We have fixed this typo in the revised paper. Any further feedback is appreciated!
> > >
> > > ---
> > > Thank you for your detailed review again, and your valuable suggestions indeed help us improve the quality of our work. If you have any further questions or suggestions, we are glad to discuss with you.

---

### Official Review · Reviewer_mNjY · 2025-10-31

**Soundness:** 3
**Presentation:** 3
**Contribution:** 4
**Rating:** 8
**Confidence:** 3

**Summary:**

This paper addresses the critical problem of intrinsic system symmetries (e.g., $SE(3)$ symmetry in molecular generation) that complicates generative modeling. Through a theoretically rigorous approach, it establishes a formal framework for diffusion models on a general quotient space $\mathcal{M}/\mathcal{G}$. The method involves projecting the standard SDE onto this quotient space and subsequently deriving a new "horizontal lift" SDE in the original space, which is constrained to movements that are purely horizontal to the group action. This design effectively reduces the learning space for the neural network, thereby lowering the learning difficulty. Critically, this new SDE guarantees consistency between the training objective and the inference process, ensuring sampling compatibility, which prior heuristic alignment methods lack. The paper provides a detailed and persuasive implementation, with exhaustive comparisons against these heuristic approaches, demonstrating excellent performance on key benchmarks: small molecule structure generation (on GEOM-QM9 and GEOM-DRUGS) and protein backbone design.

**Strengths:**

1. **Significance and Theoretical Contribution**: This paper addresses a highly valuable and prevalent problem in generative modeling: the inconsistency between training objectives and sampling processes caused by intrinsic system symmetries. The work is exceptionally timely, as similar issues are being reported in various domains, such as periodic crystal generation [1, 2]. The authors are encouraged to cite similar works to further strengthen significance. The paper provides a powerful and unifying theoretical lens to explain and, more importantly, solve these inconsistencies.

2. **Elegant and Sound Framework**: The proposed framework is mathematically elegant and theoretically sound. The derivation—projecting the standard diffusion SDE onto the quotient space $\mathcal{M}/\mathcal{G}$ and then deriving its corresponding "horizontal lift" SDE back in the original space—is a principled and clean solution.

3. **Excellent Analysis of Prior and Heuristic Methods**: A major strength is the paper's insightful analysis of prior work within its new framework. The discussion (and Table 1) that masterfully categorizes and explains the shortcomings of previous methods (like data augmentation or inference-only alignment) is excellent. The critique of heuristic alignment strategies (GeoDiff, AF3) is particularly brilliant, clearly articulating how they introduce high variance or suffer from sampling incompatibility.

4. **Clarity and Intuition**: The paper is well-written. Figures 1 and 2, in particular, are exceptionally clear and provide excellent intuition for the complex mathematical concepts involved, making the core idea highly accessible.

5. **Strong Empirical Validation**: The authors provide thorough ablation studies and experiments on challenging, standard benchmarks (GEOM-QM9, GEOM-DRUGS, and protein design). The empirical results largely demonstrate the superiority of this principled approach over existing methods.

[1]"Equivariant diffusion for crystal structure prediction." ICML,2024

[2]"Kinetic Langevin Diffusion for Crystalline Materials Generation."ICML,2025

**Weaknesses:**

1. **Missing Discussion on GEOM-DRUGS Results**: In Table 3, the performance on the GEOM-DRUGS dataset, while strong, appears to be below that of the MCF baseline. This is a noteworthy result, but the paper does not provide any discussion or analysis for this specific comparison. A brief discussion of this would strengthen the experimental section.

2. **Missed Visualization Opportunity**: While Figures 1 and 2 are excellent for conceptual understanding, the paper would be significantly improved by a new figure that visualizes the dynamics of the proposed horizontal diffusion process itself. For example, a qualitative visualization of the distribution's evolution on a simple manifold (e.g., $\mathcal{M}/SO(2)$) would be highly illustrative and provide a powerful complement to the static diagrams.

3. **Minor Formatting Suggestion**: In Table 3, the results for "GEOM-DRUGS" are a key contribution. Consider using bold formatting or out-standing rows to make it more clearly.

**Questions:**

1. **Guarantee of Recovering the Full G-Invariant Distribution**: The paper's primary focus is to ensure the SDE matches the quotient space distribution and maintains sampling consistency. This is a clear success. However, a question remains about learning the full G-invariant target distribution $p(x)$. For a finite dataset (which is not perfectly G-invariant), does the horizontal lift SDE guarantee that it learns the correct probabilities for all equivalent points $g \cdot x$? For example, if the dataset contains only two samples $x_1, x_2$, the true target distribution should be non-zero and have the same probability on the entire orbit $ \\{g \cdot x_1, g \cdot x_2 | g \in \mathcal{G} \\} $. Does the proposed method recover this full manifold, or does it primarily learn the quotient-space projection well? A deeper discussion on this geometric probability coverage would be valuable.

2. **Comparison to Other Lie Group Diffusion Methods**: Could the authors comment on how this quotient-space framework compares, in terms of theoretical advantages or disadvantages, to other recent work on diffusion for symmetric data? For instance, methods that operate directly on Lie groups (e.g., "Trivialized momentum facilitates diffusion generative modeling on Lie groups," ICLR 2025, "Flow matching on general geometries."ICLR 2024) also aim to simplify the learning process.

---

> ### Author Response · Authors · 2025-11-25
> **Rebuttal (Part 1/2)**
>
> Thank you for your detailed review and for characterizing our contributions on both the theoretical and empirical aspects. We are glad to know that you believe that our method is mathematically elegant and that the analysis of heuristic alignment strategies is valuable. We also thank you for your valuable suggestions and insightful questions, which help us to improve the quality of our work. Here are our responses to your questions and suggestions.
>
> ## Related works
> Thank you for providing valuable references. We have carefully read the relevant work [1,2] and find that they have reported a similar issue in the crystal structure generation task. [1] highlighted the intrinsic periodic translation symmetry that has been omitted for a long time in the field of periodic crystalline structure generation. The work designed a modified diffusion process that induces a transition kernel that is invariant under periodic translation. The resulting optimization problem, while keeping the simplicity of no data augmentation, leads to a learning target for the score model that is invariant under periodic translation. [2] proposes a novel method that generalizes the Trivialized Diffusion Model framework for fractional coordinates to model the intrinsic periodic translation symmetry using flat coordinates. The proposed method considers the process with the velocity restricted to the mean-free linear subspace.
>
> Although considering different generation tasks, both of these works have a similar motivation to reduce the learning difficulty of the model using the intrinsic symmetry of the data distribution. We have carefully cited these works and give detailed discussions in the related work sections of our revised version of the paper (L792-802).
>
> [1] Equivariant diffusion for crystal structure prediction, ICML 2024
>
> [2] Kinetic Langevin Diffusion for Crystalline Materials Generation, ICML 2025
>
> ## Discussion on GEOM-DRUGS Results
> Thank you for your kind suggestion and feedback. We apologize for missing the necessary discussions. We have carefully added discussions on these experimental results in Sec.1 and Sec.4 in our revised paper (L96-100, L465-471).
>
> ## Visualization Examples
> Thank you for your suggestion. We agree that adding visualization examples may help the reader to have a better understanding of our methods. Following your guidance, we provide a visualization example of our quotient-space diffusion model corresponding to the quotient space $\mathbb{R}^2/SO(2)$ in the revised paper (Fig.1). In this example, the lifted process only has radial movements (Fig.1(Left)) as the quotient space $\mathbb{R}^2/SO(2)$ is isomorphic to the half real line and recovers the correct target distribution as conventional equivariant diffusion models (Fig.1(Middle, Right)). Any further feedback on the revised version of our paper will be appreciated.
>
> ## Formatting Issues
> Thank you for your kind suggestion. We apologize for missing bold formatting in the GEOM-DRUGS experiments. We have fixed the issue and improved the visualization in the revised paper.

---

> > ### Author Response · Authors · 2025-11-25
> > **Rebuttal (Part 2/2)**
> >
> > ## About the $\mathcal{G}$-Invariant Distribution
> > Thank you for your insightful question. First of all, our framework can recover the $\mathcal{G}$-invariant distribution under certain conditions. As mentioned in Theorem 1, we require the drift term $\mathbf{b}(\cdot,t)$ to be $\mathcal{G}$-equivariant, as only the equivariant vector field can be viewed as a vector field in the quotient space. Under this condition, the process can be viewed as a process in the quotient space.
> > In practice, Eqn.(6) is specified by Eqn.(3), which is the generation process. So we require the score function approximated by a neural network to be $\mathcal{G}$-equivariant. This can be achieved by either using a $\mathcal{G}$-equivariant network architecture or applying data augmentation to a non-equivariant architecture. In this work, we primarily focus on the learning target of the model and improving the training objective to reduce the learning difficulty, so we follow the conventional design of network architecture and training strategy.
> > One can prove that in these scenarios, the optimal diffusion model is $\mathcal{G}$-equivariant and can recover the $\mathcal{G}$-invariant data distribution when the model is perfectly trained (we present details in Appx.F.2).
> >
> > Although either using data augmentation or an equivariant architecture can recover the $\mathcal{G}$-invariant data distribution, these methods still need to learn the redundant target in the vertical space, which is unnecessary from the quotient space viewpoint. Our method further reduces the learning difficulty in the vertical space using the projection operator that can still recover the data distribution from the quotient space viewpoint (Corollary 3). Thus, our method can indeed recover the $\mathcal{G}$-invariant data distribution when using equivariant architecture (as we have implemented in GEOM-QM9 and GEOM-DRUGS datasets) or applying data augmentation for a non-equivariant architecture (as we have done in the protein backbone generation experiments).
> >
> > ## Comparison to Other Lie Group Diffusion Methods
> > Thank you for your insightful question. [3] is an interesting work that constructs the reverse of kinetic Langevin dynamics on a Lie group to perform generative modeling. The key insight is that the Brownian motion term is added only on the tangent space of the Lie group, which can be trivialized as an Euclidean space. However, our work considers the generation on the quotient space, in which the equivalent class under a group $\mathcal{G}$ is the same. Our core idea is to simulate the quotient-space diffusion model using the coordinates of the total space. We believe that [1] has the same high-level idea as our work that we try to construct generative models on a manifold using the coordinates of another space with simpler geometric structure. It's hard to compare these methods directly as they operate on different kinds of manifolds. However, generalizing our quotient space diffusion model to the kinetic Langevin dynamics (diffusion model with momentum) and simulating the process in the total space is a very interesting future work.
> >
> > [4] is a famous work that generalizes the flow matching framework to a Riemannian manifold. However, it requires an embedding that embeds the manifold into a Euclidean space and the explicit form of the geodesic. For a general manifold, constructing an embedding and calculating the closed-form geodesic are nontrivial tasks. In our work, we leverage the quotient structure to overcome these obstacles. As we have shown in Theorem 1 and 2, we can simulate the quotient-space diffusion process using the coordinates of the total space without requiring an embedding. In addition, we do not require a closed-form geodesic of the quotient space. In conclusion, our framework overcomes the obstacles of the general Riemannian diffusion model frameworks on the quotient manifold by leveraging the nice quotient structure.
> >
> > We have carefully cited and added discussions for these works in our related work section (L770-774).
> >
> > [3] Trivialized momentum facilitates diffusion generative modeling on Lie groups, ICLR 2025
> >
> > [4] Flow matching on general geometries, ICLR 2024
> >
> > ---
> > Thank you for your detailed review again, and your valuable suggestions indeed help us improve the quality of our work. If you have any further questions and suggestions, we are glad to discuss with you.

---

### Official Review · Reviewer_k95T · 2025-11-01

**Soundness:** 4
**Presentation:** 4
**Contribution:** 4
**Rating:** 10
**Confidence:** 3

**Summary:**

The paper investigates the often overlooked issue of additional training difficulty due to symmetries in diffusion models. The paper proposes considering the projected diffusion dynamics to the quotient space of interest in order to remove the original symmetries and simplify the learning problem. The paper provides a mathematical analysis of this new diffusion process and provide an efficient approach to training and sampling within this quotient space. This approach is then validated empirically in conformation generation for small molecules and protein backbone generation demonstrating improved results.

**Strengths:**

* I think the paper is very well done. It answers an important question (that I've also been considering) for diffusion models applied to common data modalities in scientific applications.

* The mathematical derivation of quotient space diffusion is handled nicely and the resulting training and sampling algorithms are also nicely implemented in the ambient space to sidestep the complex form of the quotient space.

* Very nice summary of previous approaches of handling the additional difficulty of learning symmetries in Section 3.4, as well as an explanation as to why there are issues with sampling time mismatches. This is a useful exposition as this has been less emphasised in the literature.

* The empirical evaluation is well done with definite benefits reported for the relevant and practically important examples of conformation generation for small molecules and protein backbone generation.

**Weaknesses:**

* It is perhaps less clear how SMC/reweighing-based approaches for reward tilting could applied to this framework as I am not totally sure that transition probabilities computed from equation 9 represent the correct transition probabilities that you would want from the actual quotient space diffusion.

**Questions:**

* Are there ever numerical instabilities from the projection operator or the horizontal lift of mean curvature due to matrix inverse for near collinear systems of points?

---

> ### Author Response · Authors · 2025-11-25
> **Rebuttal (Part 1/2)**
>
> Thank you for your acknowledgment on our contributions from both the theoretical and empirical sides! We are encouraged to know that you have also been considering the question, and feel fulfilled to contribute an analysis on the question and a formal solution to the community.
> We also appreciate your insightful questions for improving our work further. Here are our responses to your questions.
>
> ## Reward tilting in our quotient-space diffusion framework
> Thank you for your insightful question! Sampling from the reward-tilted distribution is indeed a common need with significant practical relevance,
> e.g., inference-time scaling for image generation [1] and for protein structure generation [2]. This would be an important future extension in our quotient-diffusion framework as well.
>
> Regarding your specific question about the transition probability calculation corresponding to the stochastic process in Equation (9), we would like to first point out that an infinitesimal update step under say, Euler-Maruyama discretization, happens on a linear subspace at the current position defined by the projection operator $P_{\tilde{\mathbf{x}}_t}$ (note that $\tilde{\mathbf{h}}(\tilde{\mathbf{x}}_t)$ already lives in the subspace by definition); specifically, the Gaussian noise rising from the Wiener process is projected onto the subspace. A consequence of this operation is that the transition probability density function would be ill-defined if viewed as a distribution on the total space (i.e., it is not absolutely continuous with respect to the Lebesgue measure of the total space; it is infinite on the linear subspace and is zero outside). For an appropriate formulation in SMC/reweighting-based tilting sampling, the distribution should be viewed on the subspace, and its density function be defined with respect to the Lebesgue measure on the subspace.
> More concretely, using simplified symbols, the one-step transition can be seen as projecting the Gaussian random variable in the $n$-dimensional total space, $\mathbf{x} \sim \mathcal{N}(\mu, \Sigma)$, using the projection in matrix form $\mathbf{P}$. This operation can be seen as marginalizing out the component orthogonal to the projected linear subspace. To leverage this view, we can first conduct a coordinate transformation that decouples the two components. This can be done from the diagonalization $\mathbf{P} = \mathbf{Q}^{-1} \Lambda \mathbf{Q}$. Since $\mathbf{P}$ is a projection matrix satisfying $\mathbf{P}^2 = \mathbf{P}$, $\Lambda$ is a diagonal matrix consisting only 1 and 0; denote the number of 1's as $m$, which is the dimensionality of the linear subspace. Then written under the new coordinate system transformed by $\mathbf{Q}$, the projected coordinates is $\Lambda \mathbf{Q}^{-1} \mathbf{x}$, which essentially omitting the last $n-m$ components corresponding to the coordinates orthogonal to the linear subspace, so the distribution of the projected vector viewed in the $m$-dimensional linear subspace is the distribution of $\mathbf{Q}^{-1} \mathbf{x}$ marginalized over the last $n-m$ coordinates.
> Explicitly, $\mathbf{Q}^{-1} \mathbf{x} \sim \mathcal{N}(\mathbf{Q}^{-1} \mu, \mathbf{Q}^{-1} \Sigma \mathbf{Q}^{-\top})$, and its marginalized distribution for the first $m$ coordinates is a Gaussian with mean vector taking the first $m$ components of $\mathbf{Q}^{-1} \mu$, and the covariance matrix as the top-left $m\times m$ block of the $n\times n$ matrix $\mathbf{Q}^{-1} \Sigma \mathbf{Q}^{-\top}$. This is how the transition probability is calculated.
> We would like to mention that Theorem 2 and Corollary 3 guarantee that this generation process still produces the same target distribution at the end when viewed on the quotient space (which is sufficient since we do not need to care about the distribution on the equivalent class).
>
> As for the application of SMC/reweighting-based approaches, the proposal distribution (if different from the transition distribution) must also be a distribution on the linear subspace, whose probability density should also be written in that space. The step-wise importance weight would be relatively flexible to choose as long as they accumulate to the desired reward at the end.
>
> [1] Singhal, Raghav, et al. "A general framework for inference-time scaling and steering of diffusion models." arXiv preprint arXiv:2501.06848 (2025).
>
> [2] Wohlwend, Jeremy, et al. "Boltz-1 democratizing biomolecular interaction modeling." BioRxiv (2025): 2024-11.

---

> > ### Author Response · Authors · 2025-11-25
> > **Rebuttal (Part 2/2)**
> >
> > ## Numerical issues from the projection operator
> > Thank you for your question. As you have mentioned, the matrix inversion may have numerical issues for near-collinear systems of points. In practice, we add an $\epsilon \mathbf{I}$ term before conducting matrix inversion, that is, we calculate $(\epsilon \mathbf{I} + \mathcal{I})^{-1}$ in practice, where $\mathbf{I}$ is the $3\times3$ identity matrix.
> > This treatment is widely adopted in algorithms facing similar situations, e.g., the practical implementation of the Kabsch-Umeyama algorithm for alignment.
> > Our typical choice is $\epsilon=1\times 10^{-8}$ and found that the training is well stable under this setting. We have shown the training curve of the model on the protein backbone generation task in our revised paper (Fig.4 in Appx.F.1), which indicates no numerical issue arises during the training process. We have carefully added the discussions above in Appx.F.1.
> >
> > ---
> > Thank you for your questions and kind words again. If you have any further questions and suggestions, we are glad to discuss with you.

---

> > > ### Comment · Reviewer_k95T · 2025-11-26
> > >
> > > Thanks for addressing my questions. I will maintain my score. Good luck with the rest of rebuttals!

---

### Official Review · Reviewer_Th24 · 2025-11-02

**Soundness:** 4
**Presentation:** 3
**Contribution:** 3
**Rating:** 6
**Confidence:** 2

**Summary:**

This paper establishes a formal mathematical framework for diffusion modeling on quotient spaces, with applications to molecular structure generation under SE(3) symmetry. The key idea is to construct diffusion processes directly on the quotient space M/G, then lift them back to the total space M for practical implementation using horizontal projections. The framework reduces learning difficulty by removing redundant components corresponding to group actions while guaranteeing correct sampling from the target distribution.

**Strengths:**

1. The paper provides a comprehensive mathematical framework grounded in Riemannian geometry and stochastic calculus. Theorems 1-4 give explicit characterizations of the projected diffusion process and its horizontal lift.
2. The formulation of diffusion models via horizontal lifting on quotient spaces is innovative. It generalizes previous work on equivariant diffusion and provides a unifying geometric perspective that can be applied to various symmetry groups.
3. Unlike heuristic alignment strategies, the proposed method guarantees correct sampling from the target distribution while reducing learning difficulty by removing redundant degrees of freedom.

**Weaknesses:**

1. The paper doesn't discuss the computational cost of computing the horizontal projection operator $P_x$ and mean curvature vector $h$ at each step. For the SE(3) case, these involve matrix inversions and cross products - what is the actual time comparison?
2.  The heavy reliance on differential geometry, Riemannian manifolds, Lie groups, and stochastic calculus makes the paper inaccessible to much of the machine learning community. While mathematical rigor is valuable, the presentation could benefit from more intuitive explanations alongside formal derivations.
3.  While consistent, the improvements over baselines are relatively small in some cases, which raises questions about practical significance.

**Questions:**

1. How does the method perform on other symmetry groups beyond SE(3)?
2. How does the computational cost of projection scale? Are there approximations or efficient implementations for large-scale problems?

---

> ### Author Response · Authors · 2025-11-25
> **Rebuttal (Part 1/2)**
>
> Thank you for your effort in reviewing our paper. We are glad to know that you believe our method is innovative, principled, and effective. We also appreciate your constructive suggestions on how to improve our work further. Below are our detailed responses to your questions.
>
> ## The computational cost of our framework (W1 and Q2)
>
> Thanks for your valuable question. We give a detailed discussion on the computational cost of our method. As you have mentioned, we need to compute the inversion of the matrix $\mathcal{I}$ and the cross product for the horizontal projection operator $P_\mathbf{x}$ and the mean curvature vector $\tilde{\mathbf{h}}(\mathbf{x})$ (Thm 4). For the calculation of $\mathcal{I}^{-1}$, notice that $\mathcal{I}$ is always a $3\times 3$ matrix, so construction cost of $\mathcal{I}^{-1}$ is only linear $O(N)$, where $N$ is the number of atoms (linear $O(N)$ cost for constructing $\mathcal{I}$, and constant $O(1)$ cost for inversion). The cross product is conducted atom-wise, so its computational cost is also linear $O(N)$. So we can conclude that the overall computational complexity is $O(N)$ for both $P_\mathbf{x}$ and $\tilde{\mathbf{h}}(\mathbf{x})$.
>
> We would like to mention that the alignment operation adopted in the heuristic alignment-based diffusion strategies also has the same complexity. To see this, for aligning $\mathbf{x}\in\mathbb{R}^{3\times N}$ towards $\mathbf{y}\in\mathbb{R}^{3\times N}$, the Kabsch-Umeyama algorithm constructs the optimal rotation matrix as $(\mathbf{H}^T \mathbf{H})^{\frac{1}{2}} \mathbf{H}^{-1}$, where $\mathbf{H}:=\mathbf{y} \mathbf{x}^T \in \mathbb{R}^{3\times 3}$ requires a linear $O(N)$ cost. In practice, the $O(N)$ computational cost is negligible compared to the cost of gradient back-propagation through the neural network. We provide a comparison of practical training times in the following table.
>
> |Methods| Original diffusion | GeoDiff alignment | Af3 alignment | Quotient space diffusion (Ours)|
> |-|-|-|-|-|
> |training speed (iters/s on an A100 GPU)| 4.19 | 4.07 | 4.08 | 4.10 |
>
> From the results, we can see that the additional computational cost brought by the additional operation is negligible. We have added the discussions above in the revised paper (Appx.F.1).
>
> ## More intuitive explanations (W2)
> Thank you for your emphasis on this point. We are well aware of the difficulty in delivering the mathematical principles, the principled algorithms, and the difference from prior work, and we had iterated a few versions before the submission to better allow general audience to receive the gist and the unique contributions from the main paper, for example, reducing similar concepts and terms (e.g., 'orbit' and 'fiber' had all been replaced with 'equivalent class'), using perceivable descriptions (e.g., 'the tangent vector in the tangent space of the fiber' had been replaced by 'the movement within the equivalent class'), and providing illustrative figures (Fig. 2 and 3) to help understand the concepts and claims with a solid, visual example.
>
> We have further added some intuitive explanation along with the theoretical derivations in Sec.3.1, which are marked blue in our revised text. These explanations further illustrate the motivation and intuitive understanding of the theoretical results. We have also added a visualization example of the equotient-space diffusion model in the $\mathbb{R}^2/\text{SO(2)}$ case (Fig.1). Any further feedback on the revised version of our paper will be appreciated.

---

> > ### Author Response · Authors · 2025-11-25
> > **Rebuttal (Part 2/2)**
> >
> > ## About the empirical performance (W3)
> > Thank you for bringing up this question. Firstly, we suppose your evaluation on the practical significance might be based on the improvement in the absolute values, for which we would like to mention that the relative improvements, explicitly shown in the following tables, are worthy of certain significance, especially considering that this is merely from our new training and sampling paradigm while using the same neural network architecture.
> >
> > |Metrics on GEOM-QM9| Recall AMR (mean) (Å; ↓) |  Recall AMR (median)(Å; ↓) | Precision AMR (mean)(Å; ↓) | Precision AMR (median)(Å; ↓) |
> > |-|-|-|-|-|
> > |Original Diffusion Model| 0.076 | 0.030 | 0.110 | 0.047 |
> > |Quotient space diffusion model| 0.069 | 0.024 | 0.096 | 0.036 |
> > |Relative improvement| 9% | 20% | 13% | 23% |
> >
> > |Metrics on GEOM-DRUGS| Recall AMR (mean)(Å; ↓) |  Recall AMR (median)(Å; ↓) | Precision AMR (mean)(Å; ↓) | Precision AMR (median)(Å; ↓) |
> > |-|-|-|-|-|
> > |Original Diffusion Model(reproduced)| 0.541 | 0.515 | 0.724 | 0.665 |
> > |Quotient space diffusion model| 0.477 | 0.455 | 0.635 | 0.563 |
> > |Relative improvement| 12% | 12% | 13% | 15% |
> >
> > |Metrics (all unitless) on Foldseek AFDB clusters| FPSD vs. PDB(↓) | FPSD vs. AFDB(↓) | FJSD vs. PDB(↓) | FJSD vs. AFDB(↓) |
> > |-|-|-|-|-|
> > |Original Diffusion Model| 83.2 | 21.9 | 0.58 | 0.12 |
> > |Quotient space diffusion model| 69.9 | 17.6 | 0.41 | 0.11 |
> > |Relative improvement| 16% | 20% | 30% | 9% |
> >
> > Moreover, we would like to point out that the empirical improvement increases as the molecular structure generation task becomes harder, from molecules with less than 9 heavy atoms, to larger organic molecules, to biomolecules like proteins. This indicates that the practical value of our method becomes more significant on more challenging systems, which is an attractive advantage for more practically relevant problems.
> >
> > ## Application to symmetry groups beyond $\mathrm{SE}(3)$ (Q1)
> > Thank you for your valuable question. As we mentioned in Section 3, our framework can indeed generalize to
> > quotient spaces generated by symmetry groups beyond the special Euclidean group $\mathrm{SE}(3)$. Possible examples include the $\mathrm{U}(1)$ symmetry in quantum wavefunctions, the $\mathrm{SU}(2)$ symmetry in particle physics,
> > and the $\mathrm{SO}(3)$ symmetry in higher (>3) representation spaces for tasks including the mean-field electron Hamiltonian matrix prediction.
> > In this work, we focus on the $\mathrm{SE}(3)$ case for its significant relevance to scientific research (the AlphaFold structure prediction tool even won the Nobel prize!). We will continue exploring its applicability in the mentioned more diverse systems. We have added some discussions in Appx.F.4.
> >
> > ---
> > We thank you again for your efforts in reviewing our paper, and we have replied to each of your concerns. We sincerely look forward to your re-evaluation of our submission based on our responses and updated results.

---

### Meta-Review · Area_Chair_viwh · 2026-01-07

**Summary:**

This paper presents a clean, principled framework for diffusion on quotient spaces (motivated by SE(3) symmetries in molecular/protein generation) and a practical “lift” back to the ambient space via horizontal projections. Reviewers broadly agree the theory is solid and the idea meaningfully improves over heuristic alignment by removing redundant degrees of freedom and aligning training with sampling. Empirically, the method shows consistent gains on standard molecule and protein benchmarks. Overall, the consensus supports acceptance.

**Reviewer Concerns:**

Addressed: the rebuttal clarified computational cost through both asymptotic analysis and measured training speed, discussed numerical stability (regularized inversion), improved intuition/readability with added explanations/visuals, and included targeted discussions for specific empirical comparisons and related work. Outstanding: broader demonstrations beyond SE(3) and more backbones would be nice.

**Reviewer Scores:**

The strongest positive review explicitly confirmed after the rebuttal that their questions were answered and they would keep their high score. The two borderline-positive reviewers raised the main practical questions (cost, stability, accessibility, and depth of empirical evidence). The rebuttal provided direct answers with measured runtimes, explicit complexity, added figures on training/sampling behavior, and expanded explanations, so it is plausible they would tick upward slightly (or at least become more confident in their original weak-accept stance).
The already-positive reviewer who highlighted the framework’s significance and clarity would likely remain positive and possibly increase confidence given the added related-work discussion and the new visualization/analysis they requested.

---

### Decision · Program_Chairs · 2026-01-26

Accept (Oral)